# From Generative to Episodic:
# Sample-Efficient Replicable Reinforcement Learning

Max Hopkins [*1]   Sihan Liu [*2]   Christopher Ye [*2]   Yuichi Yoshida [*3]

## Abstract

The epidemic failure of replicability across empirical science and machine learning has recently motivated the formal study of replicable learning algorithms [Impagliazzo et al. (2022)]. In contrast to batch settings (i.e. data comes from a fixed i.i.d. source) where the cost of replicability is relatively well understood, there remain significant gaps in our knowledge for control settings like reinforcement learning where an agent must interact directly with a shifting environment. Indeed, there is a large gap between the best upper bound of $\tilde{O}(S^7 A^7)$ [Eaton et al. (2023)] for RL with exploration, and $\tilde{O}(S^2 A^2)$ [Karbasi et al. (2023)] for the RL 'batch' setting. This gap raises a key question in the broader theory of replicability: Is replicable exploration inherently more expensive than batch learning? Is sample-efficient replicable RL even possible?

In this work, we (nearly) resolve this problem (for low-horizon tabular MDPs): exploration is not a significant barrier to replicable learning! Our main result is a replicable RL algorithm on $\tilde{O}(S^2 A)$ samples, bridging the gap between the generative and episodic settings. We complement this with a lower bound in the episodic setting of $\tilde{\Omega}(S^2)$ showcasing the near-optimality of our algorithm with respect to the state space $S$.

## 1. Introduction

Out of a 2016 survey of $1500$ scientists (Baker, 2016), $70\%$ could not replicate the results of their colleagues; $50\%$ could not even replicate their own. In recent years the trend has only worsened, driven (in part) by the wide adoption of powerful but volatile techniques from machine learning (Ball, 2023). Motivated in this context, (Impagliazzo et al., 2022) introduced a formal framework to study the feasibility of *replicability in learning*. (Impagliazzo et al., 2022) call an algorithm $\mathcal{L}$ *replicable* if it satisfies the following strong stability guarantee: run twice on independent samples from the same underlying environment, $\mathcal{L}$ should produce *exactly the same output* with high probability. (Impagliazzo et al., 2022) showed replicability is achievable for several basic learning tasks provided the algorithm shares internal randomness between runs, raising the broader question: what problems can we solve replicably, and at what cost?

In the *batch* setting (problems such as mean estimation, distribution testing, or PAC learning where data comes in an i.i.d. batch), the statistical cost of replicability is now fairly well understood (Impagliazzo et al., 2022; Bun et al., 2023; Kalavasis et al., 2023; Hopkins et al., 2024; Liu & Ye, 2024; Kalavasis et al., 2024). The story changes, however, for *control* problems like reinforcement learning (RL) where an agent interacts *dynamically* with their environment. Here replicability seems to hit a barrier coming from the cost of exploration: for underlying environments (tabular MDPs) with $S$ states and $A$ actions, (Karbasi et al., 2023) showed *without* exploration (i.e., with access to a generative model of the environment), one can learn a near-optimal interaction strategy (policy) in $O(S^2 A^2)$ samples, while the best known upper bound *with* exploration is $O(S^7 A^7)$ (Eaton et al., 2023). Intuitively, replicable exploration indeed seems inherently harder than learning with a generative model, as the agent is subject to the environment's random variations (an issue that has plagued RL for over 30 years (White III & Eldeib, 1994; Mannor et al., 2004; Henderson et al., 2018)). Nevertheless there is no known lower bound separating the two, and given RL's widespread use in the sciences (Gow et al., 2022; Neftci & Averbeck, 2019; Martín-Guerrero & Lamata, 2021) and machine learning (e.g., in LLMs (Ouyang et al., 2022; Havrilla et al., 2024)), understanding the contrast is key to developing a broader theory of replicability:

> *Is exploration truly a barrier to efficient replicability?*

[1]Institute of Advanced Study, Princeton, NJ, USA [2]UC San Diego, La Jolla, CA, USA [3]National Institute of Informatics, Tokyo, Japan. Correspondence to: Sihan Liu <sil046@ucsd.edu>, Christopher Ye <czye@ucsd.edu>.

*Proceedings of the 43rd International Conference on Machine Learning*, Seoul, South Korea. PMLR 306, 2026. Copyright 2026 by the author(s).

In this work, we (nearly) resolve this question (for constant horizon MDPs): surprisingly, exploration is *not* a significant barrier to replicability! We give a replicable algorithm with no generative model on only $\tilde{O}(S^2 A)$ samples, and a matching lower bound in the generative setting under the common assumption that the agent samples the same number of transitions and rewards from each state-action pair (sometimes called the *parallel sampling model* (Kearns & Singh, 1998)). We complement the latter with an unconditional lower bound of $\tilde{\Omega}(S^2)$ in the episodic (no generative model) setting, showing near-optimality of our algorithm with respect to the state space $S$.

Drawing on ideas from the reward-free RL literature (Jin et al., 2020; Zhang et al., 2020), the main technical innovation of our work is a new way of separating exploration and decision making in policy estimation that allows us to make the 'output' of exploration replicable without actually requiring a fully replicable exploration process.[1] In slightly more detail, the exploration phase of our algorithm outputs a set of 'critical states' of the environment which are 1) easy to sample from, and 2) hold enough value that we can learn a good policy on just these states, essentially reducing the problem to the generative model setting. We produce such a set by first finding a 'fractional' set of critical states by estimating the mean of a non-replicable algorithm, then arguing the fractional solution can be replicably rounded to an integer one without significant loss in relevant parameters.

## 1.1. Background

We briefly recall the following standard notions from reinforcement learning. A (finite horizon, tabular) Markov Decision Process (MDP) is a tuple $\mathcal{M} = \{\mathcal{S}, \mathcal{A}, H, \mathbf{p}_0, \{\mathbf{p}_h(s,a)\}, \{\mathbf{r}_h(s,a)\}\}$ where $\mathcal{S}$ is a finite set of states, $\mathcal{A}$ is a finite set of actions, $H \in \mathbb{Z}_+$ is the time horizon, $\mathbf{p}_0$ is the initial distribution over $\mathcal{S}$, $\{\mathbf{p}_h(s,a)\}$ are the transition probabilities (over $\mathcal{S}$) for a given state-action pair $(s,a)$ at time $h \in [H]$, and $\{\mathbf{r}_h(s,a)\}$ are similarly reward distributions over $[0,1]$.

An agent interacts with the MDP $\mathcal{M}$ in the following fashion, called an *episode*. The agent starts at an initial state $x_1 \sim \mathbf{p}_0$. At time $1 \leq h \leq H$, the agent chooses an action $a \in \mathcal{A}$, receives reward $r \sim \mathbf{r}_h(x_h, a)$, and moves to state $x_{h+1} \sim \mathbf{p}_h(s,a)$ and time $h+1$. The episode terminates after the agent receives the $H$-th reward. We call a transition-reward pair observed in this process a *sample*.

A *policy* is a strategy for the agent, i.e., a set of functions

$\pi_h : \mathcal{S} \to \mathcal{A}$ dictating what action they choose given the current time step and state. The *value* of a policy $\pi = \{\pi_h\}$, denoted $V(\pi, \mathcal{M})$, is its expected total reward over the randomness of the transitions, rewards, and starting position above. Finally, a policy $\pi$ is said to be $\varepsilon$-*optimal* if its value is within $\varepsilon$ of the best possible strategy, that is $V(\pi, \mathcal{M}) \geq \max_{\pi'} V(\pi', \mathcal{M}) - \varepsilon$.

Following the standard high probability PAC-style model (Kakade, 2003), our goal is to build an algorithm which after sufficiently many episodes of interaction with any unknown MDP, outputs an $\varepsilon$-optimal policy with high probability.

**Definition 1.1** (PAC Policy Estimation). An algorithm $\mathcal{L}$ is called an $(\varepsilon, \delta)$-PAC policy estimator with sample complexity $n = n(S, A, H, \varepsilon, \delta)$ if given any $S, A, H \in \mathbb{Z}_+$, $\varepsilon, \delta > 0$, and episodic sample access to an MDP $\mathcal{M} = \{\mathcal{S}, \mathcal{A}, H, \mathbf{p}_0, \{\mathbf{p}_h(s,a)\}, \{\mathbf{r}_h(s,a)\}\}$, $\mathcal{L}$ outputs an $\varepsilon$-optimal policy with probability at least $1 - \delta$ after observing at most $n$ samples.[2]

Following (Karbasi et al., 2023; Eaton et al., 2023), we call an algorithm *replicable* if trained twice on the same MDP with shared randomness, it outputs exactly the same policy with high probability. While conceptually simple, this is fairly cumbersome to notate due to the interaction of the agent and environment, so for the moment we will opt for the following slightly informal treatment and refer the reader to Appendix B for a more exact formulation.

**Definition 1.2** (Replicable PAC Policy Estimation). An $(\varepsilon, \delta)$-PAC policy estimator $\mathcal{A}$ is called $\rho$-replicable if for every MDP $\mathcal{M}$, $\mathcal{A}$ outputs the same policy over two independent executions on $\mathcal{M}$ with high probability:

$$\Pr_{\mathcal{M}_1, \mathcal{M}_2, \xi}[\mathcal{A}(\mathcal{M}_1; \xi) = \mathcal{A}(\mathcal{M}_2; \xi)] \geq 1 - \rho$$

where $\mathcal{M}_1$ and $\mathcal{M}_2$ are independent copies of the MDP $\mathcal{M}$ and $\xi$ is a shared random string.

## 1.2. Our Results

In this work we focus on MDPs in the low-horizon setting, where $H$ should be thought of as much smaller than the number of states (e.g., polylogarithmic, or at most subpolynomial). We emphasize this is typical in settings RL is applied.[3] With this in mind, our main result is the first nearly sample-optimal replicable policy estimator in this regime:

---

[1] By 'fully replicable exploration process', note we do not mean a process in which the algorithm always explores the same sequence of states over two runs, but rather methods like (Eaton et al., 2023) which keep the internal states of the algorithm, e.g., various visitation statistics of state-action pairs, replicable throughout the entire exploration phase.

[2] The sample complexity of the algorithm is determined by the number of transitions and rewards seen. That is, an algorithm sees $2H$ samples in a full episode.

[3] Games, for instance, usually have exponentially many board positions (state space) in the number of turns (horizon), e.g., the average chess game ends in 40 moves, but has order of $10^{40}$ (indeed more) states (Shannon, 1950). Other classic examples include robotics, where specific tasks often last only a few minutes but are subject to massive variety in environment.

**Theorem 1.3** (Sample Efficient Episodic RL (Informal Theorem F.1))**.** *There is a $\rho$-replicable $(\varepsilon, \delta)$-PAC policy estimator in the episodic setting with sample complexity*

$$n(S, A, H, \varepsilon, \delta) \leq \frac{S^2 A}{\rho^2 \varepsilon^2} \cdot \mathrm{poly}\left(H, \log(1/\delta)\right).$$

We complement Theorem 1.3 with two corresponding lower bounds. First, we consider the 'generative model' setting. Recall an algorithm $\mathcal{A}$ has access to a *generative model* for MDP $\mathcal{M}$ if it can request a sample transition and reward from any state-action pair $(s, a) \in \mathcal{S} \times \mathcal{A}$ directly. Most algorithms in the generative model actually fall into a slightly more restricted setting called 'parallel sampling' which requires the algorithm query each state-action pair the same number of times (one thinks of getting a 'parallel sample' from the entire set of state-action pairs at once, see Definition B.1). Our first lower bound shows $O(S^2 A \rho^{-2} \varepsilon^{-2})$ is tight in this setting.

**Theorem 1.4** (Parallel Sampling Lower Bound)**.** *Any $\rho$-replicable algorithm that is a $(\varepsilon H, 0.001)$-PAC[4] policy estimator requires $\tilde{\Omega}\left(\frac{S^2 A H^2}{\rho^2 \varepsilon^2}\right)$ samples in the parallel sampling model.*

The key technical contribution behind this result is a new sample complexity lower bound for the replicable ($n$-dimensional) sign-one-way marginals problem that is optimal up to logarithmic factors in $n, \rho, \varepsilon$ (Theorem G.2). This resolves an open question of (Bun et al., 2023) and has a number of other immediate applications, including (near)-tight lower bounds for replicable PAC learning (Theorem I.9) and $\ell_1$-mean estimation (Theorem I.8).

While sample-adaptivity (the ability to query different state-action pairs a different number of times) is not particularly useful in standard reinforcement learning (see e.g., the nearly optimal non-adaptive algorithm of (Sidford et al., 2018)), we are not able to rule out the possibility of better sample-adaptive algorithms in the replicable case. To account for this, we also prove a slightly weaker but unconditional lower bound for replicable reinforcement learning in the episodic setting:

**Theorem 1.5** (Episodic Lower Bound)**.** *Any $\rho$-replicable $(\varepsilon H, 0.001)$-PAC policy estimator in the episodic setting must have a sample complexity of $\tilde{\Omega}\left(\frac{S^2 H^2}{\rho^2 \varepsilon^2}\right)$.*

In settings where both $A$ and $H$ are small (a more restrictive but not uncommon regime, e.g., games like Chess or Checkers), Theorems 1.3 and 1.5 essentially resolve the sample complexity of replicable reinforcement learning. It is also

worth noting that Theorem 1.5 is the first unconditional lower bound showing replicable reinforcement learning is actually asymptotically harder than standard reinforcement learning (at least when $S \gg A\sqrt{H}$), which has sample complexity $\tilde{\Theta}(SAH^3)$ (Domingues et al., 2021; Zhang et al., 2024). Prior lower bounds in (Karbasi et al., 2023) used similar ideas, but rely on an unproven conjecture on the sample complexity of replicable bias estimation and were proven under stronger optimality assumptions on the policy estimator (see Section 1.5 for further discussion).

*Remark* 1.6. Theorem 1.3 is not computationally efficient (see Theorem F.1 for runtime details) due to invoking several common sample-efficient subroutines from the replicability literature that have no known efficient implementation. In Theorem H.5, we give a computationally efficient variant of our algorithm at the cost of using slightly more samples (roughly $\frac{S^3 A}{\varepsilon^2 \rho^2}$).

### 1.3. Our Techniques

We now overview the main components in the proofs of Theorems 1.3 to 1.5.

Our algorithm for Theorem 1.3 takes inspiration from the reward-free RL literature (Jin et al., 2018; Zhang et al., 2020) which splits policy estimation into two main components: an 'exploration' stage that identifies 'critical' states in the MDP and gathers sufficiently many samples from them, and a 'decision-making' stage that computes an $\varepsilon$-optimal policy given access to these i.i.d. samples. At a high level, we can roughly view the first stage as building some 'limited generative model' for the MDP via strategic exploration, and the second stage as a replicable algorithm for PAC policy estimation in the generative model setting. As such, our first step is to give an algorithm on $\tilde{O}(S^2 A \cdot \mathrm{poly}\,(H))$ samples in the parallel sampling setting. We will later show how to modify the argument to the weaker generative model actually constructed by our exploration process.

#### 1.3.1. PARALLEL SAMPLING

**Multi-Instance Best-Arm ($H = 1$):** Let us start with an even simpler setting, where $H = 1$ and the initial distribution $\mathbf{p}_0$ is uniform over all states. Here, since the agent only picks a single action and never transitions states, it is not hard to see that policy estimation simply reduces to solving $S$ instances of the 'best-arm problem' on $A$ arms, i.e., for each state we are given access to i.i.d. samples from $A$ reward distributions, and would like to replicably compute an action (arm) that is within $\varepsilon$ of the best possible reward with high probability.

Consider just a single such instance (i.e., for $S = 1$). To solve this case, we adapt an algorithm for replicable PAC learning of (Ghazi et al., 2021) which uses the *correlated*

---

[4]While the error $\varepsilon$ is often thought of as a small constant, recall in the finite horizon setting it can range from 0 to $H$, so $\varepsilon H$ can be thought of as constant 'multiplicative' error here.

*sampling* paradigm (Holenstein, 2007), a procedure that reduces the problem of replicability to outputting a *distribution $D$* over actions such that with high probability 1) $D$ typically outputs an $\varepsilon$-optimal action and 2) run twice on independent samples, the output distributions $D_1$ and $D_2$ are close in total variation distance. Formally, correlated sampling is an algorithm $\mathcal{A}$, given distributions $\mu_1, \mu_2$ and (shared) randomness $r$, guarantees that (1) the marginals are correct i.e., $\mathcal{A}(\mu_i, r) \sim \mu_i$ for $i \in \{1, 2\}$ and (2) $\Pr(\mathcal{A}(\mu_1, r) \neq \mathcal{A}(\mu_2, r))$ is bounded by (twice) the total variation distance of $\mu_1, \mu_2$. See Lemma B.12 for details.

Following (Ghazi et al., 2021), we observe this can be done by sampling an action from the so-called *exponential mechanism* from differential privacy. Namely, for the right scaling factor $t$, drawing roughly $\frac{\log^3 A}{\rho^2 \varepsilon^2}$ samples per action and outputting the action $a$ with probability proportional to $\mu_a \sim \exp(t\hat{r}_a)$, where $\hat{r}_a$ is the empirical reward of action $A$, satisfies the desired guarantees.

We'd now like to extend this procedure to solving $S$ parallel instances of the best-arm problem. In general, it is always possible to solve $S$ instances of a problem replicably simply by running a single instance replicable algorithm $S$ times with replicability parameter $\rho/S$. In our case this leads to a policy estimator on roughly $S^3 A \varepsilon^{-2} \rho^{-2}$ samples — too much for our desired sample complexity.

To extend this procedure to solving $S$ parallel instances of the best-arm problem, we directly (correlated) sample from the *joint* mechanism of all $S$ instances over $[A]^S$, i.e., over vectors of actions $(a_1, \ldots, a_S) \in [A]^S$ with rewards $\sum_{i=1}^{S} r_{a_i}$. A (fairly) elementary extension of (Ghazi et al., 2021)'s analysis additionally leveraging tensorization of $\chi^2$ distance shows that drawing only $\tilde{O}(S \varepsilon^{-2} \rho^{-2})$ samples per state-action pair (i.e., roughly $S^2 A \varepsilon^{-2} \rho^{-2}$ in total) suffices for replicability in this case (see Appendix C.1 for formal details).[5]

**Backward Induction (General $H$):**   We'd now like to use the above procedure to solve an $H$-step MDP. This is done via a replicable variant of the standard 'backward induction' algorithm (Puterman, 2014). We remark that a similar strategy (replicable 'phased value iteration') is employed in (Eaton et al., 2023), but they incur a sample cost of $S^3 A^3$ due to estimating each state-action pair independently while we are able to improve upon the sample complexity by estimating only the state value function replicably.

1.3.2. THE EPISODIC SETTING

**Reachability and Reward-Free RL:**   To apply parallel sampling in the episodic setting, we need to give an exploration process that, through direct interaction with the MDP environment, generates enough i.i.d. samples for each state-action pair to act as a generative model. Unfortunately, this is impossible as stated: one cannot hope to collect a uniform number of samples from each state-action pair in the episodic setting as some states may be inherently difficult (even impossible) to reach via interaction with the MDP. On the other hand, given a policy $\pi$, if $\pi$ almost never reaches a state $s$ on the MDP it must be that the choice of action on $s$ has almost no effect on $\pi$'s value. This means we might hope to deal with the above issue by simply ignoring the set of 'hard to reach' states, and run our parallel sampling algorithm on the remainder from which we can collect enough samples.

This is exactly the core observation of (Jin et al., 2020; Zhang et al., 2020) in the so-called *reward-free* RL model. In their work, (Jin et al., 2020; Zhang et al., 2020) define the 'reachability' of a state $s \in S$, denoted $\delta(s)$, to be the maximum probability of reaching $s$ from the starting position under any policy.[6] They observe that the states can be partitioned into 'reachable' states with $\delta(s) \geq \frac{\varepsilon}{SH^2}$, and 'ignorable' states with $\delta(s) < \frac{\varepsilon}{SH^2}$. An elementary argument shows that running backward induction over reachable states results in an $\varepsilon$-optimal policy. Finally, (Zhang et al., 2020) show one can generate i.i.d. samples from the reachable states by simulating (Jin et al., 2018)'s $Q$-learning agent on a version of the MDP with zero reward. Namely, they show after $K$ episodes every state is sampled at least $\frac{K\delta(s)^2}{H^2 SA}$ times.

This suggests a natural two step strategy for replicable RL: replicably compute the set of reachable states, then collect sufficient i.i.d. samples from them to run replicable backward induction. This hits two immediate roadblocks. First, it is not at all a priori clear how to replicably generate the reachability partition (at least sample-efficiently). Second, even if it were, backward induction requires $SA$ samples per (reachable) state, so applying (Jin et al., 2020; Zhang et al., 2020)'s machinery directly results in the use of $O(\frac{S^4 A H^7}{\varepsilon^4 \rho^2})$ samples, a far cry from our desired dependence.[7]

---

[5]An alternative approach is to run the single-instance replicable algorithm $S$ times independently with replicability parameter $\rho/S$. Doing so avoids the exponential runtime of the performing correlated sampling on the distribution supported on $[A]^S$ but incurs a sub-optimal sample complexity of $S^3 A \varepsilon^{-2} \rho^{-2}$.

[6]Technically one defines reachability for each step $h \in H$, but we ignore this detail here for simplicity.

[7]We remark (Zhang et al., 2020) do give a more sample-efficient algorithm for task-agnostic RL than is implied by their reachability bounds (also via Q-learning on the zero-reward MDP), but their result does not come with any guarantee certifying reachability. Our method critically relies on identifying some form of reachable/ignorable states, so a priori we cannot use their improved analysis in our setting.

**Ignorable State Combinations:** Our main technical contribution is the introduction and analysis of a relaxed notion of 'reachability partition' that side-steps both these issues. In particular, we design an algorithm **RepExplore** that simultaneously **(i)** replicably identifies a relaxed *ignorable state combination*, a collection of subsets of states $\{\mathcal{I}_h\}_{h=1}^H$ such that for some optimal policy $\pi^*$

$$\sum_{h \in [H]} \Pr_{\mathcal{M}} [x_h \in \mathcal{I}_h \mid \pi^*] \ll \frac{\varepsilon}{H}, \qquad (1)$$

and **(ii)** collects a sufficient number of samples for any state $x \notin \mathcal{I}_h$ at step $h$ to run replicable backward induction.[8]

It is worth briefly comparing this to (Jin et al., 2020; Zhang et al., 2020)'s reachability partition before moving on. Much like the standard partition, it is not hard to see property **(i)** ensures the total contribution of $\{\mathcal{I}_h\}_{h=1}^H$ to $\pi^*$'s value is at most $\varepsilon$, while property **(ii)** ensures we can find a policy nearly as good as $\pi^*$ on the remaining states via a modified backward induction. Critically, *unlike* the reachability partition, ignorable state combinations can actually be generated sample-efficiently. This is subtle to show (and will be discussed in the following sections), but for the moment to highlight the difference at least observe that (given an $\varepsilon$-approximation to $\pi^*$) there is clearly a highly efficient *verification* procedure to test if $\{\mathcal{I}_h\}_{h=1}^H$ is an ignorable state combination (just run $\pi^*$ for roughly $\frac{H}{\varepsilon}$ episodes), whereas verifying a potential reachability partition is much more expensive (e.g. requiring at least $S^2 \frac{H}{\varepsilon}$ episodes). The rest of this section is devoted to arguing this intuition carries over to *generating* ignorable states, which can be done in only $\frac{SA}{\varepsilon^2} \cdot \text{Poly}(H)$ samples, along with our fractional rounding strategy that enforces replicability of the procedure up to a tight (multiplicative) overhead of $O(\frac{S}{\rho^2})$ samples.

**Design and Analysis of RepExplore:** We now overview our exploration procedure **RepExplore**. Our starting point is the concept of an *under-explored* state. Consider a (non-replicable) PAC policy estimator $\mathcal{A}_{rl}$ in the episodic setting. After an execution of $\mathcal{A}_{rl}$ on the MDP, we call a state $s \in \mathcal{S}$ *under-explored* at step $h$ if $s$ has any action that was never taken by the agent at step $h$, and write $\mathcal{U}_h$ to denote the set of under-explored states of $\mathcal{A}_{rl}$ at step $h$.

Intuitively, it is reasonable to expect the set of under-explored states $\{\mathcal{U}_h\}_h$ of an optimal policy estimator $\mathcal{A}_{rl}$ should be ignorable. Informally, if a state $s$ were 'reachable' but has an untested action $a$, the reward of $(s, a)$ could contribute significantly to the value of the optimal policy while $\mathcal{A}_{rl}$ has no way to account for this. Moreover, any explored state has been visited by $\mathcal{A}_{rl}$ at least $A$ times, meaning we might reasonably expect to collect many sam-

ples from such states. The reality of course is not so simple, but we will show that the under-explored states of (Jin et al., 2018)'s Q-learning agent after only $O(SA \cdot \text{poly}(H))$ episodes on the zero-reward MDP indeed form an ignorable state combination, and moreover that running this procedure $O(S \cdot \text{poly}(H))$ times suffices to generate enough samples for replicable backward induction.

This leaves the issue of replicability. The key is to realize we can use the above non-replicable process to compute a relatively 'stable' *fractional* ignorable state combination, then *replicably round* the fractional solution back to a discrete one. In more detail, the idea is to compute a vector of estimates $\hat{\mu}_{s,h}$ of the probability that $s \in \mathcal{U}_h$ according to $\mathcal{A}_{rl}$. By running $\mathcal{A}_{rl}$ enough times we can get a highly concentrated estimate of this 'mean' vector and round it to an actual solution for $\{\mathcal{I}_h\}$ by including each $s \in \mathcal{I}_h$ with probability $\hat{\mu}_{s,h}$ (we will argue below this procedure still outputs an ignorable combination with reasonable probability). The crux is that if over two runs our fractional estimates $\{\hat{\mu}_{s,h}^{(1)}\}$ and $\{\hat{\mu}_{s,h}^{(2)}\}$ are close, the resulting distributions over rounded state combinations $\{\mathcal{I}_h\}$ must be close as well, allowing us to apply correlated sampling to produce exactly the same collection with high probability.

We now cover each step of this strategy in greater details, split into three main parts (i) replicability of $\{\mathcal{I}_h\}$, (ii) ensuring $\{\mathcal{I}_h\}$ indeed has low-reachability under some optimal policy $\pi^*$ (Equation (1)), and (iii) how to collect enough samples from critical states $x \notin \mathcal{I}_h$ at step $h$.

**Fractional Solutions and Replicable Rounding:** As discussed, to ensure replicability we apply correlated sampling over the "average outcome" of the under-explored states of $\mathcal{A}_{rl}$. In particular, denote by $\mu_{s,h}$ the likelihood that $s \in \mathcal{S}$ is under-explored by $\mathcal{A}_{rl}$ at step $h$ in one full execution of the algorithm, and let $\mathcal{B}(\mu)$ denote the 'rounded' distribution over $\{\mathcal{I}_h\}_{h=1}^H$ given by independently adding each $s$ into $\mathcal{I}_h$ with probability $\mu_{s,h}$. A fairly elementary calculation shows that running $\mathcal{A}_{rl}$ roughly $k = \tilde{\Theta}(SH\rho^{-2})$ times suffices to get empirical estimates $\hat{\mu}^{(1)}$ and $\hat{\mu}^{(2)}$ over independent runs satisfying $d_{TV}(\mathcal{B}(\mu^{(1)}), \mathcal{B}(\mu^{(2)})) \le \rho$, allowing us to apply correlated sampling to get the desired replicability (see Lemma E.11 for details). Since each run of $\mathcal{A}_{rl}$ consumes $\tilde{\Theta}(SA \cdot \text{poly}(H) \varepsilon^{-2})$ many episodes, this gives us a total sample complexity of $\tilde{\Theta}(S^2 A \cdot \text{poly}(H) \rho^{-2} \varepsilon^{-2})$.

**Rounding Preserves Ignorability:** We now overview the argument that our candidate ignorable state combination $\mathcal{I}_h$ sampled in the way described above actually satisfies $\sum_h \Pr[x_h \in \mathcal{I}_h \mid \pi^*] \ll \varepsilon/H$ for some optimal policy $\pi^*$. Since the distribution of $\mathcal{I}_h$ is defined according to the mean of the distribution over under-explored state combinations $\{\mathcal{U}_h\}$ of $\mathcal{A}_{rl}$, the first step is to argue this would at least be true if we were to directly set $\{\mathcal{I}_h\} = \{\mathcal{U}_h\}$, i.e., there exists

---

[8]Note the 'critical states' mentioned in the introduction are simply $\mathcal{S} \setminus \mathcal{I}_h$.

some optimal policy $\pi^*$ such that $\sum_h \Pr[x_h \in \mathcal{U}_h \mid \pi^*] \ll \varepsilon/H$. In fact, we will give an agent whose under-explored states are ignorable with respect to *every* policy with high probability. However this is by far the most involved part of the argument, so we defer it for the moment and assume for now the under-explored states of $\mathcal{A}_{rl}$ are indeed ignorable.

Given this assumption, observe that if we were to run $\mathcal{A}_{rl}$ $k$ times and output $\{\mathcal{I}_h\}$ uniformly from the resulting under-explored state combinations $\{\mathcal{U}_h^{(1)}\}, \cdots, \{\mathcal{U}_h^{(k)}\}$, $\mathcal{I}_h$ would also be trivially ignorable under any policy with high probability. In reality, we are sampling $\{\mathcal{I}_h\}$ from the product distribution $\mathcal{B}(\hat{\mu})$, where $\hat{\mu}$ are computed using $\{\mathcal{U}_h^{(1)}\}, \cdots, \{\mathcal{U}_h^{(k)}\}$. Now, condition on an arbitrary collection of under-explored state combinations $\{\mathcal{U}_h^{(1)}\}, \cdots, \{\mathcal{U}_h^{(k)}\}$ that are ignorable, and denote by $\mathcal{E}$ the empirical distribution over them. Though $\mathcal{B}(\hat{\mu})$ and $\mathcal{E}$ are by no means close, the key is to realize that the distributions do have the same *marginals* by construction since $\hat{\mu}_{s,h}$ is exactly the probability of $s \in \mathcal{U}_h$ for $\{\mathcal{U}_h\} \sim \mathcal{E}$. In other words, for any state $x$, we have $\Pr_{\{\mathcal{U}_h\} \sim \mathcal{E}}[x \in \mathcal{U}_h] = \Pr_{\{\mathcal{I}_h\} \sim \mathcal{B}(\hat{\mu})}[x \in \mathcal{I}_h]$. Since reachability is defined as a linear sum over individual state probabilities, this means the expected reachability under $\mathcal{B}(\hat{\mu})$ and $\mathcal{E}$ are also identical. That is, for any policy $\pi$:[9]

$$\mathbb{E}_{\{\mathcal{I}_h\} \sim \mathcal{B}(\hat{\mu})} \left[ \sum_h \Pr_{\{x_h\} \sim \mathcal{M}^\pi} [x_h \in \mathcal{I}_h] \right]$$
$$= \mathbb{E}_{\{x_h\} \sim \mathcal{M}^\pi} \left[ \sum_h \Pr_{\{\mathcal{I}_h\} \sim \mathcal{B}(\hat{\mu})} [x_h \in \mathcal{I}_h] \right]$$
$$= \mathbb{E}_{\{x_h\} \sim \mathcal{M}^\pi} \left[ \sum_h \Pr_{\{\mathcal{U}_h\} \sim \mathcal{E}} [x_h \in \mathcal{U}_h] \right]$$
$$= \mathbb{E}_{\{\mathcal{U}_h\} \sim \mathcal{E}} \left[ \sum_h \Pr_{\{x_h\} \sim \mathcal{M}^\pi} [x_h \in \mathcal{U}_h] \right] \ll \frac{\varepsilon}{H},$$

where the last inequality is by the conditioning that all $\{\mathcal{U}_h^{(i)}\}$ are ignorable. It then follows from Markov's inequality that the ignorable state combination $\{\mathcal{I}_h\}$ sampled according to $\mathcal{B}(\hat{\mu})$ should have low-reachability under some optimal policy $\pi^*$ with high constant probability (we remark it is enough to achieve constant success since, as is typically the case, it is easy to later boost the success and replicability of the algorithm to $1 - \delta$ and $\rho$ respectively at a $\text{polylog}(\delta^{-1}\rho^{-1})\rho^{-2}$ cost in sample complexity).

**Optimistic Exploration in the Reward-Free MDP:** So far we have critically relied on the assumption that the under-explored state combination $\{\mathcal{U}_h\}$ of $\mathcal{A}_{rl}$ is an ignorable state combination with high probability. We now discuss our

implementation of $\mathcal{A}_{rl}$, an algorithm we call **QEXPLORE**, in more detail and sketch its analysis to this effect.

We first briefly recall the policy estimator from (Jin et al., 2018). At a high level, for each tuple $(s, a, h) \in (\mathcal{S}, \mathcal{A}, [H])$, the algorithm maintains an estimate $Q_h(s, a)$ for the $Q^*$-*value* $Q_h^*(s, a)$ (the maximum value achieved over policies starting at time-step $h$, state $s$, and playing action $a$). These estimates are initialized to $H$ (the maximum total reward of the MDP) and updated whenever the agent visits the state-action pair $(s, a)$ at step $h$. In particular, at the $t$-th time of visiting $(s, a)$ at $h$, the algorithm updates according to the formula: $Q_h(x, a) \leftarrow \left(1 - \frac{H+1}{H+t}\right) Q_h(x, a) + \frac{H+1}{H+t} \left( r + \max_{a'} Q_{h+1}(s_{nxt}, a') + \Theta(1) \sqrt{\frac{H^3 \log(SAH)}{t}} \right)$, where $r$ denotes the reward received, and $s_{nxt}$ denotes the next state after transition. With these estimates in hands, the learner simply follows the policy that greedily picks $\arg\max_a Q_h(s, a)$ at each state. The additive term $\sqrt{H^3 \log(SAH)/t}$ is known in the literature as *optimistic bonus*. This bonus is large at the beginning and gradually decreases as the visitation count $t$ for the tuple $(s, a, h)$ increases. Its primary role is to encourage the algorithm to optimistically explore states that are not yet fully explored. The other term $r + \max_{a'} Q_{h+1}(s_{nxt}, a')$ aligns conceptually with the backward induction step in the parallel sampling setting. As $t$ increases, the optimistic bonus vanishes and one expects the estimate to converge to the optimal value $Q_h^*(x, a)$. This intuition is formalized in the proof of Theorem 1 in (Jin et al., 2018), which employs martingale concentration techniques and a carefully constructed induction argument.

For **QEXPLORE**, we borrow the idea of (Jin et al., 2020; Zhang et al., 2020) and simply run (Jin et al., 2018)'s algorithm but on a *simulated* MDP environment where all rewards have been set to 0, and otherwise has identical transition probabilities as the original MDP. Intuitively, this further discourages exploitation and encourages the RL algorithm to explore as many new states as possible.

Assume for simplicity our starting distribution over the MDP is supported on a single state $x_{ini}$ (as it is easy to reduce to this case). The central notion in our analysis is the concept of the *reachability function* $R_h^k(s)$, which roughly corresponds to the maximum likelihood over all policies of landing in some under-explored state (with respect to the first $k$ episodes) when starting from state $s$ at step $h$. In particular, our goal is then to prove strong quantitative bounds on the convergence of $R_1^k(x_{ini})$ to 0 as $k$ grows large, since this exactly states the probability of reaching the under-explored states under any policy is small as desired.

Denote by $Q_h^k(s, a)$ the $Q$ estimate of the tuple $(s, a, h)$ at

---

[9]Below we write $\{x_h\} \sim \mathcal{M}^\pi$ to denote the random sequence of states the agent lands in following policy $\pi$.

the beginning of episode $k$. Our key lemma (see Lemma E.6) shows that with high probability:

$$R_h^k(s) \le \max_a Q_h^k(s, a) \quad \forall k \ge 0, h \in [H], s \in \mathcal{S}. \quad (2)$$

The result then follows since (Jin et al., 2018) show that $Q_1^k(x_{\text{ini}}, a)$ converges to $Q^*(x_{\text{ini}}, a)$ for the initial state $x_{\text{ini}}$ after roughly $O(SA \cdot \text{poly}(H))$ episodes, but the rewards of our simulated MDP (and therefore also $Q^*(x_{\text{ini}}, a)$) are 0, guaranteeing rapid convergence of $R_1^k(x_{\text{ini}})$ to 0 as well.

It remains to show Equation (2). If $s$ is under-explored, there is some action $a$ not taken at $s$ and the inequality follows (witnessed by $a$) as long as we set the initial optimistic bonus to be sufficiently large, i.e., on the order of $H$. Otherwise, we take advantage of the fact that $Q_h^k(s, a)$ is always updated as an affine combination of $Q_h^{k-1}(s, a)$ and $Q_{h+1}^{k-1}(s', a')$, where $(s', a')$ is the next state-action pair taken by the agent in step $h+1$ of episode $k$, and $R_h^k(s)$ is a monotonically decreasing function in $k$ (since the under-explored sets never grow). This allows us to use martingale concentration and a careful induction on $k$ to conclude the proof. The full argument is quite technical but conceptually similar to the original argument in (Jin et al., 2018) showing that $Q_h^k(s, a)$ is an upper confidence bound on $Q_h^*(s, a)$ so we omit the details here.

**Non-uniform Sample Collection** Our final step is to ensure we can collect sufficiently many samples from each non-ignorable state $s \notin \{\mathcal{I}_h\}$ to learn an $\varepsilon$-optimal policy by backward induction. Since the sample collection process for different time steps can be analyzed independently, for simplicity, we focus just on the last step $h = H$ in the overview.

Initially, a reasonable goal would be to collect roughly $S \cdot \text{poly}(H) \varepsilon^{-2} \rho^{-2}$ many samples from each state-action pair over the non-ignorable states, since this is needed for our replicable algorithm in the parallel sampling setting. Unfortunately, this is not realizable in our current framework. To see this, consider even the simplest non-trivial setting for policy estimation: $A = 2$ and $H = 1$. We may construct an MDP such that there is a subset $\mathcal{G} \in \mathcal{S}$ of half the states each with probability $\Theta\left(\frac{1}{S^{1.5}}\right)$ of occurring under the starting distribution. Note that since our total sample budget is only $\tilde{O}(S^2)$, we cannot hope to collect $\Omega(S)$ samples from any fixed $s \in \mathcal{G}$ with non-vanishing probability.

On the other hand, we argue it is actually reasonably likely that some $s \in \mathcal{G}$ ends up *outside* the final ignorable state combination $\mathcal{I}_H$ sampled, therefore requiring us to collect sufficiently many samples from $s$. To see this, first note that since the final state combination $\mathcal{I}_\mathcal{H}$ is sampled from the product distribution $\mathcal{B}(\hat{\mu})$, the expected number of $s \in \mathcal{G}$ outside $\mathcal{I}_H$ is exactly $\mathbb{E}[|\mathcal{G} \setminus \mathcal{I}_H|] = \sum_{s \in \mathcal{G}}(1 - \hat{\mu}_{s,H})$, or in other words, the average number of $s \in \mathcal{G}$ fully explored by

**QEXPLORE** per run (where the randomness is over the MDP). We argue the right-hand side is at least $\Omega(1)$. Namely, since **QEXPLORE** consumes $\Omega(S)$ samples per run, the probability of seeing any individual $s \in \mathcal{G}$ in a given run is at least $\Omega(S^{-1/2})$, and by the birthday paradox we are likely to see some $s \in \mathcal{G}$ twice over the $S$ runs in total. Since $A = 2$, such $s$ are considered 'fully explored', so the average number of explored states in $\mathcal{G}$ per run is $\Omega(1)$ as claimed.

To circumvent the issue, we exploit the fact that our multi-instance best-arm (and therefore backward induction) algorithm actually works under the weaker assumption that it receives a *variable* number $m_s$ of samples per state-action pair $(s, a)$ so long as $\{m_s\}$ jointly satisfy the inequality

$$\sqrt{\sum_{s \in \mathcal{S} \setminus \mathcal{I}_H} \frac{1}{m_s}} \ll \varepsilon \rho, \quad (3)$$

where the left hand side roughly corresponds to the total variation distance between the (joint) exponential mechanism distributions from which the final outputs are sampled across two runs of the algorithm.

Now, let us re-examine the sample collection process for the set of states $\mathcal{G}$. Though there may be a few states in $s \in \mathcal{G} \setminus \mathcal{I}_H$, it is not hard to show there will be at most $|\mathcal{G} \setminus \mathcal{I}_H| \le O(\sqrt{S})$ with high probability. This means we can collect fewer samples for these states, i.e., we can set their $m_s$ to $\tilde{O}(\sqrt{S} \varepsilon^{-2} \rho^{-2})$, while maintaining $\sqrt{\sum_{s \in \mathcal{G} \setminus \mathcal{I}_H} 1/m_s} \le O(\varepsilon \rho)$. This sample collection goal can be easily achieved — each $s \in \mathcal{G}$ has probability $S^{-1.5}$ of occurring, so in $\tilde{\Theta}(S^2 \varepsilon^{-2} \rho^{-2})$ episodes we are indeed likely to collect $\Theta(\sqrt{S} \varepsilon^{-2} \rho^{-2})$ such samples as desired.

More generally, we will actually set the sample budget $m_s$ at step $h$ to be roughly $S \varepsilon^{-2} \rho^{-2} \cdot \text{poly}(H) \cdot (1 - \mu_{s,h})$. Recall that $(1 - \mu_{s,h})$ denotes the likelihood of a state $s$ being explored by **QEXPLORE**. It then follows from standard concentration arguments that such a sample collection goal can be achieved by running **QEXPLORE** for $\tilde{O}(S \varepsilon^{-2} \rho^{-2} \cdot \text{poly}(H))$ many times. Though we are now collecting fewer samples from states that are less likely to be explored by $\mathcal{A}_{rl}$, the fact that they are less likely to be explored also ensures that they will only be excluded from $\{\mathcal{I}_H\}$ with small probability. Thus, we can argue that on average the bound $\sqrt{\sum_{s \in \mathcal{S} \setminus \mathcal{I}_h} 1/m_s}$ (where the randomness is over $\mathcal{I}_h$) remains small, ensuring that the sample size requirement of the multi-instance best-arm algorithm is met.

**Optimize $\varepsilon$ dependency via Tiered State Partition** As described, the above algorithm incurs sub-optimal dependency on $\varepsilon$ (e.g. we can neither ignore or collect enough samples from a state reached with probability $\sqrt{\varepsilon}$). In our final algorithm, we use a more fine-grained definition of

reachability and partition the state space into 'tiers' of reachability, ensuring that we are only required to collect fewer samples from less reachable states as their contribution to the optimal value function is also conceivably lower.

### 1.4. Lower Bounds

We briefly overview our lower bounds for the parallel sampling and episodic models, which follow via a reduction from the so-called *sign-one-way marginals* problem: given sample access to a product of Rademacher distributions over $\{\pm 1\}^n$ with mean $p = (p_1, \ldots, p_n)$, compute a vector $v \in [-1, 1]^n$ satisfying $\frac{1}{n} \sum_{i=1}^{n} v_j p_j \geq \frac{1}{n} \sum_{i=1}^{n} |p_j| - \varepsilon$. Our first result (Theorem G.2) shows that any $\rho$-replicable $\varepsilon$-accurate algorithm for the sign-one-way marginals problem requires $\tilde{\Omega}(n\rho^{-2}\varepsilon^{-2})$ (vector) samples. This resolves an open problem of (Bun et al., 2023), improving upon their bound of $\tilde{\Omega}(n)$.

Finally, we need to reduce sign-one-way marginals to reinforcement learning. We start with the parallel sampling setting for the special case $A = 2$.[10] Consider an instance of the $S$-dimensional sign-one-way marginal problem. We can define a single-step MDP $\mathcal{M}$ with $S$ states and action set $\{a_+, a_-\}$ with expected reward $+p_s$ and $-p_s$ respectively for each state $s$. It is not hard to see that we can simulate a parallel sample from $\mathcal{M}$ by generating two vector samples for the sign-one-way marginal problem $w^{(1)}$ and $w^{(2)}$, and taking the rewards for $(s, a_+)$ as $w_s^{(1)}$ and the reward for $(s, a_-)$ as $-w_s^{(2)}$. Conversely, given a policy $\pi$ for $\mathcal{M}$, we can convert $\pi$ to a solution to sign-one-way marginals by outputting $v$ such that $v_s = +1$ if $\pi(s) = a_+$ and $v_s = -1$ otherwise.

It is now enough to observe that the value of the policy $\pi$ is exactly $\frac{1}{S} \sum_s p_s v_s$ while the value of the optimal policy (which chooses $a_+$ if $p_s$ is positive, and $a_-$ if $p_s$ is negative) is $\frac{1}{S} \sum_s |p_s|$. Thus given an $\varepsilon$-optimal policy $\pi$ we have: $\frac{1}{S} \sum_s p_s v_s = V(\pi, \mathcal{M}) \geq V(\pi^*, \mathcal{M}) - \varepsilon = \frac{1}{S} \sum_s |p_s| - \varepsilon$, or in other words $v$ is an $\varepsilon$-accurate solution to sign-one-way marginals, completing the reduction.[11]

To obtain the lower bound for the episodic setting, consider the same MDP as above with uniform starting distribution. Since an agent that consumes $m$ episodes observes each state (roughly) $\frac{m}{S}$ times, the $m$ episodic samples can be simulated with $\frac{m}{S}$ many parallel samples. A $\tilde{\Omega}(S^2 \rho^{-2}\varepsilon^{-2})$ lower bound in the episodic setting then follows from our parallel sample lower bound.

---

[10]Note that we can generalize to larger $A$.

[11]Technically, we defined MDPs to only have positive rewards and have used negative rewards here. This is just for exposition's sake – we can always shift the rewards from $r$ to $\frac{r+1}{2}$ to make them positive and run the same argument.

### 1.5. Related Work: Replicability in RL

Our work builds on (Eaton et al., 2023; Karbasi et al., 2023), who first studied replicable policy estimation in MDPs. In the parallel sampling setting, (Eaton et al., 2023; Karbasi et al., 2023) both consider the (discounted) infinite horizon setting in which the value of a policy is measured by interacting *indefinitely* with the MDP with rewards discounted (multiplicatively) in each step by a $\gamma$-discount factor. It is easy to reduce $\varepsilon$-optimal policy estimation for $\gamma$-discounted MDPs to the finite horizon case with $H = \tilde{O}(\log(1/\varepsilon)/(1 - \gamma))$, so our results also hold in their setting.

At a technical level, our basic (non-tiered) backward induction is similar to (Eaton et al., 2023)'s replicable phased value iteration, with the main difference being our use of multi-instance best-arm to optimize the sample complexity. In the parallel sampling setting, (Karbasi et al., 2023) also use the idea of 'bootstrapping' a non-replicable algorithm to a replicable one. They estimate the value of each state-action pair non-replicably and replicably round to the output vector. A priori, it is not at all clear how to perform such a procedure in the episodic setting. Our main contribution really lies in formalizing a new notion of exploration, sample collection, and (fractional) rounding that allows this type of bootstrapping, along with showing it can be achieved sample efficiently.

Finally, (Karbasi et al., 2023) also explore sample lower bounds for policy estimation. Conditional on a conjectured lower bound for replicably estimating the bias of $N$ coins,[12] they show an $\Omega(S^2)$ sample lower bound in the parallel setting against outputting a policy that is nearly optimal on every state (a stronger algorithmic guarantee than policy estimation). In contrast, our bound holds unconditionally against any algorithm that replicably outputs an $\varepsilon$-optimal policy.

### 1.6. Open Questions

The most obvious remaining question is to close the gap between our upper bound of $\tilde{O}(S^2 A)$ (Theorem 1.3) and unconditional lower bound of $\tilde{\Omega}(S^2)$ (Theorem 1.5). The gap in sample complexity arises from the fact that our algorithm in the (non-parallel) generative model setting samples every arm equally, while our lower bound based on sign-one-way marginals constructs an MDP in which the agent can readily discard all but 2 actions for each state. Can we design more sample efficient algorithms by adaptively choosing the sample size of each arm? Or can sign-one-way marginals (or another relevant task) be generalized beyond the Boolean domain to induce a dependency on $A$ in the lower bound?

Finally, while not a focus of this work, improving the $H$ de-

---

[12](Hopkins et al., 2024) made progress on this conjecture, but does not achieve the full strength required by (Karbasi et al., 2023).

pendency remains an interesting and challenging direction. Even in the non-replicable case, settling the dependence on $H$ is a comparatively recent development (Zhang et al., 2024) (see, e.g., (Zhang et al., 2024) for a discussion of improving $H$ dependency in the non-replicable case).

## Conflict of Interest Disclosure

The authors are not aware of any financial conflicts of interest. Funding sources are acknowledged below.

## Acknowledgments

We thank Russell Impagliazzo for helpful discussion on exploration in MDPs in the early stages of this work, and anonymous reviewers for pointing out the connection between our techniques and reward-free RL. MH is supported by NSF Award DMS-2424441. SL is supported by NSF Medium Award CCF-2107547 and NSF Award CCF-1553288 (CAREER). Part of this work was done when SL was visiting the National Institute of Informatics (NII). CY is supported by NSF Award AF: Medium 2212136 and HDR TRIPODS Phase II grant 2217058. YY is supported by JSPS KAKENHI Grant Number JP24K02903.

## Impact Statement

The contribution of this work is to advance theoretical machine learning. Therefore, we believe that it does not have societal impacts that we feel must be highlighted here.

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

# A. Further Related Work

**Replicability in Learning** Replicability was introduced in (Impagliazzo et al., 2022; Ghazi et al., 2021), building on earlier notions of stability (Bousquet & Elisseeff, 2002) and differential privacy (Dwork et al., 2006; Bun et al., 2020). Efficient replicable algorithms are known many statistical tasks including, e.g., statistical queries (Impagliazzo et al., 2022), bandits (Esfandiari et al., 2023; Komiyama et al., 2024; Ahmadi et al., 2026; Zhang et al., 2025), hypothesis testing (Hopkins et al., 2024; Liu & Ye, 2024), PAC learning (Bun et al., 2023; Hopkins & Moran, 2025), reinforcement learning (Eaton et al., 2023; Karbasi et al., 2023), and more (Esfandiari et al., 2024; Kalavasis et al., 2024). Besides (Eaton et al., 2023; Karbasi et al., 2023), our work is most related to the bandits setting (essentially the case of $S = 1$). These works consider a stronger notion of replicability requiring the algorithm to play the same *sequence* of actions. This does not apply in the general MDP setting, where the main difficulty is that the agent may be sent to (or start at) completely different states even upon taking the same action.

**Replicability and Differential Privacy** Another line of work has focused on relating replicability to other established notions of stability in the literature, most notably differential privacy (Bun et al., 2023; Kalavasis et al., 2023; Moran et al., 2023; Chase et al., 2023). For batch learning problems, (Ghazi et al., 2021; Bun et al., 2023) show it is generally possible to 'losslessly' convert a replicable algorithm to a private one, and convert a private algorithm to a replicable one at the cost of quadratic blowup in sample complexity. In the MDP setting, it is likely (though not entirely clear) that a version of the private-to-replicable transform still holds. In this case, since the best known private RL algorithms use $O(S^2AH^3)$ samples (Vietri et al., 2020; Qiao & Wang, 2023), such a transform would result in a replicable algorithm on $O(S^4A^2H^6)$ samples, an improvement over (Eaton et al., 2023) but still far from our desired near-optimal dependence.

In the other direction, applying (Bun et al., 2023)'s replicable-to-private transform to our algorithm indeed results in an $(\alpha, \beta)$-approximately differential private PAC policy estimator on $O(S^2A\frac{\log(1/\beta)}{\alpha})$-samples (in fact, one achieves the stronger notion of *user-level* privacy (Ghazi et al., 2021)). This is incomparable to prior work, which achieves (pure) *joint* differential privacy with a similar number of samples.[13] It would be interesting to see whether our techniques could be applied directly to achieve sub-quadratic or even linear dependence on the state space $S$ in either of these settings.

# B. Preliminaries

**Markov Decision Process** Let $\mathcal{M}$ be a Markov Decision Process (MDP) with state space $\mathcal{S}$, action set $\mathcal{A}$, and $H$ time steps. Given a collection of subsets of states $\{\mathcal{S}_h\}_{h=1}^H$ indexed by the time step, we call them a *state combination*. We denote by $\mathbf{p}_0$ the initial distribution over starting states, $\mathbf{p}_h(s, a) \in \Delta(\mathcal{S})$ the transition distribution after taking action $a$ at state $s$ in the $h$-th time step, and $\mathbf{r}_h(s, a) \in \Delta([0, 1])$ the distribution of the reward received after taking action $a$ at state $s$ at the $h$-th time step. Unless otherwise specified, we assume that the initial distribution $\mathbf{p}_0$ is supported on a fixed state $x_{\mathrm{ini}}$. We remark that this is without loss of generality as a more general MDP with randomized starting state can be simulated by a "step 0" where the transitions of all state-action pairs are set to be the desired initial state distribution.

Sometimes we view $\mathbf{p}_h(s, a)$ as an $S$-dimensional vector encoding the corresponding transition probabilities. Under this interpretation, we define $\mathbf{p}_H(s, a)$ to be the $\mathbf{0}$ vector for all $(s, a) \in \mathcal{S} \times \mathcal{A}$ since the process terminates after the $H$-th step. Similarly, we define $\mathbf{r}_{H+1}(s, a)$ to be deterministically 0 for all $(s, a)$.

A policy $\pi$ is a collection of functions $\{\pi_h\}_{h=1}^H$, where we have $\pi_h : \mathcal{S} \to \mathcal{A}$. Given a policy $\pi$ on $\mathcal{M}$, the value function of the policy, denoted $V(\pi, \mathcal{M})$ is defined as the cumulative expected reward obtained by following policy $\pi$ on $\mathcal{M}$. A policy $\pi$ is $\varepsilon$-optimal if its value is at most $\varepsilon$ less than the maximum value of any policy, i.e. $V(\pi, \mathcal{M}) \geq \max_{\pi'} V(\pi', \mathcal{M}) - \varepsilon$.

We write $X_h^\pi$ for the distribution of the state the agent is in on the $h$-th time step following policy $\pi$ on the $\mathcal{M}$. Given a set of states $S$, we define $\Pr[x_h \in S \mid \pi] := \Pr_{x \sim X_h^\pi}[x \in S]$.

In this work, we consider two models of sample access to the $\mathcal{M}$. The first is the simpler parallel sampling model, where the algorithm has sample access to an oracle $\mathbf{PS}(G_{\mathcal{M}})$ which simultaneously provides a sample from all transition and reward distributions.

**Definition B.1** (Parallel Sampling). A single call to $\mathbf{PS}(G_{\mathcal{M}})$ will return, for every tuple $(s, a, h) \in \mathcal{S} \times \mathcal{A} \times [H]$, an i.i.d.

---

[13]In essence, joint DP promises privacy even against other users in the training data set. The notion of privacy arising from a replicable-to-DP transform is weaker in that it only protects against outside users after the policy is released, but stronger in that it protects against swapping out larger 'batches' of user training data.

sample $(s', r)$, where $s' \sim \mathbf{p}_h(s, a)$ and $r \sim \mathbf{r}_h(s, a)$.

In the second episodic setup, the algorithm must interact with the MDP $\mathcal{M}$ in episodes, only observing state-action pairs corresponding to its trajectory in a given episode. To facilitate the discussion of replicability in this setup, we formulate the episodic process in an equivalent but slightly non-standard way where a parallel sample is drawn in each step but the agent can only read the sample in its current location.

**Definition B.2** (Episodic Sample Access). Let $T := \{(t_{s,a,h}, r_{s,a,h})\}_{s \in \mathcal{S}, a \in \mathcal{A}, h \in [H]}$, where $t_{s,a,h} \sim \mathbf{p}_h(s, a), r_{s,a,h} \sim \mathbf{r}_h(s, a)$, be a parallel sample from $\mathbf{PS}(G_\mathcal{M})$. We say a reinforcement learning agent (RL Agent) has episodic access to $T$ if it reads from $T$ in an interactive way as follows:

1. The agent starts at a fixed initial state $s_1 := x_{\mathrm{ini}}$.

2. For each $h \in [H]$, the agent chooses action $a$, and observes the data $t_{s_h, a, h}$.

3. After the process terminates, the parallel sample $T$ is deleted.

Let $\mathcal{L}$ be an episodic PAC policy estimator, and $\mathcal{T}$ be a batch if i.i.d. samples from $\mathbf{PS}(G_\mathcal{M})$. We write $\mathcal{L}(\mathcal{T}; \xi)$ to denote the output of the estimator when it is given episodic access to the samples in $\mathcal{T}$ and uses the string $\xi$ as its internal random seed.

Our goal is to obtain an algorithm that replicably finds near-optimal policies with episodic sample access.

**Definition B.3.** Let $C$ be the class of MDPs with $S$ states, $A$ actions, and $H$ time steps. An algorithm is an $(\varepsilon, \delta)$-PAC policy estimator with sample complexity $m \cdot H$ if for all fixed MDPs $\mathcal{M}$, given $m$ episodic samples, the algorithm with probability at least $1 - \delta$ outputs an $\varepsilon$-optimal policy $\pi$.

**Replicability** We recall the definition of replicability in the standard batch setting.

**Definition B.4.** A randomized algorithm $\mathcal{L}$ is $\rho$-replicable if $\mathrm{Pr}_{\xi, T, T'}(\mathcal{L}(T; \xi) = \mathcal{L}(T'; \xi)) \geq 1 - \rho$, for all distributions $\mathbf{p}$, where $T, T'$ are i.i.d. samples taken from $\mathbf{p}$ and $\xi$ denotes the internal randomness of the algorithm $\mathcal{L}$.

A replicable PAC Policy Estimator is defined analogously as follows.

**Definition B.5** (Formal Version of Replicable PAC Policy Estimation). An episodic (resp. parallel) $(\varepsilon, \delta)$-PAC policy estimator $\mathcal{L}$ is called $\rho$-replicable if for every MDP $\mathcal{M}$, $\mathrm{Pr}_{\xi, \mathcal{T}, \mathcal{T}'}(\mathcal{L}(\mathcal{T}; \xi) = \mathcal{L}(\mathcal{T}'; \xi)) \geq 1 - \rho$, where $\mathcal{T}, \mathcal{T}'$ consist of independent i.i.d. parallel samples from $\mathbf{PS}(G_\mathcal{M})$ and $\xi$ denotes the internal randomness of the algorithm $\mathcal{L}$.

**Distribution Divergence and Distance** In the following, let $X, Y$ be random variables. Unless otherwise specified, the random variables are over domain $\mathcal{X}$.

**Definition B.6.** The *total variation distance* between $X$ and $Y$ is

$$d_{\mathrm{TV}}(X, Y) = \frac{1}{2} \sum_{x \in \mathcal{X}} |\mathrm{Pr}(X = x) - \mathrm{Pr}(Y = x)|.$$

**Definition B.7.** The *KL-divergence* of $X$ from $Y$ is

$$\mathrm{KL}(X||Y) = \sum_{x \in \mathcal{X}} \mathrm{Pr}(X = x) \log \frac{\mathrm{Pr}(X = x)}{\mathrm{Pr}(Y = x)}.$$

We mainly use KL divergence due to the following useful tensorization lemma.

**Lemma B.8.** *Let $X_1, \ldots, X_n, Y_1, \ldots, Y_n$ be independent random variables over $\mathcal{X}$. Define the product random variables over $\mathcal{X}^n$ as $X = (X_1, \ldots, X_n)$ and $Y = (Y_1, \ldots, Y_n)$. Then,*

$$\mathrm{KL}(X||Y) = \sum_{i=1}^{n} \mathrm{KL}(X_i||Y_i).$$

**Definition B.9.** Let $X, Y$ be random variables on $\mathcal{X}$. The $\chi^2$-*divergence* of $X$ from $Y$ is

$$\chi^2(X||Y) = \sum_{x \in \mathrm{supp}(Y)} \frac{(\mathrm{Pr}(X = x) - \mathrm{Pr}(Y = x))^2}{\mathrm{Pr}(Y = x)}.$$

Finally, the following relations between deviation measures is well known.

**Lemma B.10** (Divergence Relationships). *Let $X, Y$ be random variables on $\mathcal{X}$.*

$$2d_{\mathrm{TV}}\left(X, Y\right)^2 \leq \mathrm{KL}(X||Y) \leq \chi^2(X||Y).$$

For Bernoulli product distributions, we have the following total variation distance bound.

**Lemma B.11** (TV distance between Bernoulli Product Distributions). *Let $\mathcal{B}_1, \mathcal{B}_2$ be two Bernoulli product distributions with mean $\mu^{(1)}, \mu^{(2)} \in [0,1]^n$ respectively. It holds that*

$$d_{\mathrm{TV}}^2\left(\mathcal{B}_1, \mathcal{B}_2\right) \leq \sum_{i:\mu_i^{(1)}>0}^{n} \left(\mu_i^{(1)} - \mu_i^{(2)}\right)^2 / \mu_i^{(1)} + \sum_{i:\mu_i^{(1)}<1}^{n} \left(\mu_i^{(1)} - \mu_i^{(2)}\right)^2 / \left(1 - \mu_i^{(1)}\right).$$

*Proof.* Denote by $\mathbf{Ber}\left(\mu_i^{(j)}\right)$ the Bernoulli distribution with mean $\mu_i^{(j)}$. By Lemma B.8 and Lemma B.10, we have that

$$\mathrm{KL}\left(\mathcal{B}_1, \mathcal{B}_2\right) \leq \sum_i \mathrm{KL}\left(\mathbf{Ber}\left(\mu_i^{(1)}\right), \mathbf{Ber}\left(\mu_i^{(2)}\right)\right) \leq \sum_i \chi^2\left(\mathbf{Ber}\left(\mu_i^{(1)}\right), \mathbf{Ber}\left(\mu_i^{(2)}\right)\right)$$

$$= \sum_{i:\mu_i^{(1)}>0} \left(\mu_i^{(1)} - \mu_i^{(2)}\right)^2 / \mu_i^{(1)} + \sum_{i:\mu_i^{(1)}<1} \left(\left(\mu^{(1)}\right)_i - \mu_i^{(2)}\right)^2 / \left(1 - \mu_i^{(1)}\right),$$

where the last equality follows from the definition of $\chi^2$ divergence. Since $d_{\mathrm{TV}}^2\left(\mathcal{B}_1, \mathcal{B}_2\right)$ is at most $\mathrm{KL}\left(\mathcal{B}_1, \mathcal{B}_2\right)$ by Lemma B.10, the lemma hence follows. $\square$

**Correlated Sampling**  Given two similar distributions, we require an algorithm to sample in a consistent way. This is achieved with correlated sampling (Broder, 1997; Kleinberg & Tardos, 2002; Holenstein, 2007).

**Lemma B.12** (Correlated Sampling, Lemma 7.5 of (Rao & Yehudayoff, 2020)). *Let $\mathcal{X}$ be a finite domain. There is a randomized algorithm* **CorrSamp** *(with internal randomness $\xi \sim \mathbf{p}_R$) such that given any distribution over $\mathcal{X}$ outputs a random variable over $\mathcal{X}$ satisfying the following:*

1. *(Marginal Correctness) For all distributions $\mathbf{p}$ over $\mathcal{X}$,*

$$\Pr_{\xi \sim \mathbf{p}_R}\left(\mathbf{CorrSamp}(\mathbf{p}, \xi) = x\right) = \Pr_{X \sim \mathbf{p}}\left(X = x\right).$$

2. *(Error Guarantee) For all distributions $\mathbf{p}, \mathbf{p}'$ over $\mathcal{X}$,*

$$\Pr_{\xi \sim \mathbf{p}_R}\left(\mathbf{CorrSamp}(\mathbf{p}, \xi) \neq \mathbf{CorrSamp}(\mathbf{p}', \xi)\right) \leq 2 \cdot d_{\mathrm{TV}}\left(\mathbf{p}, \mathbf{p}'\right).$$

*Furthermore, the algorithm runs in expected time $\tilde{O}(|\mathcal{X}|)$.*

In many applications, we will specifically sample from product distributions. Unfortunately, since known algorithms for correlated sampling algorithms have complexity scaling with the size of the domain, the algorithm can become extremely inefficient on product distributions as the domain size grows exponentially. In the following, we give a procedure to sample from product distributions in a computationally efficient manner (with some extra loss in the error guarantee). The algorithm is given the description to a product distribution $\mathbf{p} = (\mathbf{p}_1, \ldots, \mathbf{p}_n)$ where each marginal $\mathbf{p}_i$ is specified by the probability mass of each point in the domain.

**Lemma B.13** (Computationally Efficient Correlated Sampling for Product Distributions). *Let $\mathcal{X}$ be a finite domain. There is a randomized algorithm* **ProdCorrSamp** *(with internal randomness $\xi \sim \mathbf{p}_R$) such that given the description of a product distribution $\mathbf{p} = (\mathbf{p}_1, \ldots, \mathbf{p}_n)$ over $\mathcal{X}^n$ and sample access to $\mathbf{p}_R$ outputs a random variable over $\mathcal{X}^n$ satisfying the following:*

1. *(Marginal Correctness) For all product distributions* $\mathbf{p}$ *over* $\mathcal{X}^n$,

$$\Pr_{\xi \sim \mathbf{p}_R} \left( \textbf{PRODCORRSAMP}(\mathbf{p}, \xi) = x \right) = \Pr_{X \sim \mathbf{p}} \left( X = x \right).$$

2. *(Error Guarantee) For all product distributions* $\mathbf{p}, \mathbf{p}'$ *over* $\mathcal{X}^n$,

$$\Pr_{\xi \sim \mathbf{p}_R} \left( \textbf{PRODCORRSAMP}(\mathbf{p}, \xi) \neq \textbf{PRODCORRSAMP}(\mathbf{p}', \xi) \right) \leq 2 \sum_{i=1}^{n} d_{\mathrm{TV}} \left( \mathbf{p}_i, \mathbf{p}'_i \right).$$

*Furthermore, the algorithm runs in expected time* $\tilde{O}(n|\mathcal{X}|)$.

*Proof.* The algorithm outputs $(\textbf{CORRSAMP}(\mathbf{p}_1, \xi_1), \dots, \textbf{CORRSAMP}(\mathbf{p}_n, \xi_n))$ for $n$ independent runs of the **CORRSAMP** algorithm (Lemma B.12). The marginal correctness follows from the marginal correctness of **CORRSAMP** and that $\mathbf{p}$ is a product distribution. We obtain the guarantee on the error bound by a union bound. $\qed$

**Randomized Rounding** Our algorithm requires a subroutine that performs replicable mean estimation for high-dimensional distribution. A particularly convenient way to do so is through randomized rounding.

**Lemma B.14** (Randomized Rounding). *There exists a randomized algorithm* **RANDROUND** *that takes as input a vector* $\mathbf{x} \in [0, 1]^n$ *and an accuracy parameter* $\varepsilon \in (0, 1)$, *produces some rounded vector* $\mathbf{y} \in \mathbb{R}^n$, *and satisfies the following guarantees: (1)* $\|\mathbf{x} - \mathbf{y}\|_\infty \leq \varepsilon$ *almost surely (2) Let* $\mathbf{x}^{(1)}, \mathbf{x}^{(2)}$ *be two different inputs satisfying* $\|\mathbf{x}^{(1)} - \mathbf{x}^{(2)}\|_2 \ll \varepsilon \rho / \log(n\rho^{-1})$ *for some number* $\rho \in (0, 1)$. *Then,* $\textbf{RANDROUND} \left( \mathbf{x}^{(1)}, \varepsilon; \xi \right) = \textbf{RANDROUND} \left( \mathbf{x}^{(2)}, \varepsilon; \xi \right)$ *with probability at least* $1 - \rho$, *where the randomness is over the shared internal random string* $\xi$ *used by the algorithm. Moreover, the runtime of the algorithm is* $\exp(O(n))$.

The above result is implicit in the proof of the replicable $\ell_\infty$ mean estimator of (Hopkins et al., 2024). First, Proposition 4.10 of (Hopkins et al., 2024) gives an algorithm with almost identical guarantees except that the rounding error (property (1)) is $\|\mathbf{x} - \mathbf{y}\|_2 \ll \varepsilon \sqrt{n} / \log(n\rho^{-1})$. By examining the proof of the $\ell_\infty$-norm mean estimator in Theorem 4.9 of (Hopkins et al., 2024), we observe that **RANDROUND** has a randomized rotation procedure that ensures that $\|x - y\|_\infty \leq \|x - y\|_2 \log(n\rho^{-1}) / \sqrt{n} \ll \varepsilon$, i.e., the error vector $x - y$ points towards a uniformly random direction and therefore has fairly uniform entries.

## C. Replicable Best Arm

We start with the simpler problem of the best arm identification.

**Definition C.1** ($(A, \varepsilon)$-Best Arm Problem). We are given $A$ unknown arms. Upon pulling the $i$-th arm, one receives a random utility following the distribution of $\mathbf{r}_i$ supported on $[0, 1]$. Let $\mathsf{OPT} = \max_i \mathbb{E}[\mathbf{r}_i]$. The $i$-th arm is $\varepsilon$-optimal for $\mathbf{r}$ if

$$\mathbb{E}[\mathbf{r}_i] \geq \mathsf{OPT} - \varepsilon.$$

**Definition C.2.** An algorithm solves the $(A, \varepsilon)$-Best Arm problem with probability $1 - \delta$ if given sample access to an instance parametrized by $\mathbf{r}$, the algorithm returns an $\varepsilon$-optimal arm for $\mathbf{r}$ with probability at least $1 - \delta$.

A naive strategy is to replicably estimate the expected rewards of each arm, and simply pick the one with the highest estimate. The sample complexity of the state-of-the-art replicable mean estimator scales quadratically with the dimension of the problem. One would then get an algorithm with roughly $A^2$ (assuming that the other relevant parameters are constant) arm queries. Inspired by (Ghazi et al., 2021)'s sample-efficient replicable PAC learning algorithm (which similarly tries to find a single $\varepsilon$-optimal hypothesis in the supervised learning setting), we argue there is an efficient replicable algorithm for best-arm identification using only linearly many queries (up to polylogarithmic factors).

**Theorem C.3.** *Let* $\delta \leq \rho \leq \frac{1}{2}$. *There exists an efficient* $\rho$-*replicable algorithm that solves the* $(A, \varepsilon)$-*best arm problem with probability at least* $1 - \delta$, *and uses* $O\left( \frac{A}{\rho^2 \varepsilon^2} \left( \log \frac{A}{\delta} \right)^3 \right)$ *arm queries.*

Like (Ghazi et al., 2021), our algorithm leverages the exponential mechanism and correlated sampling to ensure replicability, a technique also considered in (Hara & Yoshida, 2023). In particular, we draw roughly $\rho^{-2}\varepsilon^{-2}$ many samples per arm to

compute a (non-replicable) estimate of its expected utility. We then sample an arm according to the exponential mechanism with the utility function defined according to the estimates. It can be shown that the distributions induced by the exponential mechanism in two separate runs of the algorithm will be close in TV distance, and one could then leverage correlated sampling to ensure replicability.

*Proof.* Set $t = \frac{\log(2A/\delta)}{\varepsilon}$. For $a \in [A]$, let $r_a$ be its expected utility. We use the following algorithm (denoted $\mathcal{L}$):

1. For each $a \in [A]$, draw $m = \frac{\Theta(1)}{\rho^2 \varepsilon^2} \left( \log \frac{2A}{\delta} \right)^3$ samples from arm $a$.

2. Let $\hat{r}_a$ denote the empirical reward received from the $a$-th arm.

3. Define the distribution $\hat{\mathbf{p}}$ on $[A]$ according to the exponential mechanism, i.e., $a \in [A]$ is drawn with probability proportional to $\exp(t\hat{r}_a)$.

4. Using shared internal randomness $r$, output $\textbf{CORRSAMP}(\hat{\mathbf{p}}, r)$ (Lemma B.12).

We analyze the correctness and replicability of our algorithm. We condition on the event

$$|r_a - \hat{r}_a| \leq \frac{\rho \varepsilon}{10 \log(2A/\delta)}$$

for all $a$, which happens with probability at least $1 - \delta$ by the Chernoff bound and the union bound. We note that to establish correctness alone, it suffices to obtain the concentration bound $|r_a - \hat{r}_a| \leq \rho \varepsilon$. The stronger concentration bound is required due to replicability.

We begin by arguing for the algorithm's correctness. Denote by $a^* \in [A]$ the index of an optimal arm, i.e., $a^* = \operatorname{argmax}_a r_a$. Then, the probability that an arm $a \in [A]$ is sampled is at most $\exp(t(\hat{r}_a - \hat{r}_{a^*}))$. If $r_a < r_{a^*} - \varepsilon$ then by the above conditioning, $\hat{r}_a \leq r_a + \frac{\varepsilon}{4} < r_{a^*} - \varepsilon + \frac{\varepsilon}{4} \leq \hat{r}_{a^*} - \varepsilon + \frac{\varepsilon}{2}$. Thus, $\hat{r}_a \leq \hat{r}_{a^*} - \frac{\varepsilon}{2}$. Finally the probability any such $a$ is output is at most

$$\exp\left(t\varepsilon/2\right) \leq \frac{\delta}{2A}.$$

so that by a union bound, the probability that any arm with $r_a \leq r_{a^*} - \varepsilon$ is chosen is at most $\frac{\delta}{2}$. Combined with the condition event, the probability of an error is at most $\delta$.

Next, we analyze replicability. Let $S_1, S_2$ denote the samples observed by two independent runs of the algorithm $\mathcal{L}$ and $r$ the shared internal randomness. Let $\hat{r}_a^{(i)}$ denote the empirical utility observed from $S_i$. As above, we condition on the event $|\hat{r}_a^{(i)} - r_a| \leq \frac{\rho \varepsilon}{10 \log(2A/\delta)}$ for all $a \in [A], i \in [2]$. Note that this condition fails with probability at most $\frac{\delta}{2}$. Let $P(a) = \Pr(\mathcal{L}(S_1; r) = a)$ and $Q(a) = \Pr(\mathcal{L}(S_2; r) = a)$. Then by the definition of the exponential mechanism

$$P(a) = \frac{\exp(t\hat{r}_a^{(1)})}{\sum_{a'} \exp(t\hat{r}_{a'}^{(1)})}$$

$$Q(a) = \frac{\exp(t\hat{r}_a^{(2)})}{\sum_{a'} \exp(t\hat{r}_{a'}^{(2)})}.$$

We bound the total variation distance between $P$ and $Q$. Fix some $a \in A$. Then,

$$P(a) - Q(a) = \frac{\exp(t\hat{r}_a^{(1)})}{\sum_{a'} \exp(t\hat{r}_{a'}^{(1)})} - \frac{\exp(t\hat{r}_a^{(2)})}{\sum_{a'} \exp(t\hat{r}_{a'}^{(2)})}$$

$$= \left( \frac{\exp(t\hat{r}_a^{(2)})}{\sum_{a'} \exp(t\hat{r}_{a'}^{(2)})} \left( \frac{\exp(t\hat{r}_a^{(1)}) \sum_{a'} \exp(t\hat{r}_{a'}^{(2)})}{\exp(t\hat{r}_a^{(2)}) \sum_{a'} \exp(t\hat{r}_{a'}^{(1)})} - 1 \right) \right).$$

Since we have conditioned on $|\hat{r}_a^{(i)} - r_a| \leq \frac{\rho \varepsilon}{10 \log(2A/\delta)}$, the triangle inequality implies $|\hat{r}_a^{(1)} - \hat{r}_a^{(2)}| \leq \frac{\rho \varepsilon}{5 \log(2A/\delta)}$. We then apply

$$\exp(t\hat{r}_a^{(2)}) \leq \exp\left( t \left( \hat{r}_a^{(1)} + \frac{\rho \varepsilon}{5 \log(2A/\delta)} \right) \right)$$

to each $a \in [A]$ in the summation and

$$\exp(t\hat{r}_a^{(1)}) \leq \exp\left(t\left(\hat{r}_a^{(2)} + \frac{\rho\varepsilon}{5\log(2A/\delta)}\right)\right)$$

to obtain

$$\begin{aligned}
|P(a) - Q(a)| &\leq \left(\frac{\exp(t\hat{r}_a^{(2)})}{\sum_{a'}\exp(t\hat{r}_{a'}^{(2)})}\left(\exp\left(t\frac{2\rho\varepsilon}{5\log(2A/\delta)}\right) - 1\right)\right) \\
&= \left(\frac{\exp(t\hat{r}_a^{(2)})}{\sum_{a'}\exp(t\hat{r}_{a'}^{(2)})}\left(\exp\left(\frac{2\rho}{5}\right) - 1\right)\right) \\
&\leq \frac{4}{5}Q(a)\rho.
\end{aligned}$$

where in the last inequality we use $e^x \leq 1 + 2x$ for $x \leq 1$. In particular,

$$d_{\mathrm{TV}}(P, Q) = \sum_a |P(a) - Q(a)| \leq \frac{4}{5}\rho.$$

Thus, applying correlated sampling (Lemma B.12) and the union bound over our conditioned events, the two runs of the algorithm output different arms with probability at most $2\rho + \delta \leq 3\rho$. Note that a $\rho$-replicable algorithm can be obtained by sufficiently reducing the replicability parameter by a constant (with only a constant blow-up in sample complexity). $\qquad\square$

## C.1. Multiple Instances of Best Arm

In reinforcement learning, we need to simultaneously solve $S$ instances of best arm (one for each state), as well as estimate the expected rewards of the resulting selected arms to apply backwards induction. On top of this, several technical complications arise in the episodic setting where we cannot fully control the number of samples received for each instance due to random variation in the exploration process. Toward this end, let $m_s$ the number of samples received from instance $s$. We will show it is still possible to achieve replicable best-arm only under the assumption that

$$\sqrt{\sum_{s=1}^{S}\frac{1}{m_s}} \ll \rho\varepsilon \cdot \mathrm{polylog}^{-1}(SA/\rho)$$

where $S$ is the number of instances, $A$ is the number of arms per instance, $\rho$ is the replicability parameter, and $\varepsilon$ is the accuracy parameter. In fact, we will show this even when $m_s$ itself a random variable provided there exists a sufficiently good (unknown) high probability lower bound on $m_s$. We formalize the problem instance below that incorporates these extra technical complications.

**Definition C.4.** Suppose we have $S$ instances of the $(A, \varepsilon)$-best arm problem parameterized by $\{\mathbf{r}_s\}_{s=1}^S$. A solution $\{(a_s, \bar{r}_s)\}_{s=1}^S$ is $\varepsilon$-optimal if (1) $a_s$ is $\varepsilon$-optimal for $\mathbf{r}_s$ and (2) $\bar{r}_s$ is within $\varepsilon$ of the expected utility of arm $a_s$ for all $s$.

**Definition C.5.** An algorithm solves the $(S, A, \varepsilon)$-Multi-Instance Best Arm Problem if given datasets $\{\mathcal{D}_{s,a}\}$ consisting of i.i.d. samples from the $a$-th arm of the $s$-th state for all $s, a$, it simultaneously finds (1) some $\varepsilon$-optimal arm $a_s$ for each instance $s$, and (2) an estimate $\bar{r} \in \mathbb{R}^S$ such that $\bar{r}_s$ is within $\varepsilon$ of the expected utility of the arm $a_s$ in instance $s$.

**Proposition C.6** (Replicable Multi-Instance Best Arm Algorithm)**.** *Let $\delta \leq \rho \leq \frac{1}{3}$ and $\varepsilon \in (0, 1)$. Let $\{m_s\}_{s=1}^S$ be a sequence of numbers satisfying $\sum_{s=1}^{S} 1/m_s \ll \frac{\rho^2\varepsilon^2}{\log^3(SA/\delta)}$. There is a $\rho$-replicable algorithm REPVARBANDIT that given $\{\mathcal{D}_{s,a}\}$ where $\mathcal{D}_{s,a}$ are datasets of random size satisfying $|\mathcal{D}_{s,a}| \geq m_s$ almost surely for all $s, a$, solves the $(S, A, \varepsilon)$-multi-instance best arm problem, and runs in time $\mathrm{poly}\left(\sum_{s,a}|\mathcal{D}_{s,a}|\right) + \exp\left((1 + o(1))S\log A\right)$.*

The computationally efficient version of the algorithm follows by running $S$ best-arm algorithms in parallel, and then use a replicable statistic to estimate the expected utility of the selected arm. To get the sample efficient version, instead of running correlated sampling on the exponential mechanism distributions of each problem instance individually, we directly sample from their joint distribution supported on the space $[A]^S$. After that, we use the technique of high-dimensional randomized rounding to construct a replicable mean estimation of the expected utilities of the selected arms. This turns out to save a factor of $S$ in sample complexity at the cost of increasing the runtime to $\tilde{O}\left(A^S\right)$.

*Proof of Proposition C.6.* The algorithm is given in Algorithm 1. We denote by $\mathbf{r}_{s,a}$ the random variable describing the distribution of the utility of the $a$-th arm of the $s$-th instance, and define $r_{s,a} = \mathbb{E}\left[\mathbf{r}_{s,a}\right]$.

---

**Algorithm 1** REPVARBANDIT

---

**Input**: For each $s \in [S], a \in [A]$, a dataset $\mathcal{D}_{s,a}$, made up of i.i.d. samples from the distribution of $\mathbf{r}_{s,a}$, accuracy parameter $\varepsilon > 0$, and failure probability $\delta > 0$.

**Output**: For each $s \in [S]$, a selected arm $a_s \in [A]$ and an estimate $\bar{r}_s \in (0, 1)$.

1. Let $\hat{r}_{s,a}$ denote the empirical reward computed from $\mathcal{D}_{s,a}$.

2. Define the distribution $\hat{\mathbf{p}}_s$ on $[A]$ according to the exponential mechanism: $a$ is sampled with probability proportional to $\exp(t\hat{r}_{s,a})$, where $t = 2\log(3SA/\delta)/\varepsilon$.

3. Define $\hat{\mathbf{p}}$ on $[A]^S$ as the product distribution $(\hat{\mathbf{p}}_1, \ldots, \hat{\mathbf{p}}_S)$.

4. Using shared internal randomness $\xi$, sample $\{a_s\}_s \sim$ CORRSAMP$(\hat{\mathbf{p}}; \xi)$ (Lemma B.12) .

5. Construct the vector $\tilde{r} \in [0, 1]^S$, where $\tilde{r}_s = \hat{r}_{s,a_s}$.

6. Using shared internal randomness $\xi$, compute $\bar{r} \leftarrow$ RANDROUND$(\tilde{r}, \varepsilon/2; \xi)$ (Lemma B.14).

7. Output $a_s$ and $\bar{r}_s$ for each $s$.

---

We first show some preliminary claims. Recall that $\{m_s\}_{s\in[S]}$ are numbers satisfying

$$\sum_s \frac{1}{m_s} \le \frac{\rho^2 \varepsilon^2}{C(\log(3SA/\delta))^3}, \tag{4}$$

for some sufficiently large constant $C$, and $\bar{r}_{s,a}$ is the empirical reward estimate computed from the dataset $\mathcal{D}_{s,a}$ satisfying $|\mathcal{D}_{s,a}| \ge m_s$. Observe that under Equation (4), we must also have

$$\max_s \sqrt{\frac{\log(3SA/\delta)}{m_s}} \le \sqrt{\sum_s \frac{\log(3SA/\delta)}{m_s}} \le \sqrt{\log(3SA/\delta)}\sqrt{\sum_s \frac{1}{m_s}} \ll \frac{\varepsilon\rho}{2} \tag{5}$$

where the last inequality follows from Equation (4). Besides, by the Chernoff bound and the union bound, we have

$$|\hat{r}_{s,a} - r_{s,a}| \le \sqrt{\frac{\log(3SA/\delta)}{m_s}} \tag{6}$$

for all $s, a$ with probability at least $1 - \delta/3$. We will condition on this event in the remaining analysis. The proof consists of three parts. In the first part, we argue the correctness of the arms selected. In the second part, we argue the arms selected are replicable. Lastly, we show the the estimates $\bar{r}_s$ are accurate and replicable.

**Arm Selection Correctness** Following the proof of Theorem C.3, for each $s$, we proceed to bound the probability that an arm with expected utility less than $\max_a r_{s,a} - \varepsilon$ is sampled following the exponential mechanism distribution. In particular, fix some sub-optimal arm $a$ such that

$$r_{s,a} \le \max_{a'} r_{s,a'} - \varepsilon. \tag{7}$$

For this specific sub-optimal arm $a$, we have that

$$\hat{r}_{s,a} \le r_{s,a} + \sqrt{\frac{\log(3SA/\delta)}{m_s}} \le \max_a r_{s,a} - \varepsilon/2 - \sqrt{\frac{\log(3SA/\delta)}{m_s}} \le \max_a \hat{r}_{s,a} - \varepsilon/2,$$

where the first inequality follows from Equation (6), the second inequality follows from our assumption Equation (7) and Equation (5), and the last inequality follows from Equation (5) and Equation (6). It follows that the exponential mechanism selects $a$ for instance $s$ with probability at most $\exp(t\varepsilon/2) < \frac{\delta}{3SA}$ since the algorithm sets $t = 2\log(3SA/\delta)/\varepsilon$. The correctness of the algorithm then follows from the union bound.

**Arm Selection Replicability** Suppose we run the algorithm on two sets of independently sampled datasets $\{\mathcal{D}_{s,a}^{(1)}\}, \{\mathcal{D}_{s,a}^{(2)}\}_{s\in[S], a\in[A]}$, satisfying $\min_i \mathcal{D}_{s,a}^{(i)} \geq m_s$ for all $s, a$. Following earlier notations, denote by $\hat{r}_{s,a}^{(i)}$ the empirical reward estimate computed from $\mathcal{D}_{s,a}^{(i)}$. Again, we condition on the event $|\hat{r}_{s,a}^{(i)} - r_{s,a}| \leq \sqrt{\frac{\log(3SA/\delta)}{m_s}}$ for all $i, s, a$. Denote by $\hat{\mathbf{p}}^{(i)}$ the distribution on $[A]^S$ induced by the exponential mechanism using the estimates $\{\hat{r}_{s,a}^{(i)}\}_{s,a}$. Moreover, denote by $\hat{\mathbf{p}}_{s,a}^{(i)}$ the probability that arm $a$ is sampled for instance $s$ in the $i$-th run.

It follows from Lemma B.10 and Lemma B.8 that

$$
\begin{aligned}
d_{\mathrm{TV}}\left(\hat{\mathbf{p}}^{(1)}, \hat{\mathbf{p}}^{(2)}\right)^2 &\leq \mathrm{KL}(\hat{\mathbf{p}}^{(1)} || \hat{\mathbf{p}}^{(2)}) \\
&= \sum_{s=1}^{S} \mathrm{KL}\left(\hat{\mathbf{p}}_s^{(1)} || \hat{\mathbf{p}}_s^{(2)}\right) \\
&\leq \sum_{s=1}^{S} \chi^2\left(\hat{\mathbf{p}}_s^{(1)} || \hat{\mathbf{p}}_s^{(2)}\right).
\end{aligned}
\tag{8}
$$

It remains to bound the $\chi^2$ divergence of a single instance. Using our concentration bound and the triangle inequality, we observe that $|\hat{r}_{s,a}^{(1)} - \hat{r}_{s,a}^{(2)}| \leq 2\sqrt{\frac{\log(3SA/\delta)}{m_s}}$. Following similar computations as Theorem C.3 we have that for all $s, a$,

$$
|\hat{\mathbf{p}}_{s,a}^{(1)} - \hat{\mathbf{p}}_{s,a}^{(2)}| \leq \hat{\mathbf{p}}_{s,a}^{(2)} \cdot 8t \cdot \sqrt{\frac{\log(3SA/\delta)}{m_s}}.
$$

As a result, we can bound the $\chi^2$ divergence by

$$
\begin{aligned}
\chi^2\left(\hat{\mathbf{p}}_s^{(1)} || \hat{\mathbf{p}}_s^{(2)}\right) &= \sum_a \frac{\left(\hat{\mathbf{p}}_s^{(1)}(a) - \hat{\mathbf{p}}_s^{(2)}(a)\right)^2}{\hat{\mathbf{p}}_s^{(2)}(a)} \\
&\leq \sum_a \hat{\mathbf{p}}_s^{(2)}(a)\left(8t\sqrt{\frac{\log(3SA/\delta)}{m_s}}\right)^2 \\
&\leq \frac{64t^2 \log(3SA/\delta)}{m_s} \\
&= \frac{256\left(\log(3SA/\delta)\right)^3}{\varepsilon^2 m_s}.
\end{aligned}
\tag{9}
$$

Combining Equations (8) and (9) then gives that

$$
d_{\mathrm{TV}}\left(\hat{\mathbf{p}}^{(1)}, \hat{\mathbf{p}}^{(2)}\right)^2 \leq \sum_{s=1}^{S} \frac{256\left(\log(3SA/\delta)\right)^3}{\varepsilon^2 m_s} \leq \frac{256\left(\log(3SA/\delta)\right)^3}{\varepsilon^2} \frac{\rho^2 \varepsilon^2}{C\left(\log(3SA/\delta)\right)^3} \leq \frac{\rho^2}{36},
$$

where the last inequality holds as long as $C$ is set to be a sufficiently large constant. Thus, if we apply correlated sampling with shared internal randomness, it follows from Lemma B.12 that the two runs of the algorithm will have identical outputs with probability at least $1 - \rho/3$.

**Replicability and Accuracy of $\bar{r}$** Recall that we define $\tilde{r} \in [0,1]^S$ to be such that $\tilde{r}_s = \hat{r}_{s,a_s}$, where $a_s \in [A]$ is the selected arm with instance $s$. Consider the vector $r^* \in [0,1]^S$ defined as $r_s^* = r_{s,a_s}$. It follows immediately from Equation (6) and Equation (5) that

$$
\|\tilde{r} - r^*\|_2 \ll \varepsilon\rho.
\tag{10}
$$

By Lemma B.14, RANDROUND with accuracy parameter $\varepsilon$ produces some vector $\bar{r}$ such that $\|\bar{r} - \tilde{r}\|_\infty \ll \varepsilon$. By the triangle inequality, it follows that $\|\bar{r} - r^*\|_\infty \leq \varepsilon$, thus obtaining accuracy. Let $\tilde{r}^{(1)}$ and $\tilde{r}^{(2)}$ be the estimates obtained in two runs of the algorithm. Conditioned on that the arms selected in the two runs are identical, the vector $r^*$ must also be identical in the two runs. It then follows from Equation (10) and the triangle inequality that $\|\bar{r}^{(1)} - \bar{r}^{(2)}\|_2 \ll \varepsilon\rho$. Therefore, running RANDROUND with accuracy $\varepsilon$ is $\rho$-replicable.

**Runtime Analysis**   We now discuss the computational efficiency of the algorithm. All sub-routines run in polynomial time algorithm except for the correlated sampling procedure and the randomized rounding procedure. The correlated sampling procedure runs in time $\exp\left((1 + o(1))S \log A\right)$ by Lemma B.12, and the randomized rounding procedure runs in time $\exp\left((1 + o(1))S\right)$ by Lemma B.14. Therefore, the total runtime is $\text{poly}\left(SA\varepsilon^{-1}\rho^{-1}\right) + \exp\left((1 + o(1))S \log A\right)$.   □

# D. Reinforcement Learning from Best-Arm Selection

In this section, we present our algorithm for reinforcement learning given an algorithm for replicable exploration satisfying certain nice properties. The exploration algorithm will be covered in Appendix E.

**Preliminaries: Boosting Replicability and Success Probability**   Throughout this section, we give policy estimators with constant replicability and success probabilities. Here, we describe a generic procedure to arbitrarily boost replicability and success probabilities. It is known that given a PAC learner that succeeds with probability at least $2/3$, one boost its success probability to $1 - \delta$ by blowing up the sample complexity by a factor of $\log(1/\delta)$. Moreover, given a $0.1$-replicable PAC learning, one can boost its replicability by blowing up the sample complexity by a factor of $\rho^{-2}$ (Impagliazzo et al., 2022; Hopkins et al., 2024). We show a similar boosting lemma holds for PAC policy estimators as well.

**Lemma D.1** (Boosting Replicability and Success Probability of RL). *Let $\delta < \rho < 1/2$. Suppose there is an $0.1$-replicable $(\varepsilon/2, 0.1)$-PAC policy estimator with sample complexity $m$ in the episodic (resp. parallel sampling) setting. Then, there is a $\rho$-replicable $(\varepsilon, \delta)$-PAC policy estimator with sample complexity $\tilde{O}\left(m \cdot \frac{\log^4(1/\delta)}{\rho^2} + \frac{H \cdot \log^4(1/\delta)}{\varepsilon^2 \rho^2}\right)$ in the episodic setting, or $\tilde{O}\left(m \cdot \frac{\log^4(1/\delta)}{\rho^2} + \frac{SAH \cdot \log^4(1/\delta)}{\varepsilon^2 \rho^2}\right)$ in the parallel sampling setting. Furthermore, the runtime is $\text{poly}\left(mH\rho^{-1}\varepsilon^{-1} \log(1/\delta)\right)$*
*$+ \tilde{O}\left(\rho^{-2} \log^4(1/\delta)\right) T(\mathcal{L})$, where we denote by $T(\mathcal{L})$ the runtime of $\mathcal{L}$.*

A key sub-routine is the following procedure for selecting heavy hitters of a distribution replicably. This has been studied in several works (Impagliazzo et al., 2022; Kalavasis et al., 2023; Hopkins et al., 2024).

**Lemma D.2** ((Kalavasis et al., 2023; Hopkins et al., 2024)). *Let $\mathcal{X}$ be a domain. For any any distribution $\mathbf{p}$ over $\mathcal{X}$ and $\nu, \delta, \rho, \varepsilon \in (0, 1)$ with $4\delta < \rho, 4\varepsilon < \nu$, REPHEAVYHITTERS$(\mathbf{p}, \nu, \varepsilon, \rho, \delta)$ is a $\rho$-replicable algorithm outputting $S \subset \mathcal{X}$ satisfying the following with probability at least $1 - \delta$:*

1. *If $\mathbf{p}(x) > \nu'$, then $x \in S$,*

2. *If $\mathbf{p}(x) < \nu'$, then $x \notin S$,*

*where $\nu' \in (\nu - \varepsilon, \nu + \varepsilon)$. Furthermore, the algorithm has sample complexity $O\left(\frac{1}{(\nu-\varepsilon)\varepsilon^2 \rho^2} \log \frac{1}{\delta(\nu-\varepsilon)}\right)$, and is efficient.*

*Proof of Lemma D.1.*  We consider the episodic setting as the parallel sampling setting can be treated in an almost identical manner. Let $\mathcal{L}$ be a $0.1$-replicable $(\varepsilon/2, 0.1)$-PAC policy estimator in the episodic setting. Consider the following standard amplification procedure (see e.g., (Hopkins et al., 2024)). Sample $k = O(\log(1/\delta))$ random strings $\{\xi_i\}_{i=1}^{k}$. Denote by $\mathbf{p}_i$ the output distribution of $\mathcal{L}$ using $\xi_i$ as the seed, where the randomness is over that of the MDP. Set $\rho' \leftarrow \frac{\rho}{2k}$ and $\delta' \leftarrow \frac{\delta}{3k}$. For $1 \leq i \leq k$, run $S_i \leftarrow$ REPHEAVYHITTERS$(\mathbf{p}_i, 0.6, 0.05, \rho', \delta')$ to obtain heavy hitter policies. In (Hopkins et al., 2024), the rest of the procedure is to choose a random output from $\bigcup_i S_i$. In our case, we make the slight modification that we will design a $\rho$-replicable procedure that finds a nearly optimal policy among $\bigcup_i S_i$ . Note that for each policy $\pi \in \bigcup_i S_i$, we can easily evaluate its total return with $1$ episode of interaction. Therefore, this can be achieved by running the $\rho$-replicable algorithm that solves the $(A := |\bigcup_i S_i|, \varepsilon/2)$-best arm problem with probability $\delta/3$ (see Theorem C.3).

For replicability, it suffices to union bound over all $k$ instances of REPHEAVYHITTERS, each of which is $\frac{\rho}{2k}$-replicable, and the best arm identification algorithm, which is $\rho/2$-replicable.

For correctness, we need the concept of "good seed strings". We say $\xi$ is a good seed string if (1) there is a canonical hypothesis $\pi$ such that $\mathcal{L}$ running with seed $\xi_i$ produces $\pi$ with probability at least $2/3$, where the randomness is over the MDP, and (2) the canonical hypothesis $\pi$ is $\varepsilon/2$-optimal. By a counting argument similar to that in (Hopkins et al., 2024), we can conclude that a random string $\xi$ is a good seed with probability at least $2/3$. It follows that there is at least one good seed string $\xi_i$ among the $k$ random strings sampled with probability at least $1 - \delta/3$. We will condition on the above event

and that all $k$ instances of **REPHEAVYHITTERS**, and the best arm algorithm succeed, which happen with probability at least $1 - \delta$ by the union bound. For that good random string $\xi_i$, the corresponding heavy hitter is then guaranteed to be a singleton $\{\pi\}$ for some $\varepsilon/2$-optimal policy $\pi$. Then, the best-arm algorithm with accuracy $\varepsilon/2$ is guaranteed to find some policy that is $\varepsilon$-optimal.

Next, we analyze the number of episodes consumed. Each execution of **REPHEAVYHITTERS** invokes $\mathcal{L}$ for $O\left(\rho'^{-2}\log(1/\delta')\right)$ many times. Since there are $k$ executions of **REPHEAVYHITTERS**, this process consumes at most $\tilde{O}\left(\rho^{-2}\log^4(1/\delta)m\right)$ samples. The total number of arm queries of the best-arm algorithm is $\tilde{O}\left(|\bigcup_i S_i|\log^3(1/\delta)\varepsilon^{-2}\rho^{-2}\right)$. We note that the size of each $|S_i|$ is at most 1 as there can be at most 1 element in a distribution with mass more than $0.6$. Moreover, each arm query is simulated by running the corresponding policy $\pi$ with one episode, which consumes $H$ samples. Therefore, the number of samples consumed by the best arm algorithm is at most $\tilde{O}\left(\log^4(1/\delta)\varepsilon^{-2}\rho^{-2}\right)$.

Lastly, we analyze the runtime of the algorithm. Given oracle access to $\mathcal{L}$, our algorithm simply executes **REPHEAVYHITTERS** and the best arm algorithm of Theorem C.3. Both of these algorithms have run-time polynomial in sample complexity. The number of times we invoke $\mathcal{L}$ is at most $\tilde{O}\left(\rho^{-2}\log^4(1/\delta)\right)$ as discussed in the sample complexity analysis. Thus, in total, the runtime is at most $\text{poly}\left(mH\rho^{-1}\varepsilon^{-1}\log(1/\delta)\right) + T(\mathcal{L})\ \tilde{O}\left(\rho^{-2}\log^4(1/\delta)\right)$. This concludes the proof of Lemma D.1. $\qquad\square$

### D.1. Replicable Tiered Backward Induction

We begin with an algorithm that replicably produces a policy given sufficiently large datasets $\mathcal{D}_{s,a,h}$ consisting of data sampled from the transition/reward distributions of the MDP. At a high level, the algorithm replicably reduces offline reinforcement learning to best arm, and then leverages the result of Proposition C.6.

Informally, the result serves as an intermediate step between the full episodic setting and the easier parallel sampling setting. In the latter, one can collect an equal number of samples from the transition and reward distributions from each state-action pair. Yet, this is challenging to achieve in the episodic setting as it is often more difficult to collect samples from certain states than the others. To accommodate this technical issue, we consider a partition of states into tiers based on their reachability under some optimal policy $\pi^*$.

**Definition D.3** (Tiered Reachability Partition). Fix some MDP $\mathcal{M}$. Let $\{\mathcal{S}_h^\ell\}_{h\in[H],\ell\in[L]}$ be a collection of subsets of states such that $\{\mathcal{S}_h^\ell\}_{\ell\in[L]}$ form a partition of $\mathcal{S}$ for each $h \in [H]$. We say they form a tiered reachability partition of $\mathcal{M}$ if there exists some optimal policy $\pi^*$ such that for each $\ell \geq 2$ the state combination $\{\mathcal{S}_h^\ell\}_{h\in[H]}$ satisfies:

$$\sum_{h\in[H]} \Pr[x_h \in \mathcal{S}_h^\ell \mid \pi^*] \leq 2^{1-\ell}. \tag{11}$$

For states within tiers $\ell \gg \log(H/\varepsilon)$, their reachability will be much smaller than $\varepsilon/H$ under the optimal policy $\pi^*$ by Equation (11). As a result, their contribution to the $V^*/Q^*$ values of the $\mathcal{M}$ will be much smaller than $\varepsilon$. Thus, we can essentially ignore these states when designing the offline RL algorithm. Following similar reasoning, one can conclude that it suffices to learn the $V^*/Q^*$ values of states within the $\ell$-th tier up to accuracy roughly $(\varepsilon/H)2^\ell$ to obtain an $\varepsilon$-optimal policy, allowing us to adopt different sample requirements for different tiers.

A further complication arises from the fact that the number of samples collected from states within a single tier may also differ due to statistical fluctuations in the episodic setting. We hence further relax the sample requirement to allow each dataset $\mathcal{D}_{s,a,h}$ to have a random number of samples as long as their sizes are sufficient for the application of the multi-instance best arm algorithm from Proposition C.6 within the tier. Formally, we give the definition of $\zeta$-nice Tiered Offline Datasets, where $\zeta$ controls the overall size of the datasets collected (i.e., smaller $\zeta$ leads to larger datasets), and consequently influences the accuracy of the policy learned by our RL algorithm.

**Definition D.4** (Nice Tiered Offline Datasets). Fix some MDP $\mathcal{M}$. Let $\zeta > 0$, $\{\mathcal{S}_h^\ell\}_{h\in[H],\ell\in[L]}$ be a reachability state partition of $\mathcal{M}$, and $\{\mathcal{D}_{s,a,h}\}_{s\in\mathcal{S},a\in\mathcal{A},h\in[H]}$ be some random datasets consisting of samples of the form $\mathcal{S} \times [0,1]$. We say the distribution of $\{\mathcal{D}_{s,a,h}\}_{s\in\mathcal{S},a\in\mathcal{A},h\in[H]}$ is $\zeta$-nice with respect to $\{\mathcal{S}_h^\ell\}_{h\in[H],\ell\in[L]}$[14] if the following are satisfied:

1. **Sample Independence** For each $s \in \mathcal{S}, a \in \mathcal{A}, h \in [H]$, conditioning on the sizes of each dataset $n_{s,a,h} := |\mathcal{D}_{s,a,h}|$,

---

[14] When it is clear from the context that $\{\mathcal{D}_{s,a,h}\}$ are random datasets, we will directly write that $\{\mathcal{D}_{s,a,h}\}$ are $\zeta$-nice with respect to $\{\mathcal{S}_h^\ell\}_{h,\ell}$.

the resulting conditional joint distribution over the datasets is exactly the product distribution $\prod_{(s,a,h)}(p_h(s,a) \otimes r_h(s,a))^{n_{s,a,h}}$.

2. **Variable Sample Lower Bound** For every $\ell \in [L-1], h \in [H]$, and $s \in \mathcal{S}_h^\ell$, there exists a number $m_{s,h,\ell}$ such that $\Pr[\min_a |\mathcal{D}_{s,a,h}| \geq m_{s,h,\ell}] = 1$. Moreover, these numbers satisfy that:

$$\sum_{h \in [H]} \sqrt{\sum_{s \in \mathcal{S}_h^\ell} 1/m_{s,h,\ell}} \leq 2^\ell \zeta. \tag{12}$$

We will show in Lemma E.1 how to simultaneously identify a tiered reachability partition $\{\mathcal{S}_h^\ell\}_{h,\ell}$ and build a collection of random datasets $\{\mathcal{D}_{s,a,h}\}$ that is $\zeta$-nice with respect to the partition.

Our main result of this subsection is as follows: given some reachability state partition $\{\mathcal{S}_h^\ell\}_{h,\ell}$ and some collection of random datasets $\{\mathcal{D}_{s,a,h}\}$ that is sufficiently nice with respect to $\{\mathcal{S}_h^\ell\}_{h,\ell}$, we given an algorithm that computes an $\varepsilon$-optimal policy.

**Theorem D.5** (Tiered Backward Induction). *Fix some MDP $\mathcal{M}$. Let $\varepsilon > 0$, $\zeta \ll \varepsilon/\left(H^2 \log^5(SAH\varepsilon^{-1}\delta^{-1})\right)$ and $L = \lceil \log(1/\zeta) \rceil$. Let $\{\mathcal{S}_h^\ell\}_{h \in [H], \ell \in [L]}$ be a reachability state partition of $\mathcal{M}$ and $\{\mathcal{D}_{s,a,h}\}$ be a collection of random datasets that are $\zeta$-nice with respect to $\{\mathcal{S}_h^\ell\}_{h \in [H], \ell \in [L]}$. $\mathbf{REPRLBANDIT}$ (Algorithm 2) is a $0.01$-replicable[15] algorithm that outputs an $\varepsilon$-optimal policy on $\mathcal{M}$ with probability $1 - \delta$. Moreover, the algorithm runs in time $\mathrm{poly}\left(\sum_{s,a,h} |\mathcal{D}_{s,a,h} + 1|\right) + HL \exp\left((1 + o(1))S \log A\right)$.*

Combining Lemma E.1 and Theorem D.5 will then yield our main RL algorithm in the episodic setting. In the rest of this subsection, we will present the detail of the algorithm $\mathbf{REPRLBANDIT}$ from Theorem D.5 and its analysis.

---

**Algorithm 2 REPRLBANDIT**

---

**Input:** collections of states $\{\mathcal{S}_h^\ell\}_{h \in [H], \ell \in [L]}$ and datasets $\{\mathcal{D}_{s,a,h}\}_{s \in \mathcal{S}, a \in \mathcal{A}, h \in [H]}$.
**Parameters:** accuracy parameter $\varepsilon$ and failure parameter $\delta$.
**Output:** policy $\hat{\pi} : [H] \times \mathcal{S} \to \mathcal{A}$.

1. Construct the dataset $\mathcal{R}_{s,a} := \{r : \forall(\perp, r) \in \mathcal{D}_{s,a,H}\}$ for each $s \in \mathcal{S}, a \in \mathcal{A}$. $\perp$ *denotes that no transition is observed at step $H$.*

2. For $h \in [H]$ in decreasing order:0

    (a) For each $\ell \in [L-1]$:

        i. Compute $\{a_s^{(h)}, \hat{r}_s^{(h)}\}_{s \in \mathcal{S}_h^\ell} \leftarrow \mathbf{REPVARBANDIT}\left(\{\mathcal{R}_{s,a}\}_{s \in \mathcal{S}_h^\ell, a \in \mathcal{A}}, \frac{2^\ell \varepsilon}{8HL}, \frac{\delta}{HL}\right)$ for arbitrarily small constant error probability $c$.

        ii. For $s \in \mathcal{S}_h^\ell$, let $\bar{r}_s^{(h)} \leftarrow \hat{r}_s^{(h)} - \frac{2^\ell \varepsilon}{8HL}$. *This ensures that $\bar{r}_s^{(h)}$ is an* underestimate *of the expected utility of $a_s^{(h)}$ with high probability.*

    (b) For $\ell = L$ and $s \in \mathcal{S}_h^\ell$, set $a_s^{(h)}$ to the first action, and $\bar{r}_s^{(h)} = 0$.

    (c) Let $\hat{\pi}_h(s) \leftarrow a_s^{(h)}$ for $s \in \mathcal{S}$.

    (d) Construct a new dataset $\mathcal{R}_{s,a} := \{r + \bar{r}_x^{(h)} : \forall(x, r) \in \mathcal{D}_{s,a,h-1}\}$ for each $s \in \mathcal{S}, a \in \mathcal{A}$.

3. Return $\hat{\pi}$.

---

Our algorithm is structurally similar to standard backwards induction but takes additional advantage of the knowledge of the tiered reachability partition $\{\mathcal{S}_h^\ell\}$ to determine the requisite estimation accuracy for the value function of each state. Fix some $\varepsilon_0 = \frac{\varepsilon}{8HL}$. At the $H$-th time step, we have a partition of the states $\mathcal{S}$ into tiers $\{\mathcal{S}_H^\ell\}_{\ell \in [L]}$. For each tier $\ell$, we

---

[15]Here we slightly abuse the notation of replicability as technically the definition of replicability requires the algorithm's outputs to be identical with high probability regardless of the distribution of the input data. However, $\mathbf{REPRLBANDIT}$ is only replicable when $\{\mathcal{D}_{s,a,h}\}$ are promised to be $\zeta$ nice.

invoke **REPVARBANDIT** (Proposition C.6) on states $\mathcal{S}_H^\ell$ using the random datasets $\{\mathcal{D}_{s,a,H}\}_{s \in \mathcal{S}_H^\ell, a \in \mathcal{A}}$ with accuracy $2^\ell \varepsilon_0$ and failure probability $\frac{\delta}{HL}$.

Since the random datasets are sufficiently nice with respect to the partition, within each tier $\ell$, we can show that **REPVARBANDIT** must replicably outputs a $(2^\ell \varepsilon_0)$-optimal arm $a_s^{(H)}$ and a $(2^\ell \varepsilon_0)$-accurate estimate $\bar{r}_s^{(H)}$ of the expected reward of the chosen arm. Intuitively, the error $(2^\ell \varepsilon_0)$ will be good enough for tier $\ell$ as their reachability should be at most $O(2^{-\ell})$ (see Equation (11)).

Note that the estimate $\bar{r}_s^{(H)}$ can be thought as estimate of $V_H^*(s)$. Thus, we can construct a new dataset $\mathcal{R}_{s,a} := \{r + \bar{r}_s^{(H)} : \forall (x,r) \in \mathcal{D}_{s,a,H-1}\}$, which can be used to estimate $Q_{H-1}^*(s,a)$. In step $H-1$, **REPVARBANDIT** is again used to choose a nearly-optimal action for each state, and to estimate the $V^*$ values. We proceed in this way until a nearly-optimal action has been chosen for all states at all time steps, which then yields the final output policy.

**Analysis of Reinforcement Learning Algorithm**  We begin by showing the replicability of the algorithm.

**Lemma D.6** (Policy Replicability). *Let $\varepsilon, \delta, \zeta, L$ be defined as in Theorem D.5. Let $\{\mathcal{S}_h^\ell\}_{h \in [H], \ell \in [L]}$ be a fixed tiered reachability partition of $\mathcal{M}$. Suppose $\{\mathcal{D}_{s,a,h}\}$ and $\{\mathcal{D}'_{s,a,h}\}$ are collections of random datasets that are both $\zeta$-nice with respect to $\{\mathcal{S}_h^\ell\}$. Then it holds that*

$$\Pr\left(\textbf{REPRLBANDIT}(\varepsilon, \delta, \{\mathcal{S}_h^\ell\}, \{\mathcal{D}_{s,a,h}\}; \xi) \neq \textbf{REPRLBANDIT}(\varepsilon, \delta, \{\mathcal{S}_h^\ell\}, \{\mathcal{D}'_{s,a,h}\}); \xi) \leq 0.01.$$

*where the randomness is over $\mathcal{D}_{s,a,h}, \mathcal{D}'_{s,a,h}$ and the shared internal randomness of $\xi$.*

*Proof.* We first establish the following guarantee on the dataset sample sizes.

*Claim* D.7. For all $\ell \in [L-1]$, we have that $\min_a |\mathcal{R}_{s,a}| \geq m_{s,h,\ell}$, where $m_{s,h,\ell}$ are numbers satisfying

$$\sum_h \sqrt{\sum_{s \in \mathcal{S}_h^\ell} \frac{1}{m_{s,h,\ell}}} \leq 2^\ell \zeta \leq \frac{\left(\frac{0.01}{L}\right)\left(\frac{2^\ell \varepsilon}{HL}\right)}{CH \log^{3/2}(SAHL/\delta)} \tag{13}$$

for a sufficiently large constant $C$.

*Proof.* By the definition of $\zeta$-niceness, for each $\ell \in [L], h \in [H]$, and $s \in \mathcal{S}_H^\ell$,

$$\sum_{h \in [H]} \sqrt{\sum_{s \in \mathcal{S}_h^\ell} \frac{1}{m_{s,h,\ell}}} \leq 2^\ell \zeta \ll \frac{2^\ell \varepsilon}{H^2 \log^5(SAH\varepsilon^{-1}\delta^{-1})}.$$

Note that for such $\zeta$ we have $L = O(\log(1/\zeta)) = O\left(\log(H/\varepsilon) + \log\log(SAH\varepsilon^{-1}\delta^{-1})\right)$. Then

$$\zeta L^3 \ll \frac{\varepsilon L^3}{H^2 \log^5(SAH\varepsilon^{-1}\delta^{-1})} \ll \frac{\varepsilon}{H^2 \log^2(SAH\varepsilon^{-1}\delta^{-1})} \ll \frac{\varepsilon}{H^2 \log^{3/2}(SAHL\delta^{-1})}.$$

Rearranging terms gives us the desired bound of Equation (13). $\square$

Rearranging Equation (13) gives that

$$\forall \ell \in [L], \frac{CH \log^{3/2}(SAHL/\delta)}{\left(\frac{2^\ell \varepsilon}{HL}\right)} \sum_h \sqrt{\sum_{s \in \mathcal{S}_h^\ell} \frac{1}{m_{s,h,\ell}}} \leq \frac{0.01}{L}$$

If we define

$$\rho_{h,\ell} := \frac{CH \log^{3/2}(SAHL/\delta)}{\left(\frac{2^\ell \varepsilon}{HL}\right)} \sqrt{\sum_{s \in \mathcal{S}_h^\ell} \frac{1}{m_{s,h,\ell}}},$$

it follows that

$$\sum_{h,\ell} \rho_{h,\ell} \leq 0.01. \tag{14}$$

Consider the execution of **REPVARBANDIT** at Line 2(a)i when $h = H$ and $\ell \in [\log(1/\zeta) - 1]$. Note that $\mathcal{R}_{s,a}$ just consists of i.i.d. samples from $\mathbf{r}_H(s,a)$. So it follows that $\mathcal{R}_{s,a}$ is drawn from the same distribution in both runs of the algorithm.

Thus, Proposition C.6 ensures that the outputs of **REPVARBANDIT** are $\rho_{H,\ell}$-replicable when we run it on the states $\mathcal{S}_H^\ell$. By a union bound over $\ell$, in the $H$-th iteration, the policy $\hat{\pi}_H$ and the estimates $\hat{r}_s^{(H)}, \overline{r}_s^{(H)}$ are identical in two runs of the algorithm with probability at least $1 - \sum_\ell \rho_{H,\ell}$.

For the step $h \leq H - 1$, we will use induction and proceed by a similar argument. We will condition on that the estimates $\{\hat{r}_s^{(h+1)}\}_{s \in \mathcal{S}}$ are identical over both runs of the algorithm in the inductive step. In Line 2d, we update the dataset $\mathcal{R}_{s,a}$ to be $\{r + \overline{r}_x^{(h+1)} : \forall (x,r) \in \mathcal{D}_{s,a,h}\}$. Since we condition on that $\overline{r}_x^{(h+1)}$ is identical in the two run, and $D_{s,a,h-1}$ just consists of i.i.d. samples from the corresponding transition/reward distributions of the MDP, it follows that the distribution of $\mathcal{R}_{s,a}$ is identical in two runs of the algorithm. Therefore, following an argument that is almost identical to the one used in the base case ($h = H$), we can conclude that $\hat{\pi}_h$ and $\hat{r}^{(h)}$ (and therefore $\overline{r}^{(h)}$) are identical with probability at least $1 - \sum_{\ell=1}^L \rho_{h,\ell}$. Union bounding over all $H$ steps, we obtain that the final policy $\hat{\pi}$ is identical with probability at least $1 - \sum_{h,\ell} \rho_{h,\ell} \geq 0.99$, where the last inequality follows from Equation (14). This concludes the proof of Lemma D.6. $\square$

To analyze the correctness of our algorithm, we introduce the concept of a Truncated Markov Decision Process.

**Definition D.8** (Truncated MDP). Let $\mathcal{M}$ be an MDP with $h$ time steps, and $V : \mathcal{S} \to \mathbb{R}_+$ be some function. Let $\mathbf{r}_{h-1} : \mathcal{S} \times \mathcal{A} \to \Delta(\mathcal{S})$ be the reward distributions, and $\mathbf{p}_{h-1} : \mathcal{S} \times \mathcal{A} \to \Delta(\mathcal{S})$ be the transition distributions at the $(h-1)$-th step of $\mathcal{M}$. We define the truncated MDP of $\mathcal{M}$ with substitution function $V$ as the MDP $\mathcal{M}'$ with $h - 1$ time steps that

1. has the same transition and rewards as $\mathcal{M}$ for the first $(h-2)$-th time steps,

2. receives the reward $r + V(x)$, where $r \sim \mathbf{r}_h(s,a)$ and $x \sim \mathbf{p}_h(x,a)$ for the state-action pair $(s,a)$ at the $(h-1)$-th time step.

The key idea is to consider a sequence of Truncated MDPs induced by the estimates $\{\overline{r}_s^{(h)}\}_{s,h}$. In particular, we denote by $\mathcal{M}^{(H)}$ the original MDP $\mathcal{M}$, and $\mathcal{M}^{(h)}$ the truncated MDP of $\mathcal{M}^{(h+1)}$ with the substitution function $V_{h+1}(s) := \overline{r}_s^{(h)}$. We first show that the truncation does not significantly affect the value of the optimal policy.

**Lemma D.9** (Truncation Approximately Preserves MDP Values). *Let $h \in [H] \backslash \{1\}$ and $\pi^*$ be some optimal policy for the MDP $\mathcal{M}$. It holds that $V(\pi^*; \mathcal{M}^{(h-1)}) \geq V(\pi^*; \mathcal{M}^{(h)}) - \frac{\varepsilon}{H}$ with probability at least $1 - \frac{\delta}{H}$.*

*Proof.* We introduce the following extra notations. Denote by $X_t$ the distribution of the state at step $t$ in the original MDP $\mathcal{M}$ under the policy $\pi^*$ for all $t \in [H]$, and $\mu_h(s,a)$ the expected value of the distribution $\mathbf{r}_h(s,a)$. The two MDPs are identical up to the first $(h-2)$ steps. In particular, the expected rewards at step $t \leq h-2$ in $\mathcal{M}^{(h-1)}$ and $\mathcal{M}^{(h)}$ are both $\mathbb{E}_{x \sim X_t}[\mu_t(x, \pi^*(x))]$. At the $(h-1)$-th step, the agent in $\mathcal{M}^{(h-1)}$ will receive in expectation a reward of

$$\mathbb{E}_{x \sim X_{h-1}}\left[\mu_{h-1}(x, \pi_{h-1}^*(x)) + \mathbb{E}_{y \sim \mathbf{p}_{h-1}(x, \pi_{h-1}^*(x))}\left[\overline{r}_y^{(h)}\right]\right] = \mathbb{E}_{x \sim X_{h-1}}\left[\mu_{h-1}(x, \pi_{h-1}^*(x))\right] + \mathbb{E}_{y \sim X_h}\left[\overline{r}_y^{(h)}\right],$$

where in the equality we use the observation that $y \sim \mathbf{p}_{h-1}(x, \pi_{h-1}^*(x))$ for $x \sim X_{h-1}$ is distributed exactly the same as $y \sim X_h$. In $\mathcal{M}^{(h)}$, the agent will receive in expectation a total reward of

$$\mathbb{E}_{x \sim X_{h-1}}\left[\mu_{h-1}(x, \pi_{h-1}^*(x))\right] + \mathbb{E}_{x \sim X_h}\left[\mu_h(x, \pi_h^*(x)) + \mathbb{E}_{y \sim \mathbf{p}_h(x, \pi_h^*(x))}\left[\overline{r}_y^{h+1}\right]\right]$$

in step $h - 1$ and step $h$. From the above discussion, we obtain that

$$V(\pi^*, \mathcal{M}^{(h)}) - V(\pi^*, \mathcal{M}^{(h-1)}) = \mathbb{E}_{s \sim X_h}\left[\left(\mu_h(s, \pi_h^*(s)) + \mathbb{E}_{y \sim \mathbf{p}_h(s, \pi_h^*(s))}[\overline{r}_y^{(h+1)}] - \overline{r}_s^{(h)}\right)\right]. \tag{15}$$

In the following analysis, we will condition on some arbitrary values of $\{\hat{r}_s^{(h+1)}\}_{s\in\mathcal{S}}$ and prove some concentration bounds on $\hat{r}_s^{(h)}$. Recall that $\mathcal{S}_h^\ell$ for $1 \le \ell \le L = \lceil \log(1/\zeta) \rceil$ is a partition of $\mathcal{S}$. Fix some $h \in [H]$ and $\ell < L$. By the definition of $\zeta$-niceness and the assumption on the size of $\zeta$, for each $s \in \mathcal{S}_h^\ell$ and action $a \in \mathcal{A}$, the dataset $\mathcal{D}_{s,a,h}$ is of size at least $m_{s,h,\ell}$, where $\{m_{s,h,\ell}\}_{s\in\mathcal{S}_h^\ell}$ satisfy the guarantees of Claim D.7.

The dataset is then transformed into $\mathcal{R}_{s,a} := \{r + \overline{r}_x^{(h+1)} : \forall (x,r) \in \mathcal{D}_{s,a,h}\}$ of the same size in Line 2d. Observe that a sample in $\mathcal{R}_{s,a}$ can therefore be viewed as a sample from $r + \overline{r}_y^{(h+1)} \in [0, H]$, where $r \sim \mathbf{r}_h(s,a)$ and $y \sim \mathbf{p}_h(s,a)$. Recall that we apply **REPVARBANDIT** with input datasets $\{\mathcal{R}_{s,a}\}_{s\in\mathcal{S}_h^\ell, a\in\mathcal{A}}$, accuracy $\frac{2^\ell \varepsilon}{8HL}$, and failure probability $\frac{\delta}{HL}$ in Line 2(a)i. By Proposition C.6 and the dataset size lower bound stated in Claim D.7, with probability at least $1 - \frac{\delta}{HL}$, the action $a_s^{(h)}$ learned must satisfy that

$$\forall s \in \mathcal{S}_h^\ell : \mu_h(s, a_s^{(h)}) + \mathbb{E}_{z\sim\mathbf{p}_h(s,a_s^{(h)})}[\overline{r}_z^{(h+1)}] \ge \max_{a\in\mathcal{A}} \mu_h(s,a) + \mathbb{E}_{z\sim\mathbf{p}_h(s,a)}[\overline{r}_z^{(h+1)}] - \frac{2^\ell \varepsilon}{8HL} \tag{16}$$

and the estimate $\hat{r}^{(h)}$ must satisfy that

$$\forall s \in \mathcal{S}_h^\ell : \left| \hat{r}_s^{(h)} - \left( r_h(s, a_s^{(h)}) + \mathbb{E}_{z\sim\mathbf{p}_h(s,a_s^{(h)})}[\overline{r}_z^{(h+1)}] \right) \right| \le \frac{2^\ell \varepsilon}{8HL}. \tag{17}$$

Note that Equation (16) and Equation (17) fail to hold with probability at most $\frac{\delta}{HL}$ for a fixed $\ell$. By a union bound, they hold simultaneously for all $\ell$ with probability at least $1 - \frac{\delta}{H}$. Conditioned on that, we can conclude that $\overline{r}_s^{(h)} \leftarrow \hat{r}_s^{(h)} - \frac{2^\ell \varepsilon}{8HL}$ and $a_s^{(h)}$ together enjoy the guarantee:

$$\forall s \in \mathcal{S}_h^\ell : \mu_h(s, a_s^{(h)}) + \mathbb{E}_{z\sim\mathbf{p}_h(s,a_s^{(h)})}[\overline{r}_z^{(h+1)}] - \frac{2^\ell \varepsilon}{4HL} \le \overline{r}_s^{(h)} \le \mu_h(s, a_s^{(h)}) + \mathbb{E}_{z\sim\mathbf{p}_h(s,a_s^{(h)})}[\overline{r}_z^{(h+1)}]. \tag{18}$$

Combining Equation (18) and Equation (16) gives that

$$\begin{aligned}
\overline{r}_s^{(h)} &\ge \mu_h(s, a_s^{(h)}) + \mathbb{E}_{z\sim\mathbf{p}_h(s,a_s^{(h)})}[\overline{r}_z^{(h+1)}] - \frac{2^\ell \varepsilon}{4HL} \\
&\ge \max_a \mu_h(s,a) + \mathbb{E}_{z\sim\mathbf{p}_h(s,a)}[\overline{r}_z^{(h+1)}] - \frac{2^\ell \varepsilon}{2HL} \\
&\ge \mu_h(s, \pi_h^*(s)) + \mathbb{E}_{y\sim\mathbf{p}_h(s,\pi_h^*(s))}[\overline{r}_y^{(h+1)}] - \frac{2^\ell \varepsilon}{2HL}
\end{aligned} \tag{19}$$

Thus, combining the above bound with Equation (15) gives that

$$\begin{aligned}
& V\left(\pi^*, \mathcal{M}^{(h)}\right) - V\left(\pi^*, \mathcal{M}^{(h-1)}\right) \\
&= \sum_{s\in\mathcal{S}} \Pr[X_h = s]\left(\mu_h(s, \pi_h^*(s)) + \mathbb{E}_{y\sim\mathbf{p}_h}\left[\overline{r}_y^{(h+1)}\right] - \overline{r}_s^{(h)}\right) \\
&= \sum_{\ell=1}^{L} \sum_{s\in\mathcal{S}_h^\ell} \Pr[X_h = s]\left(\mu_h(s, \pi_h^*(s)) + \mathbb{E}_{y\sim\mathbf{p}_h(s,\pi_h^*(s))}[\overline{r}_y^{(h+1)}] - \overline{r}_s^{(h)}\right) \\
&\le \left(\sum_{\ell=1}^{L-1} \frac{2^\ell \varepsilon}{2HL}\left(\sum_{s\in\mathcal{S}_h^\ell} \Pr[X_h = s]\right)\right) + \left(H\sum_{s\in\mathcal{S}_h^L} \Pr[X_h = s]\right) \\
&\le \left(\sum_{\ell=1}^{L-1} \frac{2^\ell \varepsilon}{2HL} 2^{-\ell}\right) + H\zeta \\
&\le \left(\sum_{\ell=1}^{L-1} \frac{\varepsilon}{2HL}\right) + \frac{\varepsilon}{2HL} \le \frac{\varepsilon}{H},
\end{aligned}$$

where the first inequality follows from Equation (15). the second equality follows from the fact that $\{\mathcal{S}_h^\ell\}_{\ell=1}^{\lceil \log(1/\zeta)\rceil}$ forms a partition of $\mathcal{S}$, the first inequality follows from Equation (19), the second inequality follows from the reachability property Equation (11) of the partition, the third inequality follows from the assumption on the size of $\zeta$. This concludes the proof of Lemma D.9. $\square$

By the union bound on $h$, it is easy to check that we must have $V\left(\pi^*; \mathcal{M}^{(1)}\right) \geq V\left(\pi^*; \mathcal{M}\right) - \varepsilon$ with probability $1 - \delta$. However, note that $\mathcal{M}^{(1)}$ is simply a one-step MDP, which corresponds to exactly some multi-instance best arm problem. Therefore, it is not hard to show that the policy $\hat{\pi}$ learned will be a nearly-optimal policy with respect to $\mathcal{M}^{(1)}$. It then remains to relate $V(\hat{\pi}, \mathcal{M}^{(1)})$ to the expected reward of the policy under the original MDP. For that purpose, we use the fact that the estimates $\overline{r}_s^{(H)}$ with high probability underestimate the utilities of the actions chosen by $\hat{\pi}$ due to the subtraction term in Line 2(a)ii. As a result, the value of $\hat{\pi}$ under the truncated MDP is never higher than the original MDP.

**Lemma D.10** (Truncation Decreases Policy Return). *Let $\hat{\pi}$ be the policy learned by Algorithm 2, and $h \in [H]\backslash\{1\}$. We have that $V(\hat{\pi}, \mathcal{M}^{(h)}) \geq V(\hat{\pi}, \mathcal{M}^{(h-1)})$ with probability at least $1 - \delta/H$.*

*Proof.* Denote by $X_t$ the distribution of the state at step $t$ in the original MDP $\mathcal{M}$ under the policy $\hat{\pi}$ for all $t \in [H]$, and $\mu_h(s, a)$ the expected value of the distribution $\mathbf{r}_h(s, a)$. Following an argument similar to the one for Equation (15), we obtain that

$$V\left(\hat{\pi}, \mathcal{M}^{(h)}\right) - V\left(\hat{\pi}, \mathcal{M}^{(h-1)}\right) = \mathbb{E}_{s\sim X_h}\left[\left(\mu_h(s, \pi_h(s)) + \mathbb{E}_{y\sim\mathbf{p}_h(s,\hat{\pi}_h(s))}[\overline{r}_y^{(h+1)}] - \overline{r}_s^{(h)}\right)\right]. \quad (20)$$

We have already shown in the proof of Lemma D.9 that

$$\forall \ell \in \lceil \log(1/\zeta)\rceil \text{ and } s \in \mathcal{S}_h^\ell : \overline{r}_s^{(h)} \leq \mu_h(s, a_s^{(h)}) + \mathbb{E}_{z\sim\mathbf{p}_h(s,a_s^{(h)})}[\overline{r}_z^{(h+1)}]$$
$$= \mu_h(s, a_s^{(h)}) + \mathbb{E}_{z\sim\mathbf{p}_h(s,\hat{\pi}_h(x))}[\overline{r}_z^{(h+1)}] \quad (21)$$

with probability at least $1 - \delta/H$. Conditioned on Equation (21), it follows that the expression inside the expectation on the right hand side of Equation (20) is always non-negative. This then concludes the proof of Lemma D.10. $\square$

With Lemmas D.9 and D.10 at our hands, it follows almost immediately that the learned policy $\hat{\pi}$ is nearly optimal with respect to the original MDP $\mathcal{M}$.

**Lemma D.11** (Learned Policy is Nearly Optimal). *With probability at least $1 - \delta$, the policy $\hat{\pi}$ learned by REPRLBAN-DIT(Algorithm 2) is an $\varepsilon$-optimal policy.*

*Proof.* Lemma D.10, Lemma D.9, and the union bound implies that with probability at least $1 - 2\delta$

$$\forall h \in [H]\backslash\{1\} : V(\hat{\pi}, \mathcal{M}^{(h)}) \geq V(\hat{\pi}, \mathcal{M}^{(h-1)}) \text{ and } V\left(\pi^*; \mathcal{M}^{(h-1)}\right) \geq V\left(\pi^*; \mathcal{M}^{(h)}\right) - \frac{\varepsilon}{H}.$$

In particular, the above implies that

$$V(\hat{\pi}, \mathcal{M}) \geq V(\hat{\pi}, \mathcal{M}^{(1)}) \text{ and } V\left(\pi^*; \mathcal{M}^{(1)}\right) \geq V\left(\pi^*; \mathcal{M}\right) - \frac{\varepsilon(H-1)}{H}. \quad (22)$$

Note that $\mathcal{M}^{(1)}$ is a one step MDP. Moreover, as discussed in the beginning of Appendix B, we may assume that $\mathcal{M}^{(1)}$ always starts from a fixed state $x_{\text{ini}}$. It is therefore guaranteed that $x_{\text{ini}} \in \mathcal{S}_1^1$. Therefore, for each $a \in \mathcal{A}$ the dataset $\mathcal{D}_{x_{\text{ini}},a,1}$, is of size at least $m_{x_{\text{ini}},1,1} \geq 4\zeta^{-2} \gg H^4/\varepsilon^2 \log(SAH/\delta)$, where the last inequality is by the assumption on the size of $\zeta$. The transformed dataset $\mathcal{R}_{x_{\text{ini}},a}$ enjoys the same size lower bound as $\mathcal{D}_{x_{\text{ini}},a,1}$ and consists of i.i.d. samples from $r + \overline{r}_y^{(1)}$, where $r \sim \mathbf{r}_1(x_{\text{ini}}, a)$ and $y \sim \mathbf{p}_1(x_{\text{ini}}, a)$. By Proposition C.6, with probability at least $1 - \delta$ the action $a_{x_{\text{ini}}}^{(1)}$ must satisfy that

$$\mu_h(s, a_{x_{\text{ini}}}^{(1)}) + \mathbb{E}_{z\sim\mathbf{p}_h(s,a_{x_{\text{ini}}}^{(1)})}[\overline{r}_z^{(2)}] \geq \max_{a\in\mathcal{A}} \mu_h(s, a) + \mathbb{E}_{z\sim\mathbf{p}_h(s,a)}[\overline{r}_z^{(2)}] - \frac{\varepsilon}{4HL}$$
$$\geq \mu_h(x_{\text{ini}}, \pi_1^*(x_{\text{ini}})) + \mathbb{E}_{z\sim\mathbf{p}_h(s,\pi_1^*(x_{\text{ini}}))}[\overline{r}_z^{(2)}] - \frac{\varepsilon}{4HL}.$$

It follows that $V(\hat{\pi}, \mathcal{M}^{(1)}) \geq V(\hat{\pi}^*, \mathcal{M}^{(1)}) - \varepsilon/H$. Combining this with Equation (22), a union bound then yields the following with probability at least $1 - 3\delta$

$$V(\hat{\pi}, \mathcal{M}) \geq V(\hat{\pi}, \mathcal{M}^{(1)}) \geq V(\hat{\pi}^*, \mathcal{M}^{(1)}) - \varepsilon/H \geq V(\hat{\pi}^*, \mathcal{M}) - \varepsilon.$$

We can improve the success probability to $1 - \delta$ with constant factor overhead in sample complexity. This concludes the proof of Lemma D.11. $\qquad\square$

Theorem D.5 follows from the replicability analysis and the correctness analysis.

*Proof of Theorem D.5.* To analyze the runtime, we note that the algorithm invokes **REPVARBANDIT** $O(HL)$ times and all remaining operations are polynomial in sample complexity. Thus, the total runtime is $\text{poly}\left(\sum_{s,a,h} |\mathcal{D}_{s,a,h}|\right) + HL \exp((1 + o(1))S \log A)$ by Proposition C.6. $\qquad\square$

## D.2. Learning with Parallel Sampling

Using Theorem D.5, we give our algorithm for learning in the parallel sampling setting.

**Definition D.12.** Let $C$ be the class of MDPs with $S$ states, $A$ actions, and $H$ time steps. An algorithm is an $(\varepsilon, \delta)$-PAC policy estimator with sample complexity $m \cdot SAH$ in the parallel sampling model if for all fixed MDPs $\mathcal{M}$, given $m$ calls to $\textbf{PS}(G_{\mathcal{M}})$, the algorithm outputs an $\varepsilon$-optimal policy $\pi$ with probability at least $1 - \delta$.

Assume that the time horizon $H$ is some constant. A classic result (Theorem 2.9 of (Agarwal et al., 2019)) shows that one needs $\Theta\left(\log(SA)/\varepsilon^2\right)$ many calls to $\textbf{PS}(G_{\mathcal{M}})$ to find a policy whose return is within $\varepsilon$ of the optimal without replicability. Prior work in the replicable setting (Eaton et al., 2023; Karbasi et al., 2023) focused on the closely related discounted infinite horizon setting, where the agent interacts with the MDP indefinitely but has reward discounted (multiplicatively) in each step by some $\gamma$-discount factor.[16] For constant $\gamma$, (Eaton et al., 2023) obtains an algorithm with $\tilde{O}(S^2 A^2 \varepsilon^{-2} \rho^{-2})$ calls to $\textbf{PS}(G_{\mathcal{M}})$,[17] while (Karbasi et al., 2023) improves the number of calls to $\tilde{O}(SA\varepsilon^{-2}\rho^{-2})$ (which translates to $\tilde{O}(S^2 A^2 \varepsilon^{-2} \rho^{-2})$ many transition/reward samples). We improve the number of calls to $\tilde{O}(S\varepsilon^{-2}\rho^{-2})$.

**Theorem D.13** (Reinforcement Learning with Parallel Sampling). *There is a $\rho$-replicable $(\varepsilon, \delta)$-PAC Policy Estimator with sample complexity $\tilde{O}\left(\frac{S^2 A H^7 \log^4(1/\delta)}{\varepsilon^2 \rho^2}\right)$ in the parallel sampling model. The algorithm runs in time* $\text{poly}\left(H\varepsilon^{-1}\rho^{-1}\log(1/\delta)\exp((1 + o(1))S \log A)\right)$.

We remark that when $\delta, \varepsilon, \rho, H$ are some constants, the bound in Theorem D.13 is nearly optimal (up to polylogarithmic factors) in the parallel sampling model (see Theorem 1.4). We next give the proof of Theorem D.13, which is a simply application of Theorem D.5.

*Proof.* Let $C$ be some sufficiently large constant. Suppose we make $m = C \frac{SH^6 \log^C(A)}{\varepsilon^2}$ calls to $\textbf{PS}(G_{\mathcal{M}})$, which translates to $\tilde{O}\left(\frac{S^2 A H^7}{\varepsilon^2}\right)$ reward and transition samples, so that we obtain datasets $\{\mathcal{D}_{s,a,h}\}$ satisfying $|\mathcal{D}_{s,a,h}| \geq m$ for all $s, a, h$. We design a $0.1$-replicable $(\varepsilon/2, 0.1)$-PAC Policy Estimator and boost it to a $\rho$-replicable $(\varepsilon, \delta)$-PAC Policy Estimator via Lemma D.1.

Consider the trivial partition $\mathcal{S}_h^1 = \mathcal{S}$ for all $h \in [H]$. This is by definition a tiered reachability partition (see Definition D.3). We claim that the collected datasets are sufficiently nice (see Definition D.4) with respect to this trivial tiered reachability partition. In particular, set $m_{s,h,1} = m$ so we have

$$\sum_h \sqrt{\sum_{s \in \mathcal{S}} \frac{1}{m}} = \frac{H\sqrt{S}}{\sqrt{m}} := \zeta \ll \frac{\varepsilon/H}{\left(\log^5(SAH\varepsilon^{-1})\right)}$$

---

[16] Our result is stated for finite-horizon MDPs. However, standard techniques allow one to approximate an MDP with discount factor $\gamma$ with a finite-horizon MDP with $H = \tilde{O}\left(\log(1/\varepsilon)/(1 - \gamma)\right)$. Thus, our result naturally translates to the discounted MDP setting up to logarithmic factors when $\gamma$ is assumed to be constant.

[17] In the infinite-horizon setup, the transition/reward distribution is time invariant. Therefore, each call to $\textbf{PS}(G_{\mathcal{M}})$ will return an i.i.d. sample for every tuple $(s, a) \in \mathcal{S} \times \mathcal{A}$ rather than for every tuple $(s, a, h) \in \mathcal{S} \times \mathcal{A} \times \mathbb{Z}_+$.

as required by Theorem D.5. By Theorem D.5, applying **REPRLBANDIT** to the trivial partition and the obtained datasets then gives an $\varepsilon$-optimal policy with probability at least 0.99. Moreover, if we run the algorithm twice, the partition and the distribution of the datasets will be identical. Thus, the algorithm is also 0.01-replicable. Therefore, we can apply Lemma D.1 to boost the algorithm to a $\rho$-replicable one that succeeds with probability $1 - \delta$ with sample complexity

$$\tilde{O}\left(\frac{S^2 AH^7}{\varepsilon^2}\right) \cdot \tilde{O}\left(\frac{\log^4(1/\delta)}{\rho^2}\right) + H \cdot \tilde{O}\left(\frac{\log^4(1/\delta)}{\rho^2 \varepsilon^2}\right) = \tilde{O}\left(\frac{S^2 AH^7 \log^4(1/\delta)}{\varepsilon^2 \rho^2}\right).$$

This concludes the proof of Theorem D.13. To analyze the runtime, we note that the run-time is simply that of Theorem D.5 combined with Lemma D.1. □

## E. Replicable Strategic Exploration

In this subsection, we show how to replicably and strategically explore the MDP. Specifically, let $\pi$ be some potential optimal policy. The algorithm will divide the states into "tiers" based on their reachability under $\pi$. Then, for each tier of the states, the algorithm will gather sufficiently many samples so that we can leverage the multi-instance best arm algorithm to the $V^*$-values up to some error that is inversely proportional to the reachability of that tier. The formal statement is given below.

**Lemma E.1** (Replicable and Strategic Exploration). *Let $\zeta \in (0,1)$ and $L = \lceil \log(1/\zeta) \rceil$. Then, there is an algorithm* **REPLEVELEXPLORE**$(\mathcal{M}, \zeta)$ *that consumes $N := \tilde{O}(S^2 AH^7 \zeta^{-2})$ samples, and produces*

1. *A 0.01-replicable tiered reachability partition $\{\mathcal{S}_h^\ell\}_{h \in [H], \ell \in [L]}$ of $\mathcal{M}$,*

2. *A collection of datasets $\{\mathcal{D}_{s,a,h}\}_{s \in \mathcal{S}, a \in \mathcal{A}, h \in [H]}$ that are $\zeta$-nice with respect to $\{\mathcal{S}_h^\ell\}$ with probability at least 0.99.*

*Moreover, the runtime of the algorithm is at most $\mathrm{poly}\,(N) \exp\,(O(SH))$.*

We focus on proving Lemma E.1 in the rest of this section. A more convenient way to bound the probability $\sum_{h \in [H]} \Pr[x_h \in \mathcal{S}_h^\ell \mid \pi]$ for a state combination $\{\mathcal{S}_h^\ell\}_{h=1}^H$ is through the following *reachability function*. In particular, the function allows us to define the probability of reaching certain states *conditioned on* the state reached at the $h$-th step following some policy $\pi$. We remark that the function can be alternatively thought as the value function (with respect to the policy $\pi$) of a modified MDP where a unit reward is given if and only if the agent lands in a predefined state combination $\{\mathcal{I}_h\}_{h=1}^H$.

**Definition E.2** (Reachability Function). Given a state combination $\{\mathcal{I}_h\}_{h=1}^H$ and some policy $\pi$, we define the reachability function as follows:

$$R_{H+1}^\pi\left(x; \{\mathcal{I}_h\}_{h=1}^H\right) = 0,$$
$$\forall h \in [H] : R_h^\pi(x; \{\mathcal{I}_h\}_{h=1}^H) = \mathbb{1}\{x \in \mathcal{I}_h\} + \mathbb{E}_{x' \sim \mathbf{p}_h(x, \pi_h(x))}\left[R_{h+1}^\pi(x'; \{\mathcal{I}_h\}_{h=1}^H)\right].$$

Recall we assume the MDP always starts at some fixed initial state $x_{\mathrm{ini}} \in \mathcal{S}$. It is not hard to show that $\sum_{h \in [H]} \Pr[x_h \in \mathcal{I}_h \mid \pi]$ equals $R_1^\pi(x_{\mathrm{ini}}; \{\mathcal{I}_h\}_{h=1}^H)$.

**Lemma E.3** (State Combination Reachability). *Let $\{\mathcal{I}_h\}_{h=1}^H$ be a state combination and $\pi$ be some policy. It holds that $\sum_{h \in [H]} \Pr[x_h \in \mathcal{I}_h \mid \pi] = R_1^\pi(x_{\mathrm{ini}}; \{\mathcal{I}_h\}_{h=1}^H)$.*

*Proof.* We will show the lemma via induction on the time step. Denote by $X_t^\pi$ the distribution of the state the agent is in in the $t$-th step under policy $\pi$. The inductive hypothesis is then

$$\sum_{h=t}^H \Pr[x_h \in \mathcal{I}_h \mid \pi] = \mathbb{E}_{s \sim X_t^\pi}\left[R_t^\pi\left(s; \{\mathcal{I}_h\}_{h=1}^H\right)\right], \tag{23}$$

and we induct on $t$. We first verify the base case when $t = H$. Then we have

$$\Pr[x_H \in \mathcal{I}_H \mid \pi] = \mathbb{E}_{s \sim X_H^\pi}\left[\mathbb{1}\{s \in \mathcal{I}_H\}\right] = \mathbb{E}_{s \sim X_H^\pi}\left[R_H^\pi(s; \{\mathcal{I}_h\}_{h=1}^H)\right].$$

Assume Equation (23) holds for all $t + 1$. We will show that Equation (23) holds for $t$ as well. In particular, applying the inductive hypothesis for $t = t' + 1$ and the definition of the reachability function gives that

$$\sum_{h=t}^{H} \Pr[x_h \in \mathcal{I}_h \mid \pi] = \mathbb{E}_{s \sim X_t^\pi} \left[ \mathbb{1}\{s \in \mathcal{I}_t\} \right] + \mathbb{E}_{s \sim X_{t+1}^\pi} \left[ R_{t+1}^\pi \left( s; \{\mathcal{I}_h\}_{h=1}^H \right) \right].$$

Note that the distribution of $X_{t+1}^\pi$ is the same as the distribution of $s \sim \mathbf{p}_t(s, \pi_{t'}(s))$, where $s \sim X_t^\pi$. Thus, we further have that

$$\sum_{h=t}^{H} \Pr[x_h \in \mathcal{I}_h \mid \pi] = \mathbb{E}_{s \sim X_t^\pi} \left[ \mathbb{1}\{s \in \mathcal{I}_t\} + \mathbb{E}_{s' \sim \mathbf{p}_t(s, \pi_t(s))} \left[ R_{t+1}^\pi(s'; \{\mathcal{I}_h\}_{h=1}^H) \right] \right]$$

$$= \mathbb{E}_{s \sim X_t^\pi} \left[ R_t^\pi(s; \{\mathcal{I}_h\}_{h=1}^H) \right].$$

This concludes the inductive step. Applying Equation (23) with $t = 1$ gives that

$$\sum_{h=1}^{H} \Pr[x_h \in \mathcal{I}_h \mid \pi] = \mathbb{E}_{s \sim X_1^\pi} \left[ R_1^\pi \left( s; \{\mathcal{I}_h\}_{h=1}^H \right) \right].$$

Since we assume that the MDP always starts at a fixed state $x_{\text{ini}}$, we know that $X_1^\pi$ is supported solely on $x_{\text{ini}}$. This concludes the proof of Lemma E.3. $\qquad\square$

### E.1. Efficient Non-replicable Exploration

Our first step is to design a *non-replicable* procedure that explores as many state-action pairs as possible, i.e., collecting data from the corresponding reward/transition distributions, until the remaining under-explored state-action pairs have low reachability under *any* policy. If so, we are certain that the policy obtained will not be significantly sub-optimal even if we ignore the under-explored state-action pairs in the learning process. Our main tool used for such strategic exploration is a variant of the $Q$-learning algorithm from (Jin et al., 2018), included below for completeness.

---

**Algorithm 3** QAGENT

**Input:** Episodic Access to $\mathcal{M}$, failure probability $\iota$, total number of episodes $K$.
**Output:** Record of all interactions.

1. Initialize $Q_h(x, a) \leftarrow H$, $V_h(x) \leftarrow H$, and $N_h(x, a) \leftarrow 0$ for all $(x, a, h) \in \mathcal{S} \times \mathcal{A} \times [H]$.

2. For convenience, set $V_{H+1}(x) \leftarrow 0$ for all $x \in \mathcal{S}$.

3. For episode $k = 1 \cdots K$:

   (a) Receive $x_1$.
   (b) For step $h = 1 \cdots H$:
      i. Take action $a_h \leftarrow \text{argmax}_{a'} Q_h(x_h, a')$, and observe $x_{h+1}$.
      ii. Set $t = N_h(x_h, a_h) \leftarrow N_h(x_h, a_h) + 1$, $b_t \leftarrow c\sqrt{H^3 \log(SAKH)/t}$ for some appropriate constant $c$, $\alpha_t \leftarrow \frac{H+1}{H+t}$.
      iii. $Q_h(x_h, a_h) \leftarrow (1 - \alpha_t) Q_h(x_h, a_h) + \alpha_t (r_h(x_h, a_h) + V_{h+1}(x_{h+1}) + b_t)$
      iv. $V_h(x_h) \leftarrow \min (H, \max_{a'} Q_h(x_h, a'))$.

---

At a high-level, QAGENT maintains an estimate of the $Q^*$-value of each state-action pair. To ensure that the agent is incentivized to pick under-explored state-action pairs, the $Q^*$-value estimates for all state-action pairs are initialized to $H$, and an optimistic bonus term (inversely proportional to the visitation count) is added to every state-action pair. In every episode, the agent interacts with the environment by greedily picking the action with the maximum $Q^*$-value estimates computed from the data collected from previous episodes.

The key technical lemma is to show that if we run the $Q$-learning agent from (Jin et al., 2018) for sufficiently many episodes, then the collection of states whose actions are not fully explored by the learning agent must have low reachability under any policy. Formally, we will use the following definition of under-explored states.

**Definition E.4** (Under-explored States). Let $\mathcal{A}_{rl}$ be a learning agent in the episodic MDP setting. Let $\mathcal{P}_h^k$ be the set of state-action pairs that are not visited by the $Q$-learning agent at step $h$ for more than $H$ times until the beginning of episode $k$, and $\mathcal{U}_h^k := \{s \in \mathcal{S} \mid \exists a \in \mathcal{A} \text{ such that } (s,a) \in \mathcal{P}_h^k\}$. We say $\{\mathcal{U}_h^k\}_{h=1}^H$ is the under-explored state combination of $\mathcal{A}_{rl}$ by the end of episode $k$.

A subtle technical detail is that, for our purpose, we need to explicitly set the rewards of the MDP to be deterministically $0$ while running the $Q$-learning agent so that the $Q$-learning agent ignores all the reward signals. At a high-level, this modification prevents the agent from exploiting actions that lead to high rewards and force the agent to continuously explore new states whenever possible. The formal statement of the exploration guarantee is given below.

**Lemma E.5** (Optimistic Exploration). *Let $\lambda, \iota \in (0,1)$. Suppose we run the $Q$-learning agent* **QAGENT** *of ([Jin et al., 2018](#)) with failure probability $\iota$ for $K$ episodes while setting all the rewards received to $0$. Let $\{\mathcal{I}_h\}_{h=1}^H$ be the under-explored state combination of* **QAGENT** *by the end of episode $K$. Assume that $K \gg SAH^5 \log(SAH/\iota)\lambda^{-2}$. With probability at least $1 - \iota$, it holds that $R_1^\pi(x_{\mathrm{ini}}; \{\mathcal{I}_h\}_{h=1}^H) \leq \lambda$ for any policy $\pi$.*

The lemma instantiates the following wishful thinking. If $s$ is still an under-explored state after the learning phase ends, the $Q$-learning algorithm will not be able to rule out the possibility that one could collect large rewards from the state. However, the final policy produced by the $Q$-learning algorithm is provably optimal. Therefore, the only possibility is that $s$ must have low reachability under any policy that will offset its potential rewards.

The proof of Lemma E.5 makes critical use of **QAGENT**'s 'optimistic estimates' $V_h(s)$, roughly a quantity estimating the optimal $V^*$-values maintained by the learning algorithm that guides its interactions with the MDP. ([Jin et al., 2018](#)) show that conditioned on certain high probability events, these estimates always bound from above the optimal $V_h^*(s)$ (hence the term 'optimistic'), and converge to $V_h^*(s)$ as the algorithm consumes the data from more episodes. The key argument is to show that: as these estimates $V_h(s)$ and the set of under-explored states $\{\mathcal{I}\}_{h=1}^H$ evolve throughout the execution of the learning algorithm, the estimates $V_h(s)$ will stay larger than the reachability function $R_h^\pi(s; \{\mathcal{I}_h^k\}_{h=1}^H)$ with high probability as well. The formal statement is given below.

**Lemma E.6** (Upper Confidence Bound on Reachability). *Suppose we run the $Q$-learning agent* **QAGENT** *of ([Jin et al., 2018](#)) with failure probability $\iota$ for $K$ episodes while setting all the rewards received to $0$. For $k \in [K], x \in \mathcal{S}, a \in \mathcal{A}, h \in [H]$, denote by $Q_h^k(x,a)$ the estimate maintained by the $Q$-learning agent from ([Jin et al., 2018](#)) at the beginning of episode $k$, and $\{\mathcal{U}_h^k\}_{h=1}^H$ the under-explored state combination of* **QAGENT** *by the end of episode $k$. With probability at least $1 - \iota$, it holds that*

$$\max_a Q_h^k(s,a) \geq R_h^\pi(s; \{\mathcal{U}_h^k\}_{h=1}^H) \tag{24}$$

*for all $h \in [H], s \in \mathcal{S}, k \in [K]$, and any policy $\pi$ at the same time.*

Since its proof is quite technical but conceptually similar to the argument from ([Jin et al., 2018](#)) showing that the estimate must bound from above the corresponding $V^*$-value, we defer it to Appendix E.4. Given Lemma E.6, the rest of the proof of Lemma E.5 boils down to bounding $V_1^k(x_{\mathrm{ini}})$ (see Appendix E.4 for formal definitions and notation) from above. Denote by $\pi_k$ the policy used by **QAGENT** to interact with the environment in the $k$-th episode, and $V_1^{\pi_k}(x_{\mathrm{ini}})$ the expected reward the agent can collect. The guarantees from ([Jin et al., 2018](#)) state that the difference $|V_1^k(x_{\mathrm{ini}}) - V_1^{\pi_k}(x_{\mathrm{ini}})|$ must decrease as $k$ increases. However, since we explicitly set the rewards of the MDP to $0$, $V_1^{\pi_k}(x_{\mathrm{ini}})$ will simply be $0$. This demonstrates that $V_1^k(x_{\mathrm{ini}})$ must be sufficiently small for large $k$.

*Proof of Lemma E.5.* Let $\{\mathcal{U}_h^k\}_{h=1}^H$ be the under-explored state combination of the **QAGENT** at the beginning of episode $k$. By Lemma E.6, it holds that

$$R_1^\pi\left(x_{\mathrm{ini}}; \{\mathcal{U}_h^k\}_{h=1}^H\right) \leq V_1^k(x_{\mathrm{ini}}) \tag{25}$$

for any policy $\pi$ and $k \in [K]$ with probability at least $1 - \iota$.

We will show that the estimate $V_1^{\tilde{k}}(x_{\mathrm{ini}})$ cannot be too large for some $\tilde{k} \in [K]$. Let $V_h^{\pi_k}$ be the value function of the policy followed by the agent in the $k$-th iteration. Since we set the rewards of the MDP to be $0$, we immediately have that $V_1^{\pi_k}(x_{\mathrm{ini}}) = 0$ regardless of the choice of $\pi_k$. Denote by $\lambda_h^{(k)} := \left(V_h^k - V_h^{\pi_k}\right)(x_h^k)$. By the proof of Theorem 2 in ([Jin et al., 2018](#)), it holds that

$$\sum_{k=1}^K \lambda_1^{(k)} \leq O\left(H^2 SA + \sqrt{H^4 SAHK \log(SAHK/\iota)}\right)$$

with probability at least $1 - \iota$. Conditioned on that, there must exist some $\tilde{k} \in [K]$ such that

$$\lambda_1^{(\tilde{k})} \le O\left( \frac{H^2 SA}{K} + \sqrt{\frac{H^5 SA \log(SAHK/\iota)}{K}} \right) \le \lambda,$$

where the second inequality is true as long as $K \gg SAH^5 \log(SAH/\iota)\lambda^{-2}$. Since $V_h^{\tilde{k}}(x_{\text{ini}}) - V_h^{\pi_{\tilde{k}}}(x_{\text{ini}}) = \lambda_1^{(\tilde{k})}$ and $V_h^{\pi_{\tilde{k}}} = 0$, it follows that $V_1^{\tilde{k}}(x_{\text{ini}}) \le \lambda$. Combining this with Equation (25) then yields that

$$R_1^\pi \left( x_{\text{ini}}; \{\mathcal{U}_h^{\tilde{k}}\}_{h=1}^H \right) \le V_1^{\tilde{k}}(x_{\text{ini}}) \le \lambda.$$

Lemma E.5 then follows from the fact that the under-explored state combination can only shrink as the algorithm interacts with the environment for more episodes.

$\square$

If we set the accuracy parameter $\lambda$ of $Q$-learning agent from (Jin et al., 2018) to $\varepsilon/2$, Lemma E.5 guarantees that the set of under-explored states $\mathcal{I}_h$ can be safely ignored as their contribution to the total value of the MDP is at most $\varepsilon/2$. For the rest of the states, we wish to generate enough samples so that we can feed them to the bandit algorithm to learn the optimal policy. By the definition of the under-explored state combination $\{\mathcal{I}_h\}_{h=1}^H$, it is clear that for all $x, a, h$ such that $x \notin \mathcal{I}_h$ we have collected at least $H$ samples from the transition $\mathbf{p}_h(x, a)$. Denote by $\mathcal{D}_{x,a,h}$ the dataset for the samples collected from $\mathbf{p}_h(x, a)$. A technical issue is that $D_{x,a,h}$ and $D_{x',a',h+1}$ are not necessarily independent of each other as there are complicated statistical dependencies between the transition at step $h$ and the number of times we have seen a certain state $x'$ at step $h + 1$. We handle this issue by running **QAGENT** on a simulated MDP with $A$ additional "phantom" actions used specifically for sample collection.

---

**Algorithm 4** QEXPLORE

**Input:** Access to the MDP and the the $Q$-learning agent **QAGENT** from (Jin et al., 2018).
**Output:** State combination $\{\mathcal{I}_h\}_{h=1}^H$ and datasets $\{\mathcal{D}_{s,a,h}\}_{s\in\mathcal{S},a\in\mathcal{A},h\in[H]}$.

1. Initialize empty datasets $\mathcal{D}_{s,a,h}$ for $s \in \mathcal{S}, a \in \mathcal{A}, h \in [H]$.

2. For episode $k = 1 \cdots K$:

    (a) While the MDP has not terminated:

        i. Denote by $\bar{a}$ the corresponding phantom action of $a \in \mathcal{A}$. Present **QAGENT** the actions $\mathcal{A} \cup \{\bar{a} : a \in \mathcal{A}\}$.

        ii. If **QAGENT** chooses $a \in \mathcal{A}$:

            A. Interact with the real MDP by taking action $a$. Receive $s_{\text{nxt}}$ and reward $r$.

            B. Inform **QAGENT** the next state is $s_{\text{nxt}}$ and the reward is $0$.

        iii. If **QAGENT** chooses a phantom action $\bar{a}$:

            A. Interact with the real MDP by taking action $a$. Receive $s_{\text{nxt}}$ and reward $r$.

            B. Add $(s_{\text{nxt}}, r)$ to the corresponding $\mathcal{D}_{s,a,h}$.

            C. Inform **QAGENT** the next state is the terminal state and the reward is $0$.

            D. Terminate the current MDP episode.

3. For each $(s, h) \in \mathcal{S} \times [H]$, add $s$ to $\mathcal{I}_h$ if there exists $a \in \mathcal{A}$ such that $|\mathcal{D}_{s,a,h}| < H$.

4. Output $\{\mathcal{I}_h\}_{h=1}^H$ and $\{\mathcal{D}_{s,a,h}\}$.

---

In particular, when the learning algorithm chooses to take a phantom action, the simulator takes the corresponding real action, records the transition and reward seen to our dataset, but hides the transition and reward information from the algorithm, and instead instructs the agent that it has received a reward of $0$ and the simulation has terminated. That way, it is guaranteed that the data collected from the phantom state-action pairs must be independent as an agent can take at most $1$ phantom action per episode. Moreover, from the perspective of the learning agent, it is just interacting with another MDP who has (1) an extra terminal state and (2) has twice as many actions as the original MDP. If we let the agent **QAGENT**

interact with the simulated MDP, for any tuple $s, a, h$ such that $s$ is not under explored at step $h$, **QAGENT** must have visited the pair $(s, a')$, where $a'$ is the phantom action of $a$, at step $h$ for at least $H$ times. This therefore gives an independent dataset $\mathcal{D}_{s,a,h}$ that has at least $H$ samples from the reward/transition distributions.

**Corollary E.7** (Decouple Data Dependency with Phantom Actions). *Given $\iota, \lambda \in (0, 1)$, there is an algorithm **QEXPLORE** that consumes at most $K := O(SAH^5 \log(SAH/\iota)\lambda^{-2})$ episodes, runs in time $\mathrm{poly}\,(K)$, and produces a state combination $\{\mathcal{I}_h\}_{h=1}^H$ such that (i) $R_1^\pi(x_{\mathrm{ini}}; \{\mathcal{I}_h\}_h) \leq \lambda$ for any policy $\pi$ with probability at least $1 - \iota$, and (ii) for each $s \in \mathcal{S}, a \in \mathcal{A}, h \in [H]$ satisfying $x \notin \mathcal{I}_h$, the algorithm produces a dataset $\mathcal{D}_{x,a,h}$ made up of at least $H$ samples of $(s, r) \sim \mathbf{p}_h(x, a) \otimes \mathbf{r}_h(x, a)$. Moreover, conditioning on the sizes of each dataset $n_{x,a,h} := |\mathcal{D}_{x,a,h}|$, the resulting conditional joint distribution over the datasets is exactly the product distribution $\prod_{(x,a,h)} (p_h(x, a) \otimes r_h(x, a))^{n_{x,a,h}}$.*

### E.2. Replicably Finding Ignorable States and Sample Collection

Corollary E.7 gives an algorithm that explores the states in a non-replicable manner. To turn it into a replicable algorithm, we will consider the distribution of the under-explored state combination $\{\mathcal{I}_h\}_{h=1}^H$ produced by **QEXPLORE**. Running the algorithm many times, we can expect its "average output" to concentrate around its mean, which we view as a 'fractional' ignorable state combination. We then exploit the concentration of this output to perform a correlated rounding procedure of the fractional solution to an integer solution, ensuring replicability in the process.

**Definition E.8** (Distribution of Under-Explored State Combination). Let $\{\mathcal{I}_h^{\lambda,\iota}\}_{h=1}^H$ be the random state combination produced by **QEXPLORE** with parameter $\lambda, \iota \in (0, 1)$ specified as in Corollary E.7, where the randomness is over the internal randomness of the MDP. For $h \in [H]$, we define $D_{\mathcal{I}_h}^{\lambda,\iota}$ to be the distribution over the vector $u$ indexed by the states $\mathcal{S}$ and $[H]$ such that $u_{s,h} = \mathbb{1}\{s \in \mathcal{I}_h^{\lambda,\iota}\}$. Moreover, we write its expectation as $\mu_{s,h}^{(\lambda,\iota)} := \mathbb{E}_{u \sim D_{\mathcal{I}_h}^{\lambda,\iota}}[u_{s,h}]$.

By running **QEXPLORE** for multiple times, we can effectively sample from $D_{\mathcal{I}_h}^{\lambda,\iota}$ to compute an estimate of $\mu_{s,h}^{\lambda,\iota}$. We round $\mu^{\lambda,\iota}$ back to an integer solution in the natural fashion, simply by including each $(s, h)$ pair independently with probability proportional to $\hat{\mu}_{(s,h)}^{\lambda,\iota}$.

**Definition E.9** (Product State Combination Distribution). Let $\{\hat{\mu}_{s,h}\}_{s \in \mathcal{S}, h \in [H]} \subset [0, 1]$. Consider a state combination $\{\hat{\mathcal{I}}_h\}_{h=1}^H$ sampled as follows: for each state $s \in \mathcal{S}$ and step $h \in [H]$, we add $s$ to $\hat{\mathcal{I}}_h$ with probability $\hat{\mu}_{s,h}$ independently. We define $\mathcal{B}(\hat{\mu})$ to be the distribution of such $\{\hat{\mathcal{I}}_h\}_{h=1}^H$.

Given that $\hat{\mu}_{s,h}$ is an accurate enough estimate of $\mu_{s,h}^{\lambda,\iota}$ (cf. Definition E.8), though the distribution $\mathcal{B}(\hat{\mu})$ can still be far from the original under-explored state combination distribution $\mathcal{D}_I^{\lambda,\iota}$, we show that sampling according to $\mathcal{B}(\hat{\mu})$ nonetheless yields a state combination with small reachability as well with high probability. The proof largely follows from the fact that the reachability function is linear with respect to the indicator vector of the input state combination, and thus it is enough that $B(\hat{\mu})$ essentially has the right marginal probabilities.

**Lemma E.10** (Sampling Preserves Reachability). *Let $\lambda, \iota, \kappa \in (0, 1)$, and $m \in \mathbb{Z}_+$. Let $\{\hat{\mu}_{s,h}\}_{s \in \mathcal{S}, h \in [H]}$ be the empirical estimates of $\{\mu_{s,h}^{\lambda,\iota}\}_{s \in \mathcal{S}, h \in [H]}$ computed from $m$ samples from $D_{\mathcal{I}}^{\lambda,\iota}$, and $\{\mathcal{I}_h\}_{h=1}^H$ be a state combination sampled from $\mathcal{B}(\hat{\mu})$. Let $\pi$ be some arbitrary policy. Then it holds that*

$$R_1^\pi(x_{\mathrm{ini}}; \{\mathcal{I}_h\}_{h=1}^H) \leq \kappa^{-1}\lambda$$

*with probability at least $1 - m\iota - \kappa$.*

*Proof.* Let $\mathcal{E}$ be the empirical distribution over $m$ state combinations computed from the **QEXPLORE** algorithm from Corollary E.7 with accuracy $\lambda$ and failure probability $\iota$ as inputs, and $\hat{\mu}$ be the estimates

$$\hat{\mu}_{s,h} = \mathbb{E}_{\{\mathcal{I}_h\}_{h=1}^H \sim \mathcal{E}}\left[\mathbb{1}\{s \in \mathcal{I}_h\}\right].$$

Fix $\pi$ to be an arbitrary policy. By the union bound, the algorithm succeeds in all $m$ runs with probability at least $1 - m\iota$. Conditioned on that, by Corollary E.7, it follows that

$$\mathbb{E}_{\{\mathcal{I}_h\}_{h=1}^H \sim \mathcal{E}}\left[R_1^\pi(x_{\mathrm{ini}}; \{\mathcal{I}_h\}_{h=1}^H)\right] \leq \lambda. \tag{26}$$

We claim that

$$\mathbb{E}_{\{\mathcal{I}_h\}_{h=1}^H \sim \mathcal{E}} \left[ R_t^\pi(x; \{\mathcal{I}_h\}_{h=1}^H) \right] = \mathbb{E}_{\{\mathcal{I}_h\}_{h=1}^H \sim \mathcal{B}(\hat\mu)} \left[ R_t^\pi(x; \{\mathcal{I}_h\}_{h=1}^H) \right] \tag{27}$$

for all $x \in \mathcal{S}$ and $t \in [H]$.

We will show Equation (27) via induction on $t$. In the base case, we have $t = H + 1$, and the equality is true by definition of $R_{H+1}^\pi$. Suppose that Equation (27) holds for all $t' \geq t + 1$ and $x \in \mathcal{S}$. By the definition of $R_t^\pi$ and linearity of expectation, we have that

$$
\begin{aligned}
&\mathbb{E}_{\{\mathcal{I}_h\}_{h=1}^H \sim \mathcal{E}} \left[ R_t^\pi(x; \{\mathcal{I}_h\}_{h=1}^H) \right] \\
&= \mathbb{E}_{\{\mathcal{I}_h\}_{h=1}^H \sim \mathcal{E}}[\mathbb{1}\{x \in \mathcal{I}_h\}] + \mathbb{E}_{\{\mathcal{I}_h\}_{h=1}^H \sim \mathcal{E}} \left[ \mathbb{E}_{x' \sim \mathbf{p}_t(x, \pi_t(x))} \left[ R_{t+1}^\pi(x'; \{\mathcal{I}_h\}_{h=1}^H) \right] \right] \\
&= \mathbb{E}_{\{\mathcal{I}_h\}_{h=1}^H \sim \mathcal{B}(\hat\mu)}[\mathbb{1}\{x \in \mathcal{I}_h\}] + \mathbb{E}_{\{\mathcal{I}_h\}_{h=1}^H \sim \mathcal{E}} \left[ \mathbb{E}_{x' \sim \mathbf{p}_t(x, \pi_t(x))} \left[ R_{t+1}^\pi(x'; \{\mathcal{I}_h\}_{h=1}^H) \right] \right] \\
&= \mathbb{E}_{\{\mathcal{I}_h\}_{h=1}^H \sim \mathcal{B}(\hat\mu)}[\mathbb{1}\{x \in \mathcal{I}_h\}] + \mathbb{E}_{\{\mathcal{I}_h\}_{h=1}^H \sim \mathcal{B}(\hat\mu)} \left[ \mathbb{E}_{x' \sim \mathbf{p}_t(x, \pi_t(x))} \left[ R_{t+1}^\pi(x'; \{\mathcal{I}_h\}_{h=1}^H) \right] \right] \\
&= \mathbb{E}_{\{\mathcal{I}_h\}_{h=1}^H \sim \mathcal{B}(\hat\mu)}[R_t^\pi(x; \{\mathcal{I}_h\}_{h=1}^H)],
\end{aligned}
$$

where the first equality uses the definition of $R_t^\pi$ and linearity of expectation, the second equality uses the fact that $\mathbb{E}_{\{\mathcal{I}_h\}_{h=1}^H \sim \mathcal{B}(\hat\mu)}[\mathbb{1}\{x \in \mathcal{I}_h\}] = \hat\mu_{x,h} = \mathbb{E}_{\{\mathcal{I}_h\}_{h=1}^H \sim \mathcal{E}}[\mathbb{1}\{x \in \mathcal{I}_h\}]$, the third equality uses the inductive hypothesis, and the last equality again uses the definition of $R_t^\pi$. This shows Equation (27).

Combining Equations (26) and (27) then gives that

$$\mathbb{E}_{\{\mathcal{I}_h\}_{h=1}^H \sim \mathcal{B}(\hat\mu)} \left[ R_1^\pi(x_{\mathrm{ini}}; \{\mathcal{I}_h\}_{h=1}^H) \right] \leq \lambda.$$

It then follows from Markov's inequality that $R_1^\pi(x_{\mathrm{ini}}; \{\mathcal{I}_h\}_{h=1}^H) \leq \lambda\kappa^{-1}$ with probability at least $1-\kappa$ for $\{\mathcal{I}_h\}_{h=1}^H \sim \mathcal{B}(\hat\mu)$. This concludes the proof of Lemma E.10. □

Since sampling according to $\mathcal{B}(\hat\mu)$ preserves the reachability function value, we can use correlated sampling to obtain a *replicable* state combination with low reachability. Consider the binary product distribution $\mathcal{B}(\mu^{\lambda,\iota})$. If we take roughly $SH/\kappa^2$ many samples, we can compute some $\hat\mu$ such that the total variation distance between $\mathcal{B}(\mu^{\lambda,\iota})$ and $\mathcal{B}(\hat\mu)$ is at most $\kappa$.

**Lemma E.11** (Total Variation Bound). *Let $\lambda, \iota, \kappa \in (0,1)$. Let $\hat\mu_{s,h}$ be estimates of $\mu_{s,h}^{\lambda,\iota}$ computed from $m$ samples from $D_{\mathcal{I}}^{\lambda,\iota}$. Assume that $m \gg \log(SH/\kappa)SH\kappa^{-2}$. Then with probability at least $1 - \kappa$ it holds that $d_{\mathrm{TV}}\left(\mathcal{B}(\hat\mu), \mathcal{B}(\mu^{\lambda,\iota})\right) \leq \kappa$.*

*Proof.* By the Chernoff bound, we have that

$$\left(\hat\mu_{s,h} - \mu_{s,h}^{\lambda,\iota}\right)^2 \leq \frac{\log(SH/\kappa)\mu_{s,h}^{\lambda,\iota}\left(1 - \mu_{s,h}^{\lambda,\iota}\right)}{m} \tag{28}$$

with probability at least $1 - \kappa/SH$. By the union bound, the above holds for all $s \in \mathcal{S}, h \in [H]$ simultaneously with probability at least $1 - \kappa$. We will condition on the above event throughout the analysis.

Assume that $m \gg \log(SH/\kappa)SH\kappa^{-2}$. We proceed to bound from above the $\chi^2$-divergence between $\mathcal{B}(\hat\mu)$ and $\mathcal{B}(\mu^{\lambda,\iota})$. In particular, we have that

$$
\begin{aligned}
d_{\mathrm{TV}}^2\left(\mathcal{B}(\hat\mu), \mathcal{B}(\mu^{\lambda,\iota})\right) &\leq \sum_{s,h:0<\mu_{s,h}^{\lambda,\iota}} \frac{\left(\hat\mu_{s,h} - \mu_{s,h}^{\lambda,\iota}\right)^2}{\mu_{s,h}^{\lambda,\iota}} + \sum_{s,h:\mu_{s,h}^{\lambda,\iota}<1} \frac{\left(\hat\mu_{s,h} - \mu_{s,h}^{\lambda,\iota}\right)^2}{1 - \mu_{s,h}^{\lambda,\iota}} \\
&\leq O(1) \cdot \sum_{s \in \mathcal{S}, h \in [H]} \frac{\log(SH/\kappa)}{m} \leq \kappa^{-2},
\end{aligned}
$$

where the first inequality follows from Lemma B.11, the second inequality follows from Equation (28), and the last inequality follows from our assumption that $m \gg \log(SH/\kappa)SH\kappa^{-2}$. Taking the square roots of both sides concludes the proof of Lemma E.11. □

Given a state combination $\{\hat{\mathcal{I}}_h\}_{h=1}^H$ sampled from $\mathcal{B}(\hat{\mu})$, one last thing that needs to be addressed is that we can provide a large enough independent dataset $\mathcal{D}_{s,a,h}$ for every $(s,a,h)$ such that $s \notin \hat{\mathcal{I}}_h$.

**Lemma E.12** (Efficient Sample Collection). *Let $\lambda, \iota, \kappa \in (0,1)$, and $\{\hat{\mu}_{s,h}\}_{s\in\mathcal{S},h\in[H]}$ be estimates of $\mu_{s,h}^{\lambda,\iota}$ computed from $m$ samples. Suppose we have run the algorithm from Corollary E.7 for another $M$ times with accuracy $\lambda$ and failure probability $\iota$. Let $\mathcal{D}_{s,a,h}$ be the combined dataset from these $M$ runs, and $\{\hat{\mathcal{I}}_h\}_{h=1}^H$ be a state combination sampled from $\mathcal{B}(\hat{\mu})$. Assume that $M \gg \log^2(SH/\kappa)m$ and $M\iota \ll \kappa$. Then there exists some number $m_{s,h}$ such that the following hold with probability at least $1 - \kappa$*

  *(i) Lower bounded support:*
$$\min_a |\mathcal{D}_{s,a,h}| \geq m_{s,h}$$

  *(ii) Niceness of $\{m_{s,h}\}$:*
$$\sum_{h\in[H]} \sqrt{\sum_{s:s\notin\hat{\mathcal{I}}_h} \frac{1}{m_{s,h}}} \leq O(1)\kappa^{-1}\sqrt{\frac{H \cdot S \log(SH/\kappa)}{M}}. \tag{29}$$

Before we move on, it is worth recalling why we need the somewhat cumbersome guarantees above rather than simply collecting $S \cdot \mathrm{poly}(H)$ many samples for each dataset $\mathcal{D}_{s,a,h}$ as we would do in the parallel sampling model. This comes from the fact that some states may be inherently more difficult to collect samples from. For instance, there may be states such that $1 - \mu_{s,h}^{\lambda,\iota} = \Theta(1/S)$. Since we run **QEXPLORE** $S$ times, these states may actually appear outside the final ignorable state combination, but to collect $S\mathrm{poly}(H)$ many samples from them would require running **QEXPLORE** for at least $\Theta(S^2) \cdot \mathrm{poly}(H)$ many times (resulting in $\Omega(S^3)$ total episodes).

To overcome the hurdle, we set the sample lower bound $m_{s,h}$ in Lemma E.12 to be roughly $MH\left(1 - \mu_{s,h}^{\lambda,\iota}\right)$. A simple concentration argument shows that we can gather at least $m_{s,h} = \tilde{\Omega}\left(MH\left(1 - \mu_{s,h}^{\lambda,\iota}\right)\right)$ many samples from the pair $(s,h)$ using the **QEXPLORE** algorithm. On the other hand, the likelihood of having $s \notin \hat{\mathcal{I}}_h$ is always roughly proportional to $\left(1 - \mu_{s,h}^{\lambda,\iota}\right)$. This ensures that the expected value of $\mathbb{1}\{s \notin \hat{\mathcal{I}}_h\}/m_{s,h}$ is roughly the same for all pairs $(s,h)$, making it handy to bound the left hand side of Equation (29) from above. It is not hard to see that setting $M = \tilde{\Theta}_{\rho,\kappa,\varepsilon,H}(S)$ will meet the requirement of the replicable best arm algorithm (cf. Proposition C.6), leading to optimal dependency on $S$.

*Remark* E.13. Note the numbers $m_{s,h}$, while unknown to the algorithm, do not depend on the samples collected. Thus, we may assume $\{m_{s,h}\}$ are the same set of numbers across two different runs of the algorithm. This is crucial in ensuring replicability of the downstream application of the best arm algorithm.

*Proof.* By the Chernoff bound, we have that
$$\left(\hat{\mu}_{s,h} - \mu_{s,h}^{\lambda,\iota}\right)^2 \leq \log(2SH/\kappa)\mu_{s,h}^{\lambda,\iota}\left(1 - \mu_{s,h}^{\lambda,\iota}\right)/m, \tag{30}$$
$$\left(\min_a |\mathcal{D}_{s,a,h}|/H - M\left(1 - \mu_{s,h}^{\lambda,\iota}\right)\right)^2 \leq \log(2SH/\kappa)\left(1 - \mu_{s,h}^{\lambda,\iota}\right)M, \tag{31}$$

with probability at least $1 - \kappa/(SH)$. By the union bound, the above holds for all $s,h$ simultaneously with probability at least $1 - \kappa$. We will condition on the above event throughout the analysis.

The numbers $m_{s,h}$ will be set differently depending on the relative size of $\mu_{s,h}^{\lambda,\iota}$ and $1/(10m\log(SH/\kappa))$. In particular define:
$$m_{s,h} := \begin{cases} MH\left(1 - \mu_{s,h}^{\lambda,\iota}\right)/2 & \text{if } \left(1 - \mu_{s,h}^{\lambda,\iota}\right) > 1/(10m\log(SH/\kappa)) \\ 0 & \text{otherwise} \end{cases}$$

We first show property (i). In the latter case above, we have that $m_{s,h} = 0$ so $\min_a |\mathcal{D}_{s,a,h}| \geq m_{s,h}$ is satisfied trivially. In the former case, note that $M(1 - \mu_{s,h}^{\lambda,\iota}) \geq \Omega(M/(m\log(SH/\kappa)))$. Since we assume that $M \gg \log^2(SH/\rho)m$, it follows that $M(1 - \mu_{s,h}^{\lambda,\iota}) \gg \log(SH/\kappa)$, which implies
$$M(1 - \mu_{s,h}^{\lambda,\iota}) \gg \sqrt{\log(SH/\kappa) \, M(1 - \mu_{s,h}^{\lambda,\iota})}.$$

Combining the above with Equation (31) then gives that

$$\min_a |\mathcal{D}_{s,a,h}| \geq \frac{1}{2} MH \left(1 - \mu_{s,h}^{\lambda,\iota}\right) := m_{s,h},$$

concluding the proof of property (i).

It is left to show property (ii). Since the expression will not even be finitely bounded if some $s \notin \hat{\mathcal{I}}_h$ has $m_{s,h} = 0$, our first step is to show that, assuming Equation (30), any such $s$ is in $\hat{\mathcal{I}}_h$. By our construction, $m_{s,h} = 0$ if and only if $\left(1 - \mu_{s,h}^{\lambda,\iota}\right) < 1/(10m\log(SH/\kappa))$. Combining the case assumption with Equation (30) yields that

$$(1 - \hat{\mu}_{s,h}) \leq \left(1 - \mu_{s,h}^{\lambda,\iota}\right) + \frac{1}{2m} < \frac{1}{m},$$

which implies that $m(1 - \hat{\mu}_{s,h}) < 1$. Note that $m(1 - \hat{\mu}_{s,h})$ counts the number of times that the state $s$ is included in the state combination $\{\mathcal{I}_h\}_{h=1}^H$ produced by the first $m$ runs of QEXPLORE. As a result this quantity is integral, and in particular it follows that $\hat{\mu}_{s,h} = 1$. Finally, by construction such an $s$ is included in $\hat{\mathcal{I}}_h$ with probability $\hat{\mu}_{s,h} = 1$ as desired.

On the other hand, if $\left(1 - \mu_{s,h}^{\lambda,\iota}\right) > 1/(10m\log(SH/\kappa))$, Equation (30) then implies that

$$(1 - \hat{\mu}_{s,h}) \leq \left(1 - \mu_{s,h}^{\lambda,\iota}\right) + \sqrt{\left(1 - \mu_{s,h}^{\lambda,\iota}\right) \log(2SH/\kappa)/m}$$

$$\leq \left(1 - \mu_{s,h}^{\lambda,\iota}\right) + O(1)\sqrt{\left(1 - \mu_{s,h}^{\lambda,\iota}\right)^2 \log^2(SH/\kappa)}$$

$$\leq O\left(\log(SH/\kappa)\right) \left(1 - \mu_{s,h}^{\lambda,\iota}\right).$$

Recall that we set $m_{s,h} = MH(1 - \mu_{s,h}^{\lambda,\iota})/2$. It follows that

$$\mathbb{E}\left[\frac{\mathbb{1}\{s \notin \hat{\mathcal{I}}_h\}}{m_{s,h}}\right] \leq \frac{O(1) \cdot \log(SH/\kappa)}{MH}. \tag{32}$$

Fixing some $h$ and summing over all $s$ such that $m_{s,h} > 0$ gives that

$$\mathbb{E}\left[\sum_{s:m_{s,h}>0} \frac{\mathbb{1}\{s \notin \hat{\mathcal{I}}_h\}}{m_{s,h}}\right] \leq \frac{O(1) \cdot S\log(SH/\kappa)}{MH}.$$

By Jensen's inequality, we have that

$$\mathbb{E}\left[\sqrt{\sum_{s:m_{s,h}>0} \frac{\mathbb{1}\{s \notin \hat{\mathcal{I}}_h\}}{m_{s,h}}}\right] \leq O(1)\sqrt{\frac{S\log(SH/\kappa)}{MH}}.$$

Summing over $H$ therefore gives that

$$\mathbb{E}\left[\sum_{h\in[H]} \sqrt{\sum_{s:m_{s,h}>0} \frac{\mathbb{1}\{s \notin \hat{\mathcal{I}}_h\}}{m_{s,h}}}\right] \leq O(1)H\sqrt{\frac{S\log(SH/\kappa)}{MH}}.$$

Hence, by Markov's inequality, with probability at least $1 - \kappa$ we have that

$$\sum_{h\in[H]} \sqrt{\sum_{s:m_{s,h}>0} \frac{\mathbb{1}\{s \notin \hat{\mathcal{I}}_h\}}{m_{s,h}}} \leq O(1)\kappa^{-1}\sqrt{\frac{H \cdot S\log(SH/\kappa)}{M}}.$$

Recall that we have shown that if $m_{s,h} = 0$, then $s \in \hat{\mathcal{I}}_h$ conditioned on Equation (30). Combining this observation with above then shows property (ii), and concludes the proof of Lemma E.12. $\square$

Putting things together gives us an algorithm **REPEXPLORE** that can at the same time replicably identify a state combination that has low-reachability and collect enough i.i.d. samples from the rest of states to apply our variable best-arm algorithm.

The algorithm takes as inputs $\kappa$, which controls the level of replicability, $\lambda$, which controls the level of reachability of the state combination produced, and $\beta$, which controls the number of samples one would like to collect. The algorithm is made up of the following steps.

---

**Algorithm 5 REPEXPLORE**

---

**Input:** $\kappa, \lambda, \beta \in (0, 1)$ which controls the replicability, the reachability, and the sample complexity aspects of the algorithm.
**Output:** A state combination $\{\mathcal{I}_h\}_{h=1}^H$ and datasets $\{\mathcal{D}_{s,a,h}\}_{s\in\mathcal{S}, a\in\mathcal{A}, h\in[H]}$.

1. Set the failure probability $\iota$ to be a sufficiently large polynomial in $S, A, H, \lambda^{-1}, \kappa^{-1}, \beta^{-1}$.

2. Run **QEXPLORE** with accuracy parameter $(\lambda\kappa)$ and failure probability $\iota$ for $m := \Theta(SH\log(SH/\kappa)\kappa^{-2})$ many times. Denote by $\hat{\mu}_{s,h}$ the fraction of times that $(s, h)$ is *not* fully explored by the **QEXPLORE** algorithm.

3. Use correlated sampling to sample a state combination $\{\mathcal{I}_h\}_{h=1}^H$ from $\mathcal{B}(\hat{\mu})$ (see Definition E.9)

4. Run the **QEXPLORE** algorithm with accuracy parameter $(\lambda\kappa)$ and failure probability $\iota$ for $M := \Theta\left(\log^2(SH/\kappa)m\right) + \Theta\left(SH\log(SH/\kappa)\kappa^{-2}\beta^{-2}\right)$ many times to generate an independent dataset $\mathcal{D}_{s,a,h}$ made up of i.i.d. samples from $\mathbf{p}_h(s, a) \otimes \mathbf{r}_h(s, a)$.

5. Output the state combination $\{\mathcal{I}_h\}_{h=1}^H$ and the datasets $\{\mathcal{D}_{s,a,h}\}$.

---

**Lemma E.14** (Single-Tier Replicable Exploration). *Fix an arbitrary policy $\pi$ on $\mathcal{M}$. Let $\lambda, \kappa, \beta \in (0, 1)$. There exists some positive integers $\{m_{s,h}\}_{s\in\mathcal{S}, h\in[H]}$, and an algorithm* **REPEXPLORE** *that consumes at most $K := \tilde{\Theta}(S^2AH^6\lambda^{-2}\kappa^{-4}\beta^{-2})$ episodes, runs in time $\text{poly}\,(K)\,\exp\left((1+o(1))SH\right)$, and produces a state combination $\{\mathcal{I}_h\}_{h=1}^H$, and an independent dataset $\mathcal{D}_{s,a,h}$ made up of i.i.d. samples from $\mathbf{p}_h(s, a) \otimes \mathbf{r}_h(s, a)$ for all $s \in \mathcal{S}, a \in \mathcal{A}, h \in [H]$ such that (i) the output set $\{\mathcal{I}_h\}_{h=1}^H$ is $O(\kappa)$-replicable, (ii) the following holds with probability at least $1 - O(\kappa)$:*

*(a)* $R_1^\pi(x_{\text{ini}}; \{\mathcal{I}_h\}_{h=1}^H) \leq \lambda$ ,

*(b)* $\min_a |\mathcal{D}_{s,a,h}| \geq m_{s,h}$ *and* $\sum_{h\in[H]} \sqrt{\sum_{s:s\in\mathcal{I}_h} (1/m_{s,h})} \leq \kappa\beta$.

*Moreover, the algorithm runs in time $\text{poly}\,(K)\exp\left(O(SH)\right)$.*

*Proof.* We first argue the replicability of the state combination produced. By Lemma E.11, the estimates $\{\hat{\mu}_{s,h}\}_{s\in\mathcal{S}, h\in[H]}$ are accurate enough such that
$$d_{\text{TV}}\left(\mathcal{B}\,(\hat{\mu})\,, \mathcal{B}(\mu^{\lambda\kappa,\iota})\right) \leq \kappa,$$
where $\{\mu_{s,h}\}_{s\in\mathcal{S}, h\in[H]}$ are defined in Definition E.8. If we denote by $\{\hat{\mu}_{s,h}^{(1)}\}_{s\in\mathcal{S}, h\in[H]}$, $\{\hat{\mu}_{s,h}^{(2)}\}_{s\in\mathcal{S}, h\in[H]}$ the estimates obtained in two separate runs of the **REPEXPLORE** algorithm, it follows from the triangle inequality that

$$d_{\text{TV}}\left(\mathcal{B}(\hat{\mu}^{(1)}), \mathcal{B}(\hat{\mu}^{(2)})\right) \leq 2\kappa.$$

Since the state combinations produced in the two runs are obtained from correlated sampling from $\mathcal{B}\left(\hat{\mu}^{(1)}\right)$ and $\mathcal{B}\left(\hat{\mu}^{(2)}\right)$, it follows from Lemma B.12 that the state combination $\{\mathcal{I}_h\}_{h=1}^H$ is $O(\kappa)$-replicable. This shows property (i).

We proceed to show property (ii). Recall that we run **QEXPLORE** for $m$ times with accuracy $\lambda' = (\lambda\kappa)$ and failure probability $\iota$ to obtain $\hat{\mu}_{s,h}$. Since $\iota m \ll \kappa$ , we can apply Lemma E.10, which gives that the state combination $\{\mathcal{I}_h\}_{h=1}^H \sim \mathcal{B}(\hat{\mu})$ satisfies
$$R_1^\pi(x_{\text{ini}}; \{\mathcal{I}_h\}_{h=1}^H) \leq \lambda'\kappa^{-1} = \lambda$$
with probability at least $1 - m\iota - \kappa \geq 1 - 2\kappa$. This shows that property (ii,a) holds with probability at least $1 - \kappa$.

We then move to property (ii,b). In particular, we set the numbers $m_{s,h}$ in the same way as Lemma E.12. Since $M \gg \log^2(SH/\kappa)m$ and $M\iota \ll \kappa$, Lemma E.12 is applicable and gives that $\min_a |\mathcal{D}_{s,a,h}| \geq m_{s,h}$ and

$$\sum_{h \in [H]} \sqrt{\sum_{s:s \notin \mathcal{I}_h} \frac{1}{m_{s,h}}} \leq O(1)\kappa^{-1} \sqrt{\frac{SH \log(SH/\kappa)}{M}}.$$

with probability at least $1 - \kappa$. Since $M \gg SH \log(SH/\kappa)\kappa^{-4}\beta^{-2}$, the above equation immediately implies that

$$\sum_{h \in [H]} \sqrt{\sum_{s:s \notin \mathcal{I}_h} \frac{1}{m_{s,h}}} \leq \kappa\beta.$$

This shows that property (ii,b) holds with probability at least $1 - \kappa$. Hence, by the union bound, property (ii) holds with probability at least $1 - 2\kappa$.

Lastly, we note that in total we have invoked **QEXPLORE** from Corollary E.7 for

$$m + M = O\left(SH \log^2(SH/\kappa)\kappa^{-2}\beta^{-2}\right)$$

many times with accuracy $(\lambda/\kappa)$ and failure probability $\iota$ that is the inverse of some polynomial in $S, H, \kappa, \beta, \lambda$. By Corollary E.7, one such execution of **QEXPLORE** consumes $O(SAH^5 \log(1/\iota)\kappa^{-2}\lambda^{-2})$ many episodes. Therefore, the total number of episodes consumed is at most

$$O(SAH^5 \log(1/\iota)\kappa^{-2}\lambda^{-2}) \cdot O\left(SH \log^2(SH/\kappa)\kappa^{-2}\beta^{-2}\right) \leq \tilde{O}\left(S^2 AH^6 \kappa^{-4}\lambda^{-2}\beta^{-2}\right).$$

It is not hard to see that the runtime of the algorithm is at most polynomial in the number of episodes consumed except for the correlated sampling procedure. The support size of the the the distribution $\mathcal{B}(\hat{\mu})$ is at most $2^{SH}$. Hence, by Lemma B.12, the correlated sampling procedure runs in time $\exp((1 + o(1))SH)$. This concludes the analysis of the algorithm. $\qquad\square$

### E.3. Proof of Lemma E.1: Tiered Exploration

In this subsection, we will present the algorithm **REPLEVELEXPLORE** and its analysis to conclude the proof of Lemma E.1. Ideally, we would like the state combination $\{\mathcal{I}_h\}_{h=1}^H$ to satisfy that $R_1^\pi\left(x_{\mathrm{ini}}; \{\mathcal{I}_h\}_{h=1}^H\right) \ll \varepsilon$ so that we can ignore them while using the best arm algorithm to learn an optimal policy. For the rest of states, we would like to collect enough samples from them so that the bandit algorithm can achieve $\varepsilon$ accuracy. However, to achieve the above guarantees, we would need to set both $\lambda$ and $\beta$ to $\Theta(\varepsilon)$. This will make the sample complexity have a suboptimal $\varepsilon^{-4}$ dependency.

The reason once again is that it is inherently harder to collect from certain states than others. In particular, in this case if there is a state such that no policy can reach it with probability more than $10\varepsilon$, we should not expect to collect $\varepsilon^{-2}$ many samples from it within $\varepsilon^{-2}$ episodes. On the other hand, the optimal policy also cannot reach it with probability more than $10\varepsilon$. Therefore, it suffices for us to estimate the $V^*$ values of this state with small constant accuracy, which requires significantly fewer samples.

To optimize the dependency on $\varepsilon$, we will run **REPEXPLORE** with multiple different combinations of $\lambda$ and $\beta$ such that fewer samples are collected from states that have lower reachability. This yields the algorithm **REPLEVELEXPLORE** from Lemma E.1. The algorithm takes as input a single parameter $\zeta \in (0, 1)$ that controls the number of samples we will gather for each state and will be set later roughly to $\varepsilon$ up to some poly-logarithmic factors in the final algorithm. For convenience, we assume that $\log(1/\zeta)$ is an integer.

*Proof of Lemma E.1.* We first argue the replicability of the algorithm. By Lemma E.14, for any $\ell \in [\log(1/\zeta) - 1]$, the state combination $\{\mathcal{I}_h^\ell\}_{h=1}^H$ is $\kappa = 0.01 \log^{-1}(1/\zeta)$-replicable. It follows from the union bound that the collection of the state combinations is 0.01-replicable. Since the sets $\{\mathcal{S}_h^\ell\}_{h \in [H], \ell \in [\log(1/\zeta)]}$ are constructed solely from these state combinations, we immediately have that $\{\mathcal{S}_h^\ell\}_{h \in [H], \ell \in [\log(1/\zeta)]}$ are 0.01-replicable as well.

Fix some $\ell \in [\log(1/\zeta) - 1]$. Again by the guarantees of Lemma E.14, we have that there exist some $m_{s,h,\ell} \in \mathbb{N}$ such that the following hold with probability at least $1 - 0.01 \log^{-1}(1/\zeta)$:

$$R_1^\pi\left(x_{\mathrm{ini}}; \{\mathcal{I}_h^\ell\}_{h=1}^H\right) \leq 2^{-\ell}, \tag{33}$$

---

**Algorithm 6** REPLEVELEXPLORE

---

**Input:** $\zeta \in (0, 1)$ that controls the number of samples collected by the algorithm.

**Output:** collection of subsets of states $\{\mathcal{S}_h^\ell\}_{h \in [H], \ell \in [\log(1/\zeta)]}$ and datasets $\{\mathcal{D}_{s,a,h}\}_{s \in \mathcal{S}, a \in \mathcal{A}, h \in [H]}$.

1. Run REPEXPLORE for $\log(1/\zeta) - 1$ many times. In particular, in the $\ell$-th run, we run REPEXPLORE with $\lambda := 2^{-\ell}$, $\beta := 2^\ell \zeta$, and $\kappa := 0.01 \log^{-1}(1/\zeta)$.

2. Denote by $\{\mathcal{I}_h^\ell\}_{h=1}^H$ the state combination, and $\{\mathcal{D}_{s,a,h}^\ell\}_{s \in \mathcal{S}, a \in \mathcal{A}, h \in [H]}$ the datasets produced by REPEXPLORE in the $\ell$-th run.

3. For each $h \in [H]$, we construct the final state combinations $\{\mathcal{S}_h^\ell\}_{\ell=1}^{\log(1/\zeta)}$ produced by REPLEVELEXPLORE iteratively as follows: $\mathcal{S}_h^1 = \mathcal{S} \backslash \mathcal{I}_h^{(1)}, \mathcal{S}_h^\ell = (\mathcal{S} \backslash \mathcal{I}_h^\ell) \backslash \left(\bigcup_{\ell' < \ell} \mathcal{S}_h^{\ell'}\right)$ for $2 \le \ell \le \log(1/\zeta) - 1$, and $\mathcal{S}_h^{\log(1/\zeta)} = \mathcal{I}_h^{\log(1/\zeta)-1} \backslash \bigcup_{\ell' < \log(1/\zeta)} \mathcal{S}_h^{\ell'}$.

4. For each $s, a, h$, merge the datasets $\{\mathcal{D}_{s,a,h}^\ell\}_{\ell \in [\log(1/\zeta)]}$ into a single one $\mathcal{D}_{s,a,h}$.

5. Output the collection of sets of states $\{\mathcal{S}_h^\ell\}_{h \in [H], \ell \in [\log(1/\zeta)]}$ and the datasets $\mathcal{D}_{s,a,h}$.

---

$$\forall s \in \mathcal{S}, h \in [H] \ \min_a |\mathcal{D}_{s,a,h}^\ell| \ge m_{s,h,\ell}\,, \tag{34}$$

$$\sum_{h=1}^H \sqrt{\sum_{s \notin \mathcal{I}_h^\ell} \frac{1}{m_{s,h,\ell}}} \le 2^\ell \zeta. \tag{35}$$

By the union bound, the above hold for all $\ell \in [\log(1/\zeta) - 1]$ with probability at least 0.99. We will condition on the above events in the rest of the proof.

Next, we show that REPLEVELEXPLORE returns a tiered reachability partition (Definition D.3) of $\mathcal{M}$. Fix some $1 < \ell < \log(1/\zeta)$. Recall that $\mathcal{S}_h^\ell$ is constructed as

$$\mathcal{S}_h^\ell := (\mathcal{S} \backslash \mathcal{I}_h^\ell) \backslash \left(\bigcup_{\ell' < \ell} \mathcal{S}_h^{\ell'}\right). \tag{36}$$

It follows that $\bigcup_{\ell'=1}^\ell \mathcal{S}_h^{\ell'} \supseteq \mathcal{S} \backslash \mathcal{I}_h^\ell$, which further implies that

$$\mathcal{S} \backslash \bigcup_{\ell'=1}^\ell \mathcal{S}_h^{\ell'} \subseteq \mathcal{I}_h^\ell. \tag{37}$$

From Equation (36), it is not hard to see that $\mathcal{S}_h^\ell \subseteq S \backslash \bigcup_{\ell'=1}^{\ell-1} \mathcal{S}_h^{\ell'}$. Besides, $\mathcal{S}_h^{\log(1/\zeta)} \subseteq \mathcal{I}_h^{\log(1/\zeta)-1}$ by construction. Combining this observation with Equation (37) then yields that $\mathcal{S}_h^\ell \subseteq \mathcal{I}_h^{\ell-1}$ for all $\ell > 1$. We thus have that

$$R_1^\pi(x_{\mathrm{ini}}; \{\mathcal{S}_h^\ell\}_{h=1}^H) \le R_1^\pi(x_{\mathrm{ini}}; \{\mathcal{I}_h^{\ell-1}\}_{h=1}^H) \le 2^{1-\ell}\,,$$

where the last inequality follows from Equation (33). It then follows from Lemma E.3 that $\sum_{h=1}^H \Pr[x_h \in \mathcal{S}_h^\ell | \pi] \le 2^{1-\ell}$. This concludes the proof of Property 1.

We then move to Property 2 and show that the collection of datasets $\{\mathcal{D}_{s,a,h}\}$ are $\zeta$-nice with respect to $\{\mathcal{S}_h^\ell\}$ (Definition D.4). Sample independence follows simply from the sample independence of $\mathcal{D}_{s,a,h}$ produced by REPEXPLORE and that each execution of REPEXPLORE is executed on fresh interactions with $\mathcal{M}$. We conclude with the variable sample lower bound. From Equation (34), we already have that $\min_a |\mathcal{D}_{s,a,h}^\ell| \ge m_{s,h,\ell}$ for all $s \in \mathcal{S}, h \in [h], \ell \in [\log(1/\zeta)]$. Since $\mathcal{D}_{s,a,h}$ is obtained by combining $\{\mathcal{D}_{s,a,h}^\ell\}_\ell$, we immediately have that $|\mathcal{D}_{s,a,h}| \ge \max_\ell |\mathcal{D}_{s,a,h}^\ell| \ge \max_\ell m_{s,h,\ell}$. To show Equation (12), we note that Equation (36) immediately implies that $\mathcal{S}_h^\ell \subseteq \mathcal{S} \backslash \mathcal{I}_h^\ell$ for all $1 < \ell < \log(1/\zeta)$. Besides, recall

that $\mathcal{S}_h^1 = \mathcal{S} \backslash \mathcal{I}_h^\ell$ by construction. Consequently, for all $\ell < \log(1/\zeta)$, we must have that

$$\sum_{h=1}^{H} \sqrt{\sum_{s \in \mathcal{S}_h^\ell} \frac{1}{m_{s,h,\ell}}} \leq \sum_{h=1}^{H} \sqrt{\sum_{s \in \mathcal{S} \backslash \mathcal{I}_h^\ell} \frac{1}{m_{s,h,\ell}}} = \sum_{h=1}^{H} \sqrt{\sum_{s \notin \mathcal{I}_h^\ell} \frac{1}{m_{s,h,\ell}}} \leq 2^\ell \zeta \,,$$

where the last inequality follows from Equation (35). This shows Equation (12) and concludes the proof of Property 2.

Lastly, we analyze the sample and computational complexity of the algorithm. Fix some $\ell$. By Lemma E.14, the number of episodes consumed by the $\ell$-th run of **REPEXPLORE** is at most

$$O\left(S^2 A H^6 \, 2^{2\ell} \, \log^4(1/\zeta) \, \left(2^\ell \zeta\right)^{-2}\right) = \tilde{O}\left(S^2 A H^6 \, \zeta^{-2}.\right)$$

Since there are $\log(1/\zeta) - 1$ many runs of **REPEXPLORE** in total, it follows immediately that the total sample complexity is $\tilde{\Theta}\left(S^2 A H^7 \, \zeta^{-2}\right)$. The runtime of the algorithm is polynomial in the number of episodes times $\exp\left(O(SH)\right)$ as the number of times we invoke correlated sampling on $SH$-dimensional distribution is at most polynomial. This concludes the analysis of the **REPLEVELEXPLORE** algorithm.

$\square$

### E.4. Proof of Lemma E.6: Bounding Reachability via Value Estimates

Our goal is to show that the estimates $V_h^k$ maintained by the learning agent bounds from above the reachability function $R_h^\pi(\cdot; \{\mathcal{U}_h^k\}_{h=1}^H)$, where $\{\mathcal{U}_h^k\}_{h=1}^H$ is the under-explored state combination by the end of episode $k$ and $\pi$ is some arbitrary policy. As mentioned, the argument is conceptually similar to the one given in (Jin et al., 2018) showing that the estimates must bound from above the corresponding $V^*$-value of each state. However, note that unlike $V^*$, which is an invariant function throughout the training process, the reachability function $R_h^\pi(\cdot; \{\mathcal{U}_h^k\}_{h=1}^H)$ actually depends on the under-explored state combination $\{\mathcal{U}_h^k\}_{h=1}^H$ which is evolving over time. To deal with the extra complication, our proof exploits the fact that each set $\mathcal{U}_h^k$ will only shrink as $k$ increases, and the reachability function $R_h^\pi$ is a monotone set function with respect to $\mathcal{U}_h^k$.

*Proof of Lemma E.6.* We will be using the following notations.

- $Q_h^k(x, a)$: the estimate of the $Q$-value of the tuple $(x, a, h) \in \mathcal{S} \times \mathcal{A} \times [H]$ maintained by the $Q$-learning agent **QAGENT** at the beginning of the $k$-th episode of training.

- $V_h^k(x)$: the estimate of the $V$-value of the state $x$ maintained by the $Q$-learning agent at step $h$ at the beginning of the $k$-th episode of training. Note that $V_h^k(x)$ is simply given by

$$V_h^k(x) := \min\left(H, \max_{a' \in \mathcal{A}} Q_h^k(x, a')\right) \forall x \in \mathcal{S}.$$

- $N_h^k(x, a)$ : number of times the agent has visited the state-action pair $(x, a)$ at step $h$ at the beginning of episode $k$.

For analysis purposes, we define the following "pseudo-value" functions that is closely related to the reachability function:

$$\bar{V}_{H+1}^{(k)}(x) = 0 \,,$$
$$\forall h \in [H] : \bar{V}_h^k(x) = \max_a \bar{Q}_h^k(x, a) \,,$$
$$\bar{Q}_h^k(x, a) = \begin{cases} H \text{ if } N_h^k(x, a) < H \text{ (i.e., if } x \in \mathcal{U}_h^k), \\ \mathbb{E}_{x' \sim \mathbf{P}_h(x, a)}\left[\bar{V}_{h+1}^{(k)}(x')\right] \text{ otherwise.} \end{cases}$$

We first show that

$$R_h^\pi(x; \{\mathcal{U}_h^k\}_{h=1}^H) \leq \bar{V}_h^k(x) \tag{38}$$

for all $k \in [K], h \in [H], x \in \mathcal{S}$ and policy $\pi$. We will show via induction on $h$. Equation (38) is trivially true for $h = H + 1$ as both of them are defined to be 0 in this case. Assume that Equation (38) holds for all $h' \geq h + 1$. We now show that it

also holds for $h$. We break into two cases. If $x \in \mathcal{U}_h^k$, following the definition $\mathcal{U}_h^k$ in the lemma statement, we immediately have that $\bar{V}_h^k(x) = H \geq R_h^\pi(x; \{\mathcal{U}_h^k\}_{h=1}^H)$. Otherwise, we have that

$$
\begin{aligned}
R_h^\pi(x; \{\mathcal{U}_h^k\}_{h=1}^H) &= \mathbb{E}_{x' \sim \mathbf{p}_h(x, \pi_h(x))} \left[ R_{h+1}^\pi(x'; \{\mathcal{U}_h^k\}_h) \right] \\
&\leq \mathbb{E}_{x' \sim \mathbf{p}_h(x, \pi_h(x))} \left[ \bar{V}_{h+1}^k(x') \right] \\
&\leq \max_a \mathbb{E}_{x' \sim \mathbf{p}_h(x, a)} \left[ \bar{V}_{h+1}^k(x') \right] = \bar{V}_h^k(x) \,,
\end{aligned}
$$

where the first equality follows from the definition of the reachability function and the fact that $x \notin \mathcal{U}_h^k$, the first inequality follows from the inductive hypothesis, and the final equality follows from the definition of the pseudo-value functions and the fact that $x \notin \mathcal{U}_h^k$ is explored. This concludes the proof of Equation (38). Note that since $\bar{V}_h^k$ is independent of any policy, in the remaining proof we no longer need to consider specific policies.

It then suffices to show that $\bar{V}_1^k(s) \leq \bar{V}_h^k(s)$ for all $s \in \mathcal{S}, k \in [K], h \in [H]$ with probability at least $1 - \iota$. In particular, we will show the stronger version of the statement:

$$
\forall x \in \mathcal{S}, a \in \mathcal{A}, h \in [H], k \in [K] \,, Q_h^k(x, a) \geq \bar{Q}_h^k(x, a) \tag{39}
$$

with probability at least $1 - \iota$. We will show Equation (39) via induction on $k$. In particular, the inductive hypothesis states that

$$
\forall x \in \mathcal{S}, a \in \mathcal{A}, h \in [H], k' \in [k] \,, Q_h^{k'}(x, a) \geq \bar{Q}_h^{k'}(x, a) \tag{40}
$$

with probability at least $1 - (k-1)\iota/K$.

In the base case, we have $k = 1$. Note that in this case $N_h^1(x, a)$ will be 0 for all $h, x, a$, and $Q_h^1(x, a)$ will be equal to its initial value $H$ (see Line 1 of **QAGENT**). It then follows from the definition of $\bar{Q}_h^1(x, a)$ that the inductive hypothesis holds. Now consider the $k^*$-th inductive step. We will condition on that Equation (40) holds for $k = k^* - 1$. Furthermore, fix a tuple $(x^*, a^*, h^*) \in \mathcal{S} \times \mathcal{A} \times [H]$. From now on, $x^*, a^*, h^*, k^*$ is a fixed tuple. We proceed to show that

$$
Q_{h^*}^{k^*}(x^*, a^*) \geq \bar{Q}_{h^*}^{k^*}(x^*, a^*) \,,
$$

with probability at least $1 - \iota/(SAHK)$. Equation (39) will then follow from the union bound. We begin by introducing some extra notations.

- For $t \in [H], k \in [K]$, denote by $x_t^k$ the state the agent **QAGENT** is at step $t$ in episode $k$.

- Define $[\mathbf{p}_{h^*} \cdot \bar{V}_{h^*+1}^{k^*}](x^*, a^*) := \mathbb{E}_{x' \sim \mathbf{p}_{h^*}(x^*, a^*)} \left[ \bar{V}_{h^*+1}^{k^*}(x') \right]$ and $[\hat{\mathbf{p}}_{h^*}^{k^*} \cdot \bar{V}_{h^*+1}^{k^*}](x^*, a^*) := \bar{V}_{h^*+1}^{k^*}(x_{h^*+1}^{k^*})$.

- Let $N^*$ be the visitation counts of $(x^*, a^*)$ at step $h^*$ before episode $k^*$, i.e., $N^* := N_{h^*}^{k^*}(x^*, a^*)$.

- Let $k_i^*$ be the episode index when the state action pair $(x^*, a^*)$ is taken at step $h^*$ for the $i$-th time, and we define $k_i^* = k^* + 1$ if the total visitation count $N^* = N_{h^*}^{k^*}(x^*, a^*)$ is less than $i$. Note that $k_i^* \neq k^*$ since we can never visit a state action pair $k^*$ times before the $k^*$-th episode. Thus, we always have either $k_i^* < k^*$ or $k_i^* = k^* + 1$.

- It is not hard to see that $k_i^*$ is a stopping time. We define $\mathcal{F}_i$ to be the $\sigma$-algebra generated by all random variables until episode $k_i^*$, step $h^*$ but excluding the most recent transition information, i.e., $\{x_t^k\}_{t \in [H], k < k_i^*} \cup \{x_t^{k_i^*}\}_{t \leq h^*}$.

Instead of recalling the full detail of how the estimate $Q_h^k(x, a)$ is computed by the $Q$-learning algorithm, we will use Equation 4.3 of (Jin et al., 2018) that relates $Q_h^k(x, a)$ to the past $V$-estimates before episode $k$. In particular, we have that[18]

$$
Q_{h^*}^{k^*}(x^*, a^*) = \sum_{i=1}^{N^*} \alpha_{N^*}^i \left( V_{h^*+1}^{k_i^*}(x_{h^*+1}^{k_i^*}) + b_i \right) \,, \tag{41}
$$

---

[18]Recall that we set the rewards of the MDP to 0. Hence, there are no reward terms in the decomposition. Other than that, the decompositions are identical.

where $b_i := C\sqrt{H^3 \log(SAHK/\iota)/i}$ for some sufficiently large constant $C$ is the optimistic bonus added to the estimates, and $\alpha^i_{N^*}$ are positive numbers satisfying the following properties

$$\sum_{j=1}^{N^*} \alpha^j_{N^*} = 1 \,, \tag{42}$$

$$\sum_{j=1}^{N^*} \left(\alpha^j_{N^*}\right)^2 \leq \frac{2H}{N^*} \,, \tag{43}$$

$$\sum_{i=1}^{N^*} \frac{\alpha^i_{N^*}}{\sqrt{i}} \geq \frac{1}{\sqrt{N^*}} \,. \tag{44}$$

We omit the exact expressions of $\alpha^i_{N^*}$ since they are not important for this argument.

In the rest of the proof, we will condition on the following concentration event.

$$\forall \tau \in [k^*], \left| \sum_{i=1}^{\tau} \alpha^i_\tau \mathbb{1}\{k^*_i < k^*\} \left[ \left(\mathbf{p}_{h^*} - \hat{\mathbf{p}}^{k^*_i}_{h^*}\right) \cdot \bar{V}^{k^*_i}_{h^*+1} \right](x^*, a^*) \right| \leq O(1) \sqrt{\frac{H^3 \log(SAHK/\iota)}{\tau}} \,. \tag{45}$$

In particular, we will later argue that the above inequality holds with probability at least $1 - \iota/(SAHK)$. The argument is deferred to the end of the proof.

Conditioned on Equation (45) and some arbitrary value of $N^*$, we proceed to show that $Q^{k^*}_{h^*}(x^*, a^*) \geq \bar{Q}^{k^*}_{h^*}(x^*, a^*)$. Suppose we have not visited the state-action pair $(x^*, a^*)$ at step $h^*$ before episode $k^*$, i.e., $N^* = 0$. Then $Q^{k^*}_{h^*}(x^*, a^*)$ must be equal to its initial value $H$, which is the same as $\bar{Q}^{k^*}_{h^*}(x^*, a^*)$. So the bound holds trivially. Otherwise, we have some $k^*_1, \ldots, k^*_{N^*} < k^*$. We will further break into two cases. Suppose $N^* < H$. Namely, $k^*$ is some episode before we visit the $(x^*, a^*)$ at step $h^*$ for the $H$-th time. Then the total optimism bonus $b_i$ added to the $Q$ estimate is

$$\sum_{i=1}^{N^*} \alpha^i_{N^*} b_i \geq C\sqrt{\frac{H^3 \log(SAHK/\iota)}{N^*}} \geq H \tag{46}$$

where the first inequality is by the definition of the optimistic bonus $b_i$ and Equation (44), and the second inequality is by our case assumption that $N^* < H$. On the other hand, it is clear from the definition that $\bar{Q}^{k^*}_{h^*}(x^*, a^*)$ is at most $H$. Thus, we always have that $Q^{k}_{h^*}(x^*, a^*) \geq \bar{Q}^{k}_{h^*}(x^*, a^*)$ in the case $N^* < H$.

Now consider the remaining case when $N^* \geq H$. Since we assume that we have visited $(x^*, a^*)$ at step $h^*$ for at least $H$ times before episode $k^*$, by the definition of $\bar{Q}$, we must have that

$$\bar{Q}^{k^*}_{h^*}(x^*, a^*) = \left[ \mathbf{p}_{h^*} \cdot \bar{V}^{k^*}_{h^*+1} \right](x^*, a^*).$$

Note that $\sum_{i=1}^{N^*} \alpha^i_{N^*} = 1$. Moreover, it is not hard to verify via induction that $\bar{V}^k_h(x)$ is monotonically decreasing as a function of $k$ for all $x, h$. Thus, we have that

$$\bar{Q}^{k^*}_{h^*}(x^*, a^*) \leq \sum_{i=1}^{N^*} \alpha^i_{N^*} \left[ \mathbf{p}_{h^*} \cdot \bar{V}^{k^*_i}_{h^*+1} \right](x^*, a^*). \tag{47}$$

It then follows that

$$Q^{k^*}_{h^*}(x^*, a^*) - \bar{Q}^{k^*}_{h^*}(x^*, a^*)$$
$$\geq \sum_{i=1}^{N^*} \alpha^i_{N^*} \left( V^{k^*_i}_{h^*+1} - \bar{V}^{k^*_i}_{h^*+1} \right)\left(x^{k^*_i}_{h^*+1}\right) + \sum_{i=1}^{N^*} \alpha^i_{N^*} \left( \bar{V}^{k^*_i}_{h^*+1}\left(x^{k^*_i}_{h^*+1}\right) - \left[ \mathbf{p}_{h^*} \cdot \bar{V}^{k^*_i}_{h^*+1} \right](x^*, a^*) \right) + \sum_{i=1}^{N^*} \alpha^i_{N^*} b_i.$$

For the first term, since $k^*_i < k^*$, we can use the inductive hypothesis and conclude that it is non-negative. For the second term, we note that $\bar{V}^{k^*_i}_{h^*+1}\left(x^{k^*_i}_{h^*+1}\right)$ is just $\left[ \hat{\mathbf{p}}^{k^*_i}_{h^*} \cdot \bar{V}^{k^*_i}_{h^*+1} \right](x^*, a^*)$ by definition, and hence we can write

$$\sum_{i=1}^{N^*} \alpha^i_{N^*} \left( \bar{V}^{k^*_i}_{h^*+1}\left(x^{k^*_i}_{h^*+1}\right) - \left[ \mathbf{p}_{h^*} \cdot \bar{V}^{k^*_i}_{h^*+1} \right](x^*, a^*) \right) = \sum_{i=1}^{N^*} \alpha^i_t \left[ \left(\hat{\mathbf{p}}^{k_i}_{h^*} - \mathbf{p}_{h^*}\right) \cdot \bar{V}^{k^*_i}_{h^*} \right](x^*, a^*)$$

$$\geq -O\left(\sqrt{\frac{H^3 \log(SAHK/\iota)}{N^*}}\right),$$

where the last inequality follows from applying Equation (45) with $\tau = N^*$. For the third term, Equation (46) shows that it is least $C\sqrt{H^3 \log(SAHK/\iota)/N^*}$. Thus, we can conclude that $Q_{h^*}^{k^*}(x^*, a^*) - \bar{Q}_{h^*}^{k^*}(x^*, a^*)$ is non-negative if we choose $C$ to be some sufficiently large constant. This shows that Equation (39) for a fixed tuple $(x^*, a^*, h^*, k^*)$ holds with probability at least $1 - \iota/(SAHK)$. By the union bound, Equation (39) holds for all $(x, a, h, k)$ at the same time with probability at least $1 - \iota$. This then further implies that $\bar{V}_h^k(s) \leq V_h^k(s)$ for all $s, h, k$. Combining this with Equation (38) then concludes the proof of Equation (24).

It remains to show that Equation (45) holds with probability at least $1 - \iota/(SAHK)$. Consider the sequence

$$\left\{\mathbb{1}\{k_i^* < k^*\}\left[\left(\mathbf{p}_{h^*} - \hat{\mathbf{p}}_{h^*}^{k_i^*}\right) \cdot \bar{V}_{h^*+1}^{k_i^*}\right](x^*, a^*)\right\}_{i=1}^{k^*}. \tag{48}$$

We claim that the sequence is a martingale difference sequence with respect to the filtration $\{\mathcal{F}_i\}_{i=1}^{k^*}$. First, both $\mathbb{1}\{k_i^* < k^*\}$ and $\bar{V}_{h^*+1}^{k_i^*}(\cdot)$ are measurable with respect to $\mathcal{F}_i$. The former is true by definition. For the latter, recall that $\{\bar{V}_h^{k_i^*}(x)\}_{x \in \mathcal{S}, h \in [H]}$ are functions of $\{N_h^{k_i^*}(x)\}_{x \in \mathcal{S}, h \in [H]}$, which are the visitation counts of the states *before* episode $k_i^*$. Clearly, $\{N_h^{k_i^*}(x)\}_{x \in \mathcal{S}, h \in [H]}$ are measurable with respect to $\mathcal{F}_i$, and therefore so are $\{\bar{V}_h^{k_i^*}(x)\}_{x \in \mathcal{S}, h \in [H]}$. It then remains to show that

$$\mathbb{E}\left[[\hat{\mathbf{p}}_{h^*}^{k_i^*} \cdot \bar{V}_{h^*+1}^{k_i^*}](x^*, a^*) \mid \mathcal{F}_i\right] = \mathbb{E}\left[[\mathbf{p}_{h^*} \cdot \bar{V}_{h^*+1}^{k_i^*}](x^*, a^*) \mid \mathcal{F}_i\right].$$

The left hand side is simply $\mathbb{E}\left[\hat{V}_{h^*+1}^{k_i^*}(x_{h^*+1}^{k_i^*}) \mid \mathcal{F}_i\right]$. To see why this is true, we note that since we have conditioned on $\mathcal{F}_i$, the conditional expectation of the left hand side is just over the randomness of $x_{h^*+1}^{k_i^*}$. In particular, the distribution of $x_{h^*+1}^{k_i^*} \mid \mathcal{F}_i$ is the same as the transition distribution $\mathbf{p}_{h^*}(x^*, a^*)$ since $k_i^*$ is defined to be the index of some episode when we visit $(x^*, a^*)$ at step $h^*$. The equality hence follows. This concludes the proof that the sequence in Equation (48) is indeed a martingale difference sequence with respect to $\{\mathcal{F}_i\}_{i=1}^{k^*}$.

By Azuma's inequality and the union bound, with probability at least $1 - \iota/(SAHK)$, it holds

$$\forall \tau \in [k^*], \left|\sum_{i=1}^{\tau} \alpha_\tau^i \mathbb{1}\{k_i^* < k^*\}\left[\left(\mathbf{p}_{h^*} - \hat{\mathbf{p}}_{h^*}^{k_i^*}\right) \cdot \bar{V}_{h^*+1}^{k_i^*}\right](x^*, a^*)\right| \leq O(H)\sqrt{\sum_{i=1}^{\tau}(\alpha_\tau^i)^2 \log(SAHK/\iota)}$$

$$\leq O(1)\sqrt{\frac{H^3 \log(SAHK/\iota)}{\tau}}, \tag{49}$$

where the second inequality follows from Equation (43). This therefore concludes the proof of Equation (45) as well as Lemma E.6. $\qquad\square$

# F. Proof of Theorem 1.3: Replicable Learning in the Episodic Setting

We now present our overall algorithm for reinforcement learning in the episodic setting.

**Theorem F.1** (Formal Version of Theorem 1.3). *There is a $\rho$-replicable $(\varepsilon, \delta)$-PAC policy estimator with sample complexity*

$$n(S, A, H, \varepsilon, \delta) \leq \tilde{O}\left(\frac{S^2 A H^{11} \log^4(1/\delta)}{\rho^2 \varepsilon^2}\right).$$

*Moreover, the runtime is* $\text{poly}\left(n(S, A, H, \varepsilon, \delta)\rho^{-1}\right) \cdot \exp(O(SH)).$

In contrast to the parallel sampling model, we use our exploration algorithm **REPLEVELEXPLORE** to obtain the necessary good state partitions and datasets.

*Proof of Theorem F.1.* We first give a $0.1$-replicable $(\varepsilon, 0.1)$-PAC policy estimator. The result then follows from applying the boosting (Lemma D.1).

Given $\varepsilon \in (0, 1)$ and episodic access to MDP $\mathcal{M}$, our algorithm sets $\zeta$ to be a number satisfying

$$\zeta \ll \frac{\varepsilon}{H^2 \log^5 (SAH/\varepsilon)}$$

as required by Theorem D.5. By Lemma E.1, running **REPLEVELEXPLORE** with parameter $\zeta$ on the MDP $\mathcal{M}$ produces a tiered reachability partition $\{\mathcal{S}_h^\ell\}_{h\in[H],\ell\in[L]}$, where $L = \lceil \log(1/\zeta) \rceil$, and a collection of datasets $\{\mathcal{D}_{s,a,h}\}$ that are $\zeta$-nice (Definition D.4) with respect to $\{\mathcal{S}_h^\ell\}$ with high constant probability. Then, by Theorem D.5, running **REPRLBANDIT** with accuracy produces a policy $\varepsilon/2$-optimal policy with high constant probability.

For correctness, note that if both **REPLEVELEXPLORE** and **REPRLBANDIT** succeed, which happens with high constant probability, it follows that the policy is $\varepsilon/2$-optimal. We then analyze the algorithm's replicability in two runs. By Lemma E.1, **REPLEVELEXPLORE** outputs the same collection of subsets $\{\mathcal{S}_h^\ell\}$ with probability at least 0.99, and two collections of random datasets $\{\mathcal{D}_{s,a,h}^{(1)}\}, \{\mathcal{D}_{s,a,h}^{(2)}\}$ that are both $\zeta$-nice with respect to $\{\mathcal{S}_h^\ell\}$. Conditioning on this, Theorem D.5 then produces identical policies in two runs with probability at least 0.99.

It then follows that the episodic PAC Policy Estimator has accuracy $\varepsilon/2$, fails with probability at most 0.1, and is 0.02-replicable.

Next we bound the sample complexity of our algorithm. From Lemma E.1, **REPLEVELEXPLORE** has sample complexity $\tilde{O}\left(S^2 A H^7 \zeta^{-2}\right)$. Note that our constraint on $\zeta$ is satisfied by $\zeta = c\frac{\varepsilon}{H^2 \mathrm{polylog}(SAH/\varepsilon)}$ for some sufficiently small constant $c > 0$. In particular, the sample complexity can be bounded as $\tilde{O}\left(\frac{S^2 A H^{11} \log(1/\delta)}{\varepsilon^2}\right)$.

Lastly, we discuss the runtime of our algorithm. Combining the runtime of Lemma E.1 and Theorem D.5, we have that the run-time is $\mathrm{poly}\left(n(S, A, H, \varepsilon, \delta)\rho^{-1}\right) \cdot \exp(O(SH))$ $\qquad\square$

## G. Lower Bounds for Replicable Reinforcement Learning

We begin with our lower bound in the parallel sampling setting, proving that Theorem D.13 is essentially optimal in dependence on $S, A$.

**Theorem 1.4** (Parallel Sampling Lower Bound). *Any $\rho$-replicable algorithm that is a $(\varepsilon H, 0.001)$-PAC policy estimator requires $\tilde{\Omega}\left(\frac{S^2 A H^2}{\rho^2 \varepsilon^2}\right)$ samples in the parallel sampling model.*

We obtain our lower bound via a reduction to the sign-one-way marginals problem, which we define below. A Rademacher random variable with parameter $x \in [-1, 1]$, denoted by $\mathrm{Rad}\,(x)$, is a random variable that equals $+1$ with probability $\frac{1+x}{2}$ and $-1$ with probability $\frac{1-x}{2}$. A product of Rademacher random variables with parameter $p \in [-1, +1]^n$, which we denote by $\mathrm{Rad}\,(p)$, is an independent product of $n$ Rademacher random variables where the $i$-th coordinate has parameter $p_i$.

**Definition G.1** (Sign-One-Way Marginals). Let $p \in [-1, +1]^n$. We say $v \in [-1, 1]^n$ is an $\varepsilon$-accurate solution to the $n$-dimensional sign-one-way marginals problem with respect to $\mathrm{Rad}\,(p)$ if $\frac{1}{n}\sum_{i=1}^n v_i p_i > \frac{1}{n}\sum_{i=1}^n |p_i| - \varepsilon$ or equivalently

$$\sum_{i=1}^n \left(\mathrm{sgn}(p_i) - v_i\right) p_i < \varepsilon n, \tag{50}$$

where $\mathrm{sgn} : \mathbb{R} \mapsto \{\pm 1\}$ computes the sign of the input.

Furthermore, we say an algorithm is $(\varepsilon, \delta)$-accurate for the $n$-dimensional sign-one-way marginals problem if given sample access to an arbitrary $n$-dimensional distribution $\mathrm{Rad}\,(p)$, the algorithm outputs an $\varepsilon$-accurate solution to the sign-one-way marginals problem with respect to $\mathrm{Rad}\,(p)$ with probability at least $1 - \delta$.

(Bun et al., 2023) gave a $\tilde{O}(n\rho^{-2}\varepsilon^{-2})$ sample complexity algorithm for the sign-one-way marginals problem and showed that any algorithm with constant replicability and accuracy requires $\tilde{\Omega}(n)$ samples. However, it was unclear how to obtain a tight characterization of sample complexity with respect to replicability and accuracy parameters $\rho, \varepsilon$ from their methods. As a first step towards characterizing the sample complexity of replicable reinforcement learning, we resolve this gap.

**Theorem G.2.** *Any $\rho$-replicable $(\varepsilon, 0.01)$-accurate algorithm for the $n$-dimensional sign-one-way marginals problem requires $\Omega\left(\frac{n}{\rho^2 \varepsilon^2 \log^6 n}\right)$ samples.*

We defer the proof of Theorem G.2 to Appendix I. Our result yields also tight sample complexity lower bounds for replicable $\ell_1$-mean estimation (see Theorem I.8). This extends the work of (Hopkins et al., 2024), who gave sample complexity lower bounds for $\ell_p$-mean estimation for $p \geq 2$. We therefore complete their results in the main missing regime, i.e., $\ell_1$-mean estimation.

*Proof of Theorem 1.4.* Our lower bound argues that a PAC policy estimator that requires few parallel samples induces an algorithm for sign-one-way marginals with low sample complexity. Fix some sufficiently large $S, A, H$. Let $n = SH$ and let $\mathbf{p}$ be a product of $n$ Rademacher distributions. We will construct an MDP with $S$ states, $A$ actions, and $H$ time steps according to $\mathbf{p}$. Assume without loss of generality that the action set $\mathcal{A}$ consists of two special actions $\{a_+, a_-\}$ and $A - 2$ dummy actions $\{a_\perp\}$. For clarity, we present an MDP $\mathcal{M}$ with rewards in the range $[-2, 1]$ instead of $[0, 1]$. Note that a $\varepsilon H$-optimal policy estimator for reward range $[0, 1]$ implies a $3\varepsilon H$-optimal policy estimator on $[-2, 1]$, since we may scale the rewards to $[0, 1]$ incurring a error factor of 3. Define the MDP as follows.

1. We define the transition probabilities of every state-action pair to be uniformly over all states.

2. Suppose we re-index $\mathbf{p}$ with tuples $(s, h) \in \mathcal{S} \times [H]$.

3. For the tuple $(s, a_+, h)$, we define the reward distribution $\mathbf{r}_h(s, a_+)$ to be exactly $\mathbf{p}_{(s,h)}$.

4. For the tuple $(s, a_-, h)$, we define the reward distribution $\mathbf{r}_h(s, a_+)$ to be exactly $-\mathbf{p}_{(s,h)}$.

5. For any tuple $(s, a_\perp, h)$, we define the reward to be deterministically $-2$.

We can easily simulate a call to the parallel sampling oracle $\mathbf{PS}(G_\mathcal{M})$ with 2 samples from $\mathbf{p}$. Any transition from any $(s, a, h)$ tuple and any reward from any $(s, a_\perp, h)$ can be simulated without any access to $\mathbf{PS}(G_\mathcal{M})$. Two calls to $\mathbf{PS}(G_\mathcal{M})$ suffices to simulate a sample for actions $a_+, a_-$ for all $(s, h)$ tuples.

Let $\mathcal{L}$ be an $(3\varepsilon H, 0.002)$-PAC policy estimator (on reward range $[-2, 1]$) that makes $m$ calls to the parallel sampling oracle $\mathbf{PS}(G_\mathcal{M})$. If we run $\mathcal{L}$ on $\mathcal{M}$, the algorithm produces a policy $\hat{\pi}$ that is $3\varepsilon H$-optimal with respect to $\mathcal{M}$ with high constant probability. Conditioned on that, we can construction a solution $v$ to the original sign-one-way marginal problem by setting $v_{s,h} := 1$ if $\hat{\pi}_h(s) = a_+$ and $v_{s,h} := -1$ if $\hat{\pi}_h(s) = a_-$. If $\hat{\pi}_h(s) = a^\perp$ is any of the dummy actions, we set $v_s$ arbitrarily. It then suffices to show that $v$ satisfies $\frac{1}{n} \sum_{i=1}^n v_i p_i \geq \frac{1}{n} \sum_{i=1}^n |p_i| - 0.02$, where $p_i$ is the mean of the $i$-th coordinate of $\mathbf{p}$.

Again, we will re-index $p$ by the tuple $(s, h) \in \mathcal{S} \times [H]$. The value of the policy $\hat{\pi}$ is given by

$$V(\hat{\pi}, \mathcal{M}) = \sum_{h=1}^H \sum_{s \in \mathcal{S}} \frac{p_{(s,h)}}{S} \left( \mathbb{1}\{\hat{\pi}_h(s) = a_+\} - \mathbb{1}\{\hat{\pi}_h(s) = a_-\} \right) - \frac{2}{S} \mathbb{1}\{\hat{\pi}_h(s) = a_\perp\}$$

$$= \frac{1}{|\mathcal{S}|} \sum_{h=1}^H \sum_{s \in \mathcal{S}} v_{s,h} p_{(s,h)} \mathbb{1}\{\hat{\pi}_h(s) \neq a_\perp\} - 2 \cdot \mathbb{1}\{\hat{\pi}_h(s) = a_\perp\}$$

$$\leq \frac{1}{|\mathcal{S}|} \sum_{h=1}^H \sum_{s \in \mathcal{S}} v_{s,h} p_{(s,h)}.$$

where in the final inequality we use that $-2 < v_{s,h} p_{s,h}$ for all $s, h$. It is not hard to see that the value of the optimal policy is just $V(\pi^*, \mathcal{M}) = \frac{1}{|\mathcal{S}|} \sum_{h=1}^H \sum_{s \in \mathcal{S}} |p_{(s,h)}|$. Thus, the fact that $\hat{\pi}$ is $3\varepsilon H$ accurate implies that

$$\frac{1}{n} \sum_{h=1}^H \sum_{s \in \mathcal{S}} v_{s,h} p_{(s,h)} \geq \frac{S}{n} V(\hat{\pi}, \mathcal{M}) \geq \frac{S}{n} \left( V(\pi^*, \mathcal{M}) - 3\varepsilon H \right) = \frac{1}{n} \sum_{h=1}^H \sum_{s \in \mathcal{S}} |p_{(s,h)}| - 3\varepsilon.$$

In other words, $v$ is an $3\varepsilon$-accurate solution to the sign-one-way marginals problem for $\mathbf{p}$. Thus, applying Theorem G.2 shows that the PAC policy estimator has to make at least $m = \Omega\left(\frac{n}{\rho^2 \varepsilon^2 \log^5(SH)}\right) = \Omega\left(\frac{SH}{\rho^2 \varepsilon^2 \log^5 n}\right)$ calls to $\mathbf{PS}(G_\mathcal{M})$, which translates to a total sample complexity of $\Omega\left(\frac{S^2 A H^2}{\rho^2 \varepsilon^2 \log^5(SH)}\right)$. $\qquad\square$

**Lower Bound for Episodic Reinforcement Learning**   Next, we give a lower bound for Reinforcement Learning in the episodic setting, which shows that the sample complexity of our algorithm (Theorem 1.3) is essentially optimal in its dependence on $S$.

**Theorem 1.5** (Episodic Lower Bound). *Any $\rho$-replicable $(\varepsilon H, 0.001)$-PAC policy estimator in the episodic setting must have a sample complexity of $\tilde{\Omega}\left(\frac{S^2 H^2}{\rho^2 \varepsilon^2}\right)$.*

*Proof.* Suppose we have an $\rho$-replicable algorithm $\mathcal{L}$ that is a $(\varepsilon H, 0.001)$-PAC policy estimator that consumes $m$ episodes. We can assume that $m \geq SAH$ as this is needed even without the replicability requirement. We will show a lower bound of $\Omega\left(\frac{S^2 H}{\rho^2 \varepsilon^2 \log^5(SH)}\right)$ many episodes, which will imply a sample complexity lower bound of $\Omega\left(\frac{S^2 H^2}{\rho^2 \varepsilon^2 \log^5(SH)}\right)$.

Consider the $\mathcal{M}$ defined in the proof of Theorem 1.4 with uniformly random transition probabilities and random rewards parameterized by $SH$ Rademacher distributions. We claim that we can simulate $m$ episodes of interaction with the MDP using at most $\frac{m}{S} + O\left(\sqrt{\frac{m \log(SAH)}{S}}\right)$ samples from the product Rademacher distributions with high probability.

In a single episode, we arrive at state $s$ during time step $h$ with probability $\frac{1}{S}$. Let $M_{s,a,h}$ be the random variable denoting how many samples of rewards from the tuple $(s, a, h)$ are observed over $m$ episodes. Then, since for fixed $s, a, h$ we have $\mathbb{E}[M_{s,a,h}] \leq \frac{m}{S}$ and $M_{s,a,h}$ is the sum of $m$ i.i.d. bounded random variables, applying the Chernoff bound and the union bound gives that $\max M_{s,a,h} \leq \frac{m}{S} + O\left(\sqrt{\frac{m \log(SAH)}{S}}\right)$ with high constant probability.

Conditioned on that, we can therefore simulate the reward samples from all $m$ episodes with $\frac{m}{S} + O\left(\sqrt{\frac{m \log(SAH)}{S}}\right)$ samples from the product Rademacher distribution, and $\mathcal{L}$ can therefore produce some $\varepsilon H$-optimal policy with high constant probability. Following the same argument as in Theorem 1.4, we then obtain an $\varepsilon$-accurate solution for the original sign-one-way marginal problem. Applying Theorem G.2 then shows that we must have

$$\frac{m}{S} + O\left(\sqrt{\frac{m \log(SAH)}{S}}\right) = \Omega\left(\frac{n}{\rho^2 \varepsilon^2 \log^5 n}\right) = \Omega\left(\frac{SH}{\rho^2 \varepsilon^2 \log^5(SH)}\right)$$

which implies that $\Omega\left(\frac{S^2 H}{\rho^2 \varepsilon^2 \log^5(SH)}\right)$ episodes are necessary provided that $m \geq SAH$. This concludes the proof of Theorem 1.5. $\qquad\square$

# H. Efficient RL Algorithm

We provide the details necessary to obtain a computationally efficient algorithm for replicable reinforcement learning. The goal is to obtain an algorithm using $S^3$ samples and running time polynomial in sample complexity. First, we give an efficient version of **REPEXPLORE**.

**Lemma H.1** (Computationally Efficient Single-Tier Replicable Exploration (Lemma E.14)). *Fix an arbitrary policy $\pi$. Let $\delta, \kappa, \beta \in (0, 1)$. There exists some positive integers $\{m_{s,h}\}_{s \in \mathcal{S}, h \in [H]}$, and an efficient version of **REPEXPLORE** (Algorithm 5) that consumes at most $K := \tilde{\Theta}(S^3 A H^7 \delta^{-2} \kappa^{-4} \beta^{-2})$ episodes, runs in time $\mathrm{poly}(K)$, and produces a state combination $\{\mathcal{I}_h\}_{h=1}^H$, and an independent dataset $\mathcal{D}_{s,a,h}$ made up of i.i.d. samples from $\mathbf{p}_h(s,a) \otimes \mathbf{r}_h(s,a)$ for all $s \in \mathcal{S}, a \in \mathcal{A}, h \in [H]$ such that (i) the output set $\{\mathcal{I}_h\}_{h=1}^H$ is $O(\kappa)$-replicable, (ii) the following holds with probability at least $1 - O(\kappa)$:*

*(a) $R_1^\pi(x_{\mathrm{ini}}; \{\mathcal{I}_h\}_{h=1}^H) \leq \delta$ ,*

*(b) $\min_a |\mathcal{D}_{s,a,h}| \geq m_{s,h}$ and $\sum_{h \in [H]} \sqrt{\sum_{s:s \in \mathcal{I}_h} (1/m_{s,h})} \leq \kappa \beta / \sqrt{S}$.*

*Proof.* For the efficient version of Algorithm 5, we will instead set

$$m := \Theta\left(S^2 H^2 \log(SH/\kappa)\kappa^{-2}\right), M := \Theta\left(\log^2(SH/\kappa)m\right) + \Theta\left(S^2 H^2 \log(SH/\kappa)\kappa^{-2}\beta^{-2}\right),$$

and replace the correlated sampling with its efficient counterpart for product distributions (Lemma B.13). The sample complexity of the algorithm then increases by a factor of $S$, and the runtime is at most polynomial in the number of samples.

**Property (i)** Let $\hat{\mu}_{s,h}^{(1)}, \hat{\mu}_{s,h}^{(2)}$ be the estimates obtained in two runs of the algorithm. By Lemma E.11 and the union bound, we have that $\left|\hat{\mu}_{s,h}^{(i)} - \mu_{s,h}^{\delta,\iota}\right| \leq \sqrt{\log(SH/\kappa)/m} \leq \kappa/(SH)$ for $i \in \{1,2\}$ and all $s, h$ with probability at least $1 - 2\kappa$. Conditioned on that, by the triangle inequality, we must have $\left|\hat{\mu}_{s,h}^{(1)} - \hat{\mu}_{s,h}^{(2)}\right| \leq 2\kappa/(SH)$ for all $s, h$. It then follows that the total variation distance between the marginal distributions of each coordinate of $\mathcal{B}(\hat{\mu}^{(1)})$ and $\mathcal{B}(\hat{\mu}^{(2)})$ is at most $2\kappa/(SH)$. By Lemma B.13, we thus have that the outcome $\{\mathcal{I}_h\}_{h=1}^H$ is replicable with probability at least $1 - 2\kappa$. This concludes the proof of Property (i).

**Property (ii)** The analysis of the correctness of property (ii,a) remains unchanged. We now turn to property (ii,b). Following the same argument as in Lemma E.14, we have that

$$\sum_{h \in [H]} \sqrt{\sum_{s:s \notin \mathcal{I}_h} \frac{1}{m_{s,h}}} \leq O(1)\kappa^{-1}H\sqrt{\frac{S\log(SH/\kappa)}{M}}.$$

with probability at least $1 - \kappa$. Since we now have $M \gg S^2H^2\log(SH/\kappa)\kappa^{-4}\beta^{-2}$, the above equation immediately implies that

$$\sum_{h \in [H]} \sqrt{\sum_{s:s \notin \mathcal{I}_h} \frac{1}{m_{s,h}}} \leq \frac{\kappa\beta}{\sqrt{S}}.$$

This concludes the proof of Lemma H.1. $\qquad\square$

This allows us to obtain an efficient version of **REPLEVELEXPLORE**.

**Proposition H.2** (Computationally Efficient Replicable and Strategic Exploration (Lemma E.1))**.** *Let $\zeta \in (0,1)$ and $L = \lceil \log(1/\zeta) \rceil$. Then, there is an efficient algorithm* **REPLEVELEXPLORE**$(\mathcal{M}, \zeta)$ *that consumes $K := \tilde{O}(S^3AH^7\zeta^{-2})$ episodes, runs in* poly $(K)$ *time and produces*

1. *a 0.1-replicable tiered reachability partition $\{\mathcal{S}_h^\ell\}$ of $\mathcal{M}$.*

2. *a collection of datasets $\{\mathcal{D}_{s,a,h}\}$ that are $\frac{\zeta}{\sqrt{S}}$-nice (Definition D.4) with respect to $\{\mathcal{S}_h^\ell\}$.*

*Proof.* In the efficient version of **REPLEVELEXPLORE**, we simply replace **REPEXPLORE** with its efficient version. By Lemma H.1, the total number of episodes consumed increases by a factor of $S$. The replicability analysis stays the same. The guarantees of Equation (33) and Equation (34) stay the same. For Equation (35), we get the following stronger guarantee:

$$\sum_{h=1}^H \sqrt{\sum_{s \notin \mathcal{I}_h^\ell} \frac{1}{m_{s,h,\ell}}} \leq \frac{2^\ell \zeta}{\sqrt{S}}.$$

This in turn yields the stronger guarantee that the collection of datasets are $\frac{\zeta}{\sqrt{S}}$-nice with respect to $\{\mathcal{S}_h^\ell\}$:

$$\sum_{h=1}^H \sqrt{\sum_{s \in \mathcal{S}_h^\ell} \frac{1}{m_{s,h,\ell}}} \leq \sum_{h=1}^H \sqrt{\sum_{s \notin \mathcal{I}_h^\ell} \frac{1}{m_{s,h,\ell}}} \leq \frac{2^\ell \zeta}{\sqrt{S}}.$$

The rest of the arguments are identical. This concludes the proof of Proposition H.2. $\qquad\square$

Next, we would like to develop an efficient version of **REPRLBANDIT**. As a first step, we need to make an efficient version of the key **REPVARBANDIT** sub-routine.

**Proposition H.3** (Computationally Efficient Multi-Instance Best Arm (Proposition C.6))**.** *Let $\delta \leq \rho \leq \frac{1}{3}$ and $\varepsilon \in (0,1)$. Let $\{m_s\}_{s=1}^S$ be a sequence of numbers satisfying $\sum_{s=1}^S 1/m_s \leq \frac{\rho^2\varepsilon^2}{CS(\log(3SA/\delta))^3}$, where $C$ is some sufficiently large constant. Then there is a $\rho$-replicable algorithm* **REPVARBANDIT** *that given $\left(\{\mathcal{D}_{s,a}\}_{s \in [S], a \in [A]}\right)$ where $\mathcal{D}_{s,a}$ are datasets of random size satisfying $|\mathcal{D}_{s,a}| \geq m_s$ almost surely for all $s, a$, solves the $(S, A, \varepsilon)$-multi-instance best arm problem and runs in time* poly $\left(\sum_{s=1}^S |\mathcal{D}_{s,a}|\right)$.

*Proof.* Note that the upper bound on $\sum_{s=1}^{S}(1/m_s)$ assumed in the lemma is smaller than the one assumed in Proposition C.6 by an $S$ factor. Under this stronger assumption, we show that we can construct an efficient version of Algorithm 1 that runs in time $\text{poly}\left(\sum_{s,a}|\mathcal{D}_{s,a}|\right)$. In particular, we will (1) replace the correlated sampling procedure (Line 4) with its efficient version for product distributions stated in Lemma B.13, and (2) run the randomized rounding procedure on each coordinate of $\tilde{r}$ independently with accuracy parameter $\varepsilon/2$ (Line 6). Formally speaking, we sample for each of the $S$ instances a random shift $\alpha_s \in [0,\varepsilon]$ and round $\tilde{r}_s$ to $\bar{r}_s$, the nearest $\alpha_s + k\frac{\varepsilon}{2}$ for $k \in \mathbb{Z}$.

The distribution $\hat{\mathbf{p}}$ is clearly a product distribution. So the correctness argument of the arms sampled remains valid. Following the computation in Equation (9), we still have that

$$\chi^2\left(\hat{\mathbf{p}}_s^{(1)}||\hat{\mathbf{p}}_s^{(2)}\right) \leq \frac{256\left(\log(3SA/\delta)\right)^3}{\varepsilon^2 m_s},$$

which implies that

$$\sum_s d_{\text{TV}}\left(\hat{\mathbf{p}}_s^{(1)}||\hat{\mathbf{p}}_s^{(2)}\right) \leq \frac{16\left(\log(3SA/\delta)\right)^{1.5}}{\varepsilon}\sum_s\sqrt{\frac{1}{m_s}} \qquad \text{(Lemma B.10 and summing over } s\text{)}$$

$$\leq \frac{16\left(\log(3SA/\delta)\right)^{1.5}}{\varepsilon}\sqrt{S}\sqrt{\sum_s\frac{1}{m_s}} \qquad \text{(Cauchy's Inequality)}$$

$$\leq \frac{\rho}{3},$$

where the last inequality follows from the assumption that $\sum_s(1/m_s) \leq \rho^2\varepsilon^2/(CS\left(\log(3SA/\delta)\right)^3)$ for some sufficiently large constant $C$. Therefore, the guarantees of Lemma B.13 show that the arms selected are still $\rho/3$-replicable. Lastly, we analyze the coordinate-wise rounding procedure. Note that since we have only increased the dataset sizes, we must still have $|\tilde{r}_s - r_s^*| \ll \varepsilon$. Since we run the rounding procedure with accuracy $\varepsilon/2$, it follows that $|\tilde{r}_s - \bar{r}_s| \leq \varepsilon/2$. The correctness of $\bar{r}$ then follows from the triangle inequality. We then turn to the replicability of $\bar{r}$. Denote by $\tilde{r}_s^{(1)}, \tilde{r}_s^{(2)}$ the pre-rounding estimates in two runs. We now have that

$$\left|\tilde{r}_s^{(1)} - \tilde{r}_s^{(2)}\right| \leq \sqrt{\frac{\log(3SA/\delta)}{m_s}}. \tag{51}$$

Since we sample the same shift $\alpha_s$ using shared randomness, the estimates $\tilde{r}_s$ are rounded to different $\bar{r}_s$ when there is some $k$ such that $\alpha_s + (k + \frac{1}{2})\frac{\varepsilon}{2}$ lies in $r_{s,a_s} \pm \sqrt{\log(3SA/\delta)/m_s}$. By the union bound, the probability that some $\bar{r}_s$ is non-replicable is at most

$$\sum_s\frac{O(1)}{\varepsilon}\sqrt{\frac{\log(3SA/\delta)}{m_s}} \leq O(1)\sqrt{S}\sqrt{\sum_s\frac{\log(3SA/\delta)}{\varepsilon^2 m_s}} \ll \rho,$$

where the first inequality uses Cauchy's inequality, and the second inequality follows from the assumed upper bound on $\sum_s(1/m_s)$. This concludes the proof of Proposition H.3. $\qquad\square$

We now give an efficient version of **REPRLBANDIT**.

**Theorem H.4** (Computationally Efficient Offline Policy Estimator). *Fix some MDP $\mathcal{M}$. Let $\varepsilon > 0$ and $\zeta \ll \varepsilon/(H^2(\log(SAH\varepsilon^{-1}\delta^{-1})^5)$ and $L = \lceil\log(1/\zeta)\rceil$. Let $\{\mathcal{S}_h^\ell\}_{h\in[H],\ell\in[L]}$ be a reachability state partition of $\mathcal{M}$ and $\{\mathcal{D}_{s,a,h}\}$ be a collection of random datasets that are $\zeta/\sqrt{S}$-nice with respect to $\{\mathcal{S}_h^\ell\}_{h\in[H],\ell\in[L]}$. There is a computationally efficient version of **REPRLBANDIT** that is $0.01$-replicable[19] that outputs an $\varepsilon$-optimal policy on $\mathcal{M}$ with probability $1 - \delta$. Moreover, the algorithm runs in time $\text{poly}\left(\sum_{s,a,h}|\mathcal{D}_{s,a,h}| + 1\right)$.*

---

[19]We again abuse notation and only show that **REPRLBANDIT** is replicable when $\{\mathcal{D}_{s,a,h}\}$ are promised to be $\zeta$-nice, where $\zeta$ is some sufficiently small constant multiples of $\frac{\varepsilon}{H^2\log^5(SAH/\varepsilon)}$.

*Proof.* To obtain a computationally efficient algorithm, we replace each invocation of **REPVARBANDIT** with its efficient counterpart (Proposition H.3). Examining the proof of Theorem D.5, we note that **REPRLBANDIT** replicably produces an $\varepsilon$-optimal policy as long as each invocation of **REPVARBANDIT** is replicable and correct. Consider one invocation for fixed $h \in [H]$ and $\ell \in [L]$. By the guarantee that the datasets $\{\mathcal{D}_{s,a,h}\}$ are $\zeta/\sqrt{S}$-nice (Definition D.4), we note that $\min_a |\mathcal{D}_{s,a,h}| \geq m_{s,h,\ell}$ and

$$\sum_{h \in [H]} \sqrt{\sum_{s \in \mathcal{S}_h^\ell} \frac{1}{m_{s,h,\ell}}} \leq \frac{2^\ell \zeta}{\sqrt{S}} \ll \frac{2^\ell \varepsilon}{\sqrt{S} H^2 \log^5(SAH\zeta^{-1}\delta^{-1})}.$$

Following similar arguments as Theorem D.5, if we define

$$\rho_{h,\ell} := \Theta \left( \frac{\sqrt{S} H \log^{3/2}(SAH \log(1/\zeta)\delta^{-1})}{\frac{2^\ell \varepsilon}{H \log(1/\zeta)}} \sqrt{\sum_{s \in \mathcal{S}_h^\ell} \frac{1}{m_{s,h,\ell}}} \right)$$

each invocation of Proposition H.3 is $\rho_{h,\ell}$-replicable and $\frac{\delta}{HL}$-correct. By a union bound, the output policy is not replicable with probability at most

$$\sum_{h,\ell} \rho_{h,\ell} = O \left( \frac{\sqrt{S} H \log^{3/2}(SAH \log(1/\zeta)\delta^{-1})}{\frac{2^\ell \varepsilon}{H \log(1/\zeta)}} \right) \sum_{h,\ell} \sqrt{\sum_{s \in \mathcal{S}_h^\ell} \frac{1}{m_{s,h,\ell}}}$$

$$= O \left( \frac{\sqrt{S} H^2 \log^4(SAH\zeta^{-1}\delta^{-1})}{2^\ell \varepsilon} \right) \max_\ell \sum_h \sqrt{\sum_{s \in \mathcal{S}_h^\ell} \frac{1}{m_{s,h,\ell}}}$$

$$\ll 0.01.$$

where in the first line we sum over all $\rho_{h,\ell}$, in the second we union bound over at most $L = \log(1/\zeta)$ summands $\ell$ (and rearrange terms), and in the third observe that the datasets are $\frac{\zeta}{\sqrt{S}}$-nice. Thus, the efficient version of **REPRLBANDIT** is 0.01-replicable and correct by a union bound. $\square$

Finally, we conclude with a replicable efficient algorithm for obtaining a PAC Policy Estimator.

**Theorem H.5** (Computationally Efficient Replicable PAC Policy Estimator)**.** *There is a $\rho$-replicable $(\varepsilon, \delta)$-PAC Policy Estimator with sample complexity*

$$n(S, A, H, \varepsilon, \delta) \leq \tilde{O} \left( \frac{S^3 A H^{11} \log^4(1/\delta)}{\rho^2 \varepsilon^2} \right)$$

*and run-time polynomial in sample complexity.*

*Proof.* As in Theorem 1.3, it suffices to give a computationally efficient 0.3-replicable $(\varepsilon, 0.1)$-PAC policy estimator. Since the boosting lemma (Lemma D.1) is efficient, so is the final estimator.

The procedure to obtain an efficient policy estimator is essentially identical to Theorem 1.3, except that we invoke the efficient versions of **REPLEVELEXPLORE** (Proposition H.2) and **REPRLBANDIT** (Theorem H.4). Set

$$\zeta \ll \frac{\varepsilon}{H^2 \log^5(SAH\zeta^{-1})}$$

as required by Theorem H.4. By Proposition H.2, running **REPLEVELEXPLORE** with parameter $\zeta$ on the MDP $\mathcal{M}$ produces a tiered reachability partition $\{\mathcal{S}_h^\ell\}_{h \in [H], \ell \in [L]}$, where $L = \lceil \log(1/\zeta) \rceil$, and a collection of datasets $\{\mathcal{D}_{s,a,h}\}$ that are $\zeta/\sqrt{S}$-nice (Definition D.4) with respect to $\{\mathcal{S}_h^\ell\}$. Then, we apply Theorem H.4 to obtain an $\varepsilon$-optimal policy $\hat{\pi}$. The analysis of correctness and replicability follows identically.

Finally, we bound the sample complexity of our algorithm. From Proposition H.2, our algorithm has sample complexity $\tilde{O}\left(S^3 A H^7 \zeta^{-2}\right)$. Note that our constraint on $\zeta$ is satisfied by $\zeta \leq \frac{\varepsilon}{CH^2 \text{polylog}(SAH\varepsilon^{-1})}$ for some sufficiently large constant $C > 0$. In particular, the sample complexity can be bounded as $\tilde{O}\left(\frac{S^3 A H^{11}}{\varepsilon^2}\right)$. By applying Lemma D.1, we obtain a $\rho$-replicable PAC policy estimator at the cost of an additional $\rho^{-2} \log^4(1/\delta)$ factor in sample complexity. $\square$

# I. A Lower Bound for Sign-One-Way Marginals

Recall in the sign-one-way marginals problem we are given sample access to a product of Rademacher distributions over $\{\pm 1\}^n$ with mean $p = (p_1, \ldots, p_n)$ and would like to compute $v \in [-1, 1]^n$ satisfying $\frac{1}{n} \sum_{i=1}^n v_j p_j \geq \frac{1}{n} \sum_{i=1}^n |p_j| - \varepsilon$. In this section, we present our new near-tight lower bound for sign-one-way marginals.

**Theorem G.2.** *Any $\rho$-replicable $(\varepsilon, 0.01)$-accurate algorithm for the $n$-dimensional sign-one-way marginals problem requires $\Omega\left(\frac{n}{\rho^2 \varepsilon^2 \log^6 n}\right)$ samples.*

The proof proceeds via a chain of reductions. We begin with a self-reduction to a constrained version of the problem which forces the algorithm to output some $v \in \{\pm 1\}^n$ rather than $v \in [-1, 1]^n$.

**Lemma I.1.** *Suppose there is a $\rho$-replicable algorithm that is $(\varepsilon, \delta)$-accurate for sign-one-way marginals with $m$ samples. Then, there is a $\rho$-replicable algorithm that is $(2\varepsilon, \delta)$-accurate for sign-one-way marginals with $m$ samples that always outputs $v \in \{\pm 1\}^n$.*

*Proof.* Suppose $v \in [-1, +1]^n$ is $\varepsilon$-accurate for $\mathrm{Rad}(p)$. We argue $\mathrm{sgn}(v)$ is automatically $2\varepsilon$-accurate. Thus simply outputting $\mathrm{sgn}(\mathcal{L})$ of any $\rho$-replicable, $(\varepsilon, \delta)$-accurate algorithm $\mathcal{L}$ for sign-one-way marginals gives the desired reduction.

Toward this end, note by definition is enough to prove that for every $i \in [n]$

$$(\mathrm{sgn}(p_i) - \mathrm{sgn}(v_i))p_i \leq 2(\mathrm{sgn}(p_i) - v_i)p_i.$$

We break into two cases for analysis. First, if $\mathrm{sgn}(v_i) = \mathrm{sgn}(p_i)$, observe we have $0 = (\mathrm{sgn}(p_i) - \mathrm{sgn}(v_i))p_i < (\mathrm{sgn}(p_i) - v_i)p_i$. On the other hand, if $\mathrm{sgn}(v_i) \neq \mathrm{sgn}(p_i)$, we have $(\mathrm{sgn}(p_i) - v_i)p_i \geq |p_i|$ while $(\mathrm{sgn}(p_i) - \mathrm{sgn}(v_i))p_i \leq 2|p_i|$. Combining these gives $(\mathrm{sgn}(p_i) - \mathrm{sgn}(v_i))p_i \leq 2(\mathrm{sgn}(p_i) - v_i)p_i$ as desired and completes the proof. □

Next, we show that this special class of sign-one-way marginals algorithms can be used to solve the following $\ell_\infty$-testing problem for product Rademacher distributions.

**Definition I.2** ($\ell_\infty$-Testing). Let $p \in [-1, +1]^n$. We say a vector $v \in \{\pm 1\}^n$ is an $\varepsilon$-accurate solution to the $n$-dimensional $\ell_\infty$-testing problem with respect to $\mathrm{Rad}(p)$ if $\mathrm{sgn}(p_i) = v_i$ for all $p_i \geq \varepsilon$ or equivalently

$$\max_{i \in [n]}(\mathrm{sgn}(p_i) - v_i)p_i < 2\varepsilon. \tag{52}$$

Furthermore, we say an algorithm is $(\varepsilon, \delta)$-accurate for the $n$-dimensional $\ell_\infty$-testing problem if given sample access to an arbitrary $n$-dimensional distribution $\mathrm{Rad}(p)$, the algorithm outputs an $\varepsilon$-accurate solution $\ell_\infty$-testing problem with respect to $\mathrm{Rad}(p)$ with probability at least $1 - \delta$.

Fix some product Rademacher distribution $\mathrm{Rad}(p)$. From Equations (50) and (52), it is easy to see that $v$ is $\varepsilon$-accurate to the sign-one-way marginals problem if it is $\varepsilon$-accurate to the $\ell_\infty$ testing problem. Even though the other direction is not in general true, we show certain hard instances of $\ell_\infty$ testing can be reduced to sign-one-way marginals, therefore allowing us to show lower bounds against the latter.

Formally, we say $\mathcal{L}$ is a $\rho$-replicable, $(\varepsilon, \delta)$-accurate algorithm for the $\ell_\infty$-testing problem restricted to the cube $[-\varepsilon, \varepsilon]^n$ if:

1. (Replicability) $\mathcal{L}$ is $\rho$-replicable under all distributions $\mathrm{Rad}(q)$ where $q \in [-\varepsilon, \varepsilon]^n$.

2. (Correctness) For all $q \in [-\varepsilon, \varepsilon]^n$, $\mathcal{L}$ outputs an $\varepsilon$-accurate solution to the $\ell_\infty$-testing problem over $\mathrm{Rad}(q)$ with probability at least $1 - \delta$ when given i.i.d. sample access to $\mathrm{Rad}(q)$.

Note that being $\varepsilon$-accurate is a trivial constraint whenever $q$ is in the interior of $[-\varepsilon, \varepsilon]^n$. In particular, regardless of the output $v_i$, we always have $(\mathrm{sgn}(p_i) - v_i)p_i \leq 2|p_i| < 2\varepsilon$ for all $i$.

**Lemma I.3.** *Let $\delta < 0.01$. Suppose there is a $\rho$-replicable algorithm that is $(\varepsilon, 0.01)$-accurate for sign-one-way marginals with $m$ samples. Moreover, suppose the algorithm always outputs $v \in \{\pm 1\}^n$. Then, there is a $O(\rho \log(n/\delta))$-replicable algorithm that is $(10\varepsilon, \delta)$-accurate for $\ell_\infty$-testing restricted to $[-10\varepsilon, 10\varepsilon]^n$ with $O(m \log(n/\delta))$ samples.*

Our goal is to obtain an algorithm for $\ell_\infty$ testing using an algorithm for sign-one-way marginals. The challenge is of course that sign-one-way marginals only guarantees the *average* error over coordinates is low ($\leq \varepsilon$), while $\ell_\infty$-testing requires the *maximum* error to be low (say $\leq 10\varepsilon$). Thus, our goal is to design a reduction that ensures the error of the sign-one-way marginals algorithm is evenly spread out among the coordinates. Roughly speaking, this should be possible as long as the coordinates are 'indistinguishable' to the sign-one-way marginals algorithm in some way, but there are two key issues: 1) Different coordinates may have different biases, and the algorithm may take advantage of this by focusing on coordinates having certain ranges of biases, and 2) The algorithm may intentionally treat different coordinates separately (e.g., pick a subset of coordinates arbitrarily and randomly guess their answers).

Towards the first issue, we will rely on the fact that $\ell_\infty$-testing is hard for a specific collection of distributions. Specifically, we design an $\ell_\infty$-testing algorithm over distributions $\mathrm{Rad}\,(p)$ for $p \in [-10\varepsilon, 10\varepsilon]^n$ where the algorithm only needs to output $v_i = \mathrm{sgn}(p_i)$ on the boundary $p_i \in \{\pm 10\varepsilon\}$. Thus, the biases of the coordinates requiring correctness are indeed indistinguishable: no algorithm can distinguish between coordinates with bias exactly $10\varepsilon$ (resp., $-10\varepsilon$).

Towards the second issue, we randomly permute the coordinates before giving samples to the sign-one-way marginals algorithm. Even if the sign-one-way algorithm treats different coordinates differently, the permutation ensures different input coordinates will be treated the same *on average* in the final algorithm, ensuring low average error for our $\ell_\infty$-testing algorithm.

Even with all the coordinates indistinguishable, there remains one last obstacle. Consider $q \in [-10\varepsilon, 10\varepsilon]^n$ with exactly one boundary coordinate $q_i \in \{\pm 10\varepsilon\}$. A sign-one-way marginals algorithm $\mathcal{L}$ could err on the $i$-th coordinate as long as it performs well on other coordinates. To circumvent this issue, we artificially add coordinates with bias $10\varepsilon$, to ensure that most coordinates with bias $10\varepsilon$ must be estimated accurately.

*Proof of Lemma I.3.* We begin with some preliminaries and notation. Let $\mathcal{L}$ denote the $\rho$-replicable algorithm for sign-one-way marginals. We construct an algorithm for $\ell_\infty$-testing. Suppose that we are given sample access to $\mathrm{Rad}\,(q)$ where $q \in [-1, +1]^n$. Let $\Pi(q) \subset [-1, +1]^n$ denote the orbit of $q$ under $S_n$, that is the set of vectors $q'$ such that there exists some permutation $\pi$ where $q' = \pi(q)$. In particular, $\Pi(q)$ is the set of vectors that are equivalent to $q$ up to coordinate permutation.

**Sample Simulation**   For convenience, denote by $\bar{q}$ the 'padded' $3n$-dimensional vector

$$\bar{q}_i = \begin{cases} q_i & 1 \leq i \leq n \\ 10\varepsilon & n+1 \leq i \leq 2n \\ -10\varepsilon & 2n+1 \leq i \leq 3n \end{cases} . \tag{53}$$

Given sample access to $\mathrm{Rad}\,(q)$ and an arbitrary $\pi \in S_{3n}$, we note that one can easily simulate sample access to $\mathrm{Rad}\,(\pi(\bar{q}))$.

**The Reduction.**   Recall that $\mathcal{L}$ is a $\rho$-replicable $(\varepsilon, 0.01)$-accurate algorithm. Fix $T = O(\log(n/\delta))$. We design our $(T\rho)$-replicable $(10\varepsilon, (T+1)\delta)$-accurate algorithm for $\ell_\infty$ testing, see Algorithm 7 for pseudocode.

**Analysis: Sample Complexity and Replicability.**   The sample complexity of Algorithm 7 is $Tm = O\,(m\log(n/\delta))$. We argue that Algorithm 7 is $O\,(\rho\log(n/\delta))$-replicable. Note that we invoke the sign-one-way marginals algorithm $\mathcal{L}$ at most $T$ times (each time with independently sampled shared randomness $r_t$). Furthermore, since the permutations $\pi_t$ are sampled using shared randomness, the $t$-th invocation of $\mathcal{L}$ in the two different runs samples from the same distribution $\mathrm{Rad}\,(\pi_t(\bar{q}))$. The conclusion then follows from a union bound and the fact that $\mathcal{L}$ is $\rho$-replicable.

**Analysis: Correctness.**   Finally, we argue for correctness. Let $q \in [-10\varepsilon, 10\varepsilon]^n$ denote the mean of the target distribution. For convenience, we define $p^{(t)} := \pi_t(\bar{q})$. We condition on the event that for at least $0.98$-fraction of $t$, $v^{(t)}$ is an $\varepsilon$-accurate solution to the sign-one-way marginals problem with respect to $\mathrm{Rad}\,(p^{(t)})$. By a Chernoff bound, this occurs with probability at least $e^{-\Omega(T)} < \frac{\delta}{3}$ for $T = O(\log(1/\delta))$. On the remaining $0.02T$ outputs, $v^{(t)}$ may incur catastrophic error (up to $10\varepsilon n$). In particular, for all accurate $t \in [T]$, since we have assumed $v^{(t)} \in \{\pm 1\}^{3n}$, it follows that

$$2\sum_{i=1}^{3n} \mathbb{1}\left[v_i^{(t)} \neq \mathrm{sgn}(p_i^{(t)})\right] \left|p_i^{(t)}\right| = \sum_{i=1}^{3n} \left(\mathrm{sgn}(p_i^{(t)}) - v_i^{(t)}\right) p_i^{(t)} \leq \varepsilon n.$$

**Algorithm 7** REPINFTYESTIMATE

---

**Input:** Samples drawn i.i.d. from $\mathrm{Rad}\,(q)$ with $q \in [-10\varepsilon, 10\varepsilon]^n$. Oracle access to $\rho$-replicable, $(\varepsilon, 0.01)$-accurate algorithm $\mathcal{L}$ for the $(3n)$-dimensional sign-one-way marginals problem constrained to output $\{\pm 1\}^{3n}$ with sample complexity $m$.

**Output:** Labels $v \in \{\pm 1\}^n$.

**Parameters:** Accuracy $\varepsilon$, error $\delta$, replicability $\rho$.

1. Fix $T = O(\log(n/\delta))$ for some sufficiently large constant.

2. For $t \in [T]$:

    (a) Independently sample $\pi_t \in S_{3n}$.
    (b) Let $\overline{q}$ be defined as in Equation (53). Generate a sample set $D_t \sim \mathrm{Rad}\,(\pi_t(\overline{q}))^{\otimes m}$.
    (c) Compute $v^{(t)} \leftarrow \mathcal{L}(S_t; r_t)$, where $r_t$ is some fresh random seed.

3. For $i \in [n]$, output $v_i \leftarrow \mathrm{maj}_{t \in [T]} \pi_t^{-1}\left(v^{(t)}\right)_i$.

---

Let $I_+^{(t)} := \{i \in [3n] \text{ s.t. } p_i^{(t)} = 10\varepsilon\}$ and $I_-^{(t)} := \{i \in [3n] \text{ s.t. } p_i^{(t)} = -10\varepsilon\}$ denote the indices where $p^{(t)}$ has absolute value exactly $10\varepsilon$ (i.e., indices with maximum absolute value). Let $R^{(t)} = [3n] \setminus (I_+^{(t)} \cup I_-^{(t)})$ denote the remaining indices. Then, we can rewrite the error as

$$2 \left( \sum_{i \in I_+^{(t)}} \mathbb{1}\left[v_i^{(t)} \neq \mathrm{sgn}(p_i^{(t)})\right] \left|p_i^{(t)}\right| + \sum_{i \in I_-^{(t)}} \mathbb{1}\left[v_i^{(t)} \neq \mathrm{sgn}(p_i^{(t)})\right] \left|p_i^{(t)}\right| + \sum_{i \in R^{(t)}} \mathbb{1}\left[v_i^{(t)} \neq \mathrm{sgn}(p_i^{(t)})\right] \left|p_i^{(t)}\right| \right).$$

Consider the first term. By correctness we have that

$$\sum_{i \in I_+^{(t)}} \mathbb{1}\left[v_i^{(t)} \neq \mathrm{sgn}(p_i^{(t)})\right] \left|p_i^{(t)}\right| = 10\varepsilon \sum_{i \in I_+^{(t)}} \mathbb{1}\left[v_i^{(t)} \neq \mathrm{sgn}(p_i^{(t)})\right] \leq \frac{\varepsilon}{2}n. \tag{54}$$

Let $E_+^{(t)} := \{i \in I_+^{(t)} \text{ s.t. } v_i^{(t)} \neq \mathrm{sgn}(p_i^{(t)})\}$ be the subset of error coordinates within $I_+^{(t)}$. Rearranging Equation (54) immediately yields

$$|E_+^{(t)}| \leq \frac{n}{20}.$$

Since $|I_+^{(t)}| \geq n$ by construction (see Equation (53)), we have $\frac{|E_+^{(t)}|}{|I_+^{(t)}|} \leq \frac{n/20}{n} = \frac{1}{20}$. Similarly, we can define $E_-^{(t)} := \{i \in I_-^{(t)} \text{ s.t. } v_i^{(t)} \neq \mathrm{sgn}(p_i^{(t)})\}$ and use analogous arguments to show that $\frac{E_-^{(t)}}{|I_-^{(t)}|} \leq \frac{1}{20}$.

Fix $i \in [n]$ with $|q_i| = 10\varepsilon$. If no such $i$ exists, then any output $v$ is $10\varepsilon$-accurate on $q$. Suppose without loss of generality that $q_i = 10\varepsilon$. Fix $t \in [T]$. Note that

$$\mathrm{sgn}(v_{\pi_t(i)}^{(t)}) \neq \mathrm{sgn}(q_i) \iff \mathrm{sgn}(v_{\pi_t(i)}^{(t)}) \neq \mathrm{sgn}(p_{\pi_t(i)}^{(t)}) \iff \pi_t(i) \in E_+^{(t)}.$$

By symmetry, even conditioned on the event $\mathcal{E}^{(t)}$ that $v^{(t)}$ is an accurate solution for $p^{(t)}$, we have that $\pi_t(i)$ is uniformly distributed within $I_+^{(t)}$.[20] In particular, we claim

$$\Pr\left(\mathrm{sgn}(v_{\pi_t(i)}^{(t)}) \neq \mathrm{sgn}(q_i)|\mathcal{E}^{(t)}\right) = \Pr\left(\pi_t(i) \in E_+^{(t)}|\mathcal{E}^{(t)}\right) \leq \frac{1}{20},$$

where the final inequality follows due to indistinguishability of coordinates within $I_+^{(t)}$. More formally, we can break the above probability into an expectation over permutations $\pi_t$ conditioned on 1) a fixed setting of $I_+^{(t)}$, and 2) a fixed mapping

---

[20]To be slightly more formal, Algorithm 7 generates a joint distribution over $(\pi_t, I_+^{(t)})$. We mean that for any fixed $i \in I_+^{(t)}$ and instantiation $I_+^{(t)} = S$, the conditional distribution on $\pi_t(i)$ is uniform over $S$.

of $\pi_t(j)$ for every $j \notin I_+^{(t)}$. Under every such conditioning the input to sign-one-way marginals is indistinguishable, so the set of errors $E_+^{(t)} \subset I_+^{(t)}$ is independent of $\pi_t$ and the desired inequality holds.

Since we output $v_i = \mathrm{maj}_t v_{\pi_t(i)}^{(t)}$ we have $\mathrm{sgn}(v_i) \neq \mathrm{sgn}(q_i)$ if more than half of $v_{\pi_t(i)}^{(t)}$ disagree with $\mathrm{sgn}(q_i)$. We bound the probability that this occurs. First, note that at most 0.02-fraction of $v^{(t)}$ incur catastrophic error (that is the sign-one-way marginals algorithm fails). Then, among the iterations where the sign-one-way marginals algorithm $\mathcal{L}$ succeeds, a standard Chernoff bound implies that $\pi_t(i) \in E_+^{(t)}$ in more than 0.1-fraction of $t \in [T]$ with probability at most $\exp(-T/20) < \frac{\delta}{2n}$ for $T = O(\log(n/\delta))$ for some sufficiently large constant. Thus, at most a 0.12-fraction of $v^{(t)}$ output the wrong label on the $i$-th coordinate with probability at least $1 - \delta/n$, far below the majority threshold. Applying a union bound over all $n$ coordinates, we conclude that all outputs are correct on coordinates $i$ with $|q_i| = 10\varepsilon$ with probability at least $1 - \delta$, as desired. $\qquad\square$

Finally, we conclude with a lower bound for the $\ell_\infty$ testing problem.

**Lemma I.4.** *Any $\rho$-replicable algorithm that is $(\varepsilon, 0.1)$-accurate for $\ell_\infty$-testing restricted to $[-\varepsilon, \varepsilon]^n$ requires $\Omega\left(\frac{n}{\rho^2 \varepsilon^2 \log^3 n}\right)$ samples.*

This follows via a simple reduction to the $n$-coin problem.

**Definition I.5** ($n$-coin problem). Let $\mathbf{p}$ be a product of Bernoulli random variables with parameter $p \in [0, 1]^n$. A vector $v \in \{0, 1\}^n$ is an $\varepsilon$-accurate solution to the $n$-coin problem over $\mathbf{p} \sim \mathrm{Bern}(p)$ if for all $i \in [n]$:

1. If $p_i \geq \frac{1}{2} + \varepsilon$, then $v_i = 1$.

2. If $p_i \leq \frac{1}{2} - \varepsilon$, then $v_i = 0$.

We are particularly interested in a special collection of inputs which capture the hardness of the $n$-coin problem. Formally, we say $\mathcal{L}$ is a $\rho$-replicable $(\varepsilon, \delta)$-accurate for the $n$-coin problem restricted to $\left[\frac{1}{2} - \varepsilon, \frac{1}{2} + \varepsilon\right]^n$ if:

1. (Replicability) $\mathcal{L}$ is $\rho$-replicable for all distributions $\mathrm{Bern}(p)$ where $p \in \left[\frac{1}{2} - \varepsilon, \frac{1}{2} + \varepsilon\right]^n$.

2. (Correctness) For all $p \in \left[\frac{1}{2} - \varepsilon, \frac{1}{2} + \varepsilon\right]^n$, $\mathcal{L}$ outputs an $\varepsilon$-accurate solution to the $n$-coin problem over $\mathrm{Bern}(p)$ with probability at least $1 - \delta$ when given i.i.d. sample access to $\mathrm{Bern}(p)$.

As before, the correctness condition is trivial on the interior (neither of the correctness conditions are triggered). (Hopkins et al., 2024) gives a lower bound for this problem.

**Theorem I.6** (Theorem 4.46 of (Hopkins et al., 2024)). *Any $\rho$-replicable $(\varepsilon, 0.01)$-accurate algorithm for the $n$-coin problem restricted to $\left[\frac{1}{2} - \varepsilon, \frac{1}{2} + \varepsilon\right]^n$ requires $\Omega\left(\frac{n}{\rho^2 \varepsilon^2 \log^3 n}\right)$ samples.*

Formally, (Hopkins et al., 2024) show that $\tilde{\Omega}(n^2)$ samples are necessary for algorithms that sample one coordinate at a time. Since any algorithm that makes $m$ (vector) samples makes at most $nm$ (coordinate) samples, this implies the above lower bound.

We are now ready to prove a lower bound for $\ell_\infty$-testing.

*Proof of Lemma I.4.* This immediately follows from the sample complexity lower bound for the $n$-coin problem (Hopkins et al., 2024). For completeness, we sketch a simple reduction for $\ell_\infty$-testing of Rademacher random variables to the $n$-coin problem. The crucial point here is that the hard instance given in Theorem I.6 is precisely the hard instance for which we have obtained an $\ell_\infty$-testing algorithm.

If $i \in (0.5 - \varepsilon, 0.5 + \varepsilon)$, any output is correct. Note that $\mathrm{Bern}(p)$ directly simulates $\mathrm{Rad}(2p - 1)$ so the $n$-coin problem is equivalent to $\ell_\infty$-testing with accuracy $2\varepsilon$. In particular, $\mathrm{Bern}(0.5 - \varepsilon)$ is equivalent to $\mathrm{Rad}(-2\varepsilon)$ while $\mathrm{Bern}(0.5 + \varepsilon)$ is equivalent to $\mathrm{Rad}(2\varepsilon)$. Thus, the $\ell_\infty$-testing problem over $q \in [-2\varepsilon, 2\varepsilon]^n$ is equivalent to the $n$-coin problem over $p \in [0.5 - \varepsilon, 0.5 + \varepsilon]^n$.

Thus, any $\rho$-replicable algorithm that is $(\varepsilon, \delta)$-accurate for $\ell_\infty$-testing over distribution $\mathrm{Rad}\,(q)$ with $q \in [-\varepsilon, \varepsilon]^n$ is a $\rho$-replicable $(\varepsilon/2, \delta)$-accurate algorithm for the $n$-coin problem over distributions $\mathrm{Bern}\,(p)$ with $p \in \left[\frac{1-\varepsilon}{2}, \frac{1+\varepsilon}{2}\right]^n$. The lower bound follows. $\qquad \square$

We are now ready to prove Theorem G.2.

*Proof of Theorem G.2.* Suppose we have a $\rho$-replicable algorithm that is $(\varepsilon, 0.01)$-accurate for sign-one-way marginals with sample complexity $m$. Using Lemma I.1, we obtain a $\rho$-replicable $(2\varepsilon, 0.01)$-accurate algorithm that always outputs $\{\pm 1\}^n$. Using Lemma I.3 for any $\delta > 0$, we obtain a $(\rho \log(n/\delta))$-replicable $(20\varepsilon, \delta)$-accurate algorithm for a specific hard instance of $\ell_\infty$-testing which requires $\Omega\left(\frac{n}{\rho^2 \varepsilon^2 \log^5(n/\delta)}\right)$ samples by Lemma I.4. The algorithm we obtained has sample complexity $O(m \log(n/\delta))$.

Note that Lemma I.4 requires $\delta < 0.01$ so it suffices to set $\delta$ to some sufficiently small constant. Then, we obtain the lower bound

$$\Omega\left(\frac{n}{\rho^2 \varepsilon^2 \log^6 n}\right)$$

as desired. $\qquad \square$

## I.1. Applications of Sign-One-Way Marginals

The sign-one-way marginals problem captures the statistical difficulty of many fundamental learning problems. We give two such examples below. First, we obtain an essentially tight lower bound on $\ell_1$ mean estimation, extending the sample complexity lower bounds of (Hopkins et al., 2024). Note that a matching upper bound can be obtained from an algorithm for $\ell_\infty$-mean estimation (see one-way-marginals in (Bun et al., 2023)).

**Definition I.7.** Let $\mathbf{p}$ be an arbitrary distribution over $[-1, +1]^n$ with mean $\mu$. A vector $v$ is an $\varepsilon$-accurate solution to the $\ell_1$-mean estimation problem over $\mathbf{p}$ if $||v - \mu||_1 \leq \varepsilon$.

**Theorem I.8.** *Any $\rho$-replicable $(\varepsilon, 0.001)$-accurate algorithm for $\ell_1$-mean estimation requires $\Omega\left(\frac{n}{\rho^2 \varepsilon^2 \log^6 n}\right)$ samples.*

*Proof.* We proceed via reduction to sign-one-way marginals problem. Suppose we have a $\rho$-replicable $(\varepsilon, \delta)$-accurate algorithm $\mathcal{L}$ for $\ell_1$-mean estimation with sample complexity $m$. Given samples $S \sim \mathrm{Rad}\,(\mu)^{\otimes m}$, compute $\hat{v} \leftarrow \mathcal{L}(S; r)$ and return $v \in \{\pm 1\}^n$ where $v_i = \mathrm{sgn}(\hat{v}_i)$. Clearly this algorithm has sample complexity $m$ and is $\rho$-replicable. We analyze the error of the output $v$.

Suppose $\hat{v}$ is an $\varepsilon$-accurate solution to the $\ell_1$-mean estimation problem. Then our goal is to bound the error function below.

$$2 \sum_i \mathbb{1}\left[v_i \neq \mathrm{sgn}(\mu_i)\right] |\mu_i|.$$

Note that if $v_i \neq \mathrm{sgn}(\mu_i)$, we have that $\mathrm{sgn}(\hat{v}_i) \neq \mathrm{sgn}(\mu_i)$, i.e. $|\mu_i| \leq |\mu_i - \hat{v}_i|$. Summing over all such $i$ gives

$$2 \sum_i \mathbb{1}\left[v_i \neq \mathrm{sgn}(\mu_i)\right] |\mu_i| = 2 \sum_{\mathrm{sgn}(\hat{v}_i) \neq \mathrm{sgn}(\mu_i)} |\mu_i| \leq 2 \sum_{\mathrm{sgn}(\hat{v}_i) \neq \mathrm{sgn}(\mu_i)} |\mu_i - \hat{v}_i| \leq 2\varepsilon.$$

Thus, our algorithm is $2\varepsilon$-accurate. The lower bound then follows from Theorem G.2. $\qquad \square$

Finally, we observe our new lower bound for sign-one-way marginals essentially resolves the sample complexity of agnostic PAC learning finite hypothesis classes. In the celebrated PAC learning model (Vapnik et al., 1974; Valiant, 1984), a learner $\mathcal{L}$ is given access to labeled samples $(x, y) \in \mathcal{X} \times \mathcal{Y}$ and asked to learn a function $h : \mathcal{X} \rightarrow \mathcal{Y}$. $\mathcal{L}$ is an $(\varepsilon, \delta)$-accurate (agnostic) learner for a hypothesis class $\mathcal{H} : \mathcal{X} \rightarrow \mathcal{Y}$ if given samples access to any distribution $\mathbf{p}$ over $\mathcal{X} \times \mathcal{Y}$, $\mathcal{L}$ outputs with probability $1 - \delta$ a hypothesis $h$ satisfying

$$\mathrm{err}(h) \leq \inf_{f \in \mathcal{H}} \mathrm{err}(f) + \varepsilon$$

where $\mathrm{err}(h) := \mathrm{Pr}_{\mathbf{p}}(h(x) \neq y)$ is the expected error of a hypothesis on a random sample. Note that the error probability is over the randomness of the observed samples and the internal randomness of the algorithm $\mathcal{L}$.

In the standard setting (without replicability) the complexity of a hypothesis class $\mathcal{H}$ is characterized by its VC dimension $d$ (Vapnik et al., 1974). In particular, $\Theta(\frac{d}{\varepsilon^2})$ samples are necessary and sufficient to learn a hypothesis class with VC dimension $d$. However, it is provably more expensive to learn *replicably*. In (Bun et al., 2023), the authors show via a reduction to sign-one-way marginals that any replicable learner for a hypothesis class with VC dimension $d$ requires $\tilde{\Omega}(d^2)$ samples (Bun et al., 2023). Combining Theorem G.2 and the reduction from PAC learning to sign-one-way marginals of (Bun et al., 2023), we obtain a stronger lower bound with the optimal $\varepsilon$ and $\rho$ dependence:

**Theorem I.9.** *Fix sufficiently large $d$ and a hypothesis class $\mathcal{H}$ with VC dimension $d$. Any $\rho$-replicable $(\varepsilon, 0.001)$-accurate agnostic learner for $\mathcal{H}$ requires $\tilde{\Omega}\left(\frac{d^2}{\rho^2 \varepsilon^2}\right)$ samples.*

For finite binary hypothesis classes $\mathcal{H}$ with VC dimension $\log |\mathcal{H}|$, (Bun et al., 2023) give an agnostic learner using $\tilde{O}(\frac{\log^2 |\mathcal{H}|}{\rho^2 \varepsilon^2})$ samples and a lower bound of $\tilde{\Omega}(\log^2 |\mathcal{H}|)$ samples, leaving open the optimal dependence on $\rho, \varepsilon$. While quadratic dependence on $\rho, \varepsilon$ is easily shown to be necessary as well (e.g. via reduction to bias estimation), a priori this dependence may have occurred *additively* rather than *multiplicatively*, that is it did not rule out an algorithm on say $O\left(\log^2 |\mathcal{H}| + \frac{1}{\rho^2 \varepsilon^2} + \frac{\log |\mathcal{H}|}{\varepsilon^2}\right)$ samples (indeed this type of behavior does occur in related problems, e.g., uniformity testing (Liu & Ye, 2024)). We now rule out this possibility, resolving the optimal sample complexity of replicably learning finite classes up to logarithmic factors.

