# OpenReview forum: "From Generative to Episodic: Sample-Efficient Replicable Reinforcement Learning"
_ICML.cc/2026/Conference — ICML 2026 regular_

### Official Review · Reviewer_5Day · 2026-03-13

**Soundness:** 3
**Presentation:** 1
**Significance:** 3
**Originality:** 3
**Overall Recommendation:** 4
**Confidence:** 2

**Summary:**

This paper studies replicable reinforcement learning (RL), where replicability means that the output policy must remain the same across different executions with high probability when the internal randomness is fixed. In particular, in the episodic setting without assuming a generative model, the paper proposes a replicable algorithm whose sample complexity bound is $\tilde O(S^2 A / \epsilon^2)$, improving upon previous work by a factor $\tilde O(S^7 A^7/ \epsilon^2)$. Moreover, in this setting, the papers presents a lower bound of $\tilde \Omega(S^2)$. The key technique is the exploration phase that allows the agent to identify ignorable states. On top of this technique, the reward-free algorithm is applied to obtain a policy.

**Compliance With Llm Reviewing Policy:**

Affirmed.

**Final Justification:**

The main concern was about presentation. While it is still awkward that the main text ends with Section 1.6, this is expected to be addressed in the camera-ready version. Therefore, I will increase my score.

**Key Questions For Authors:**

1. Could you compare the proposed algorithm with the preivous replicable RL algorithms ([Eaton et al.] and [Karbasi et al.]) in more details, while some part of comparision is in Section 1.5?

2. Could you explain the intuition on the reachability function $R_h^k$?

**Limitations:**

yes

**Strengths And Weaknesses:**

**Strength**
1. Without assuming a generative model, this paper achieves a sample complexity of $\tilde O(\frac{S^2A \ poly(H,\ \log(1/\delta))}{\rho^2\epsilon^2})$ for episodic RL. In the literature of replicable learning, replicable RL is more challenging due to the presence of exploration.
2. In the same setting, this paper presents a lower bound of $\Omega(S^2/\epsilon^2)$. This implies that the proposed algoirthm mathces the lower bound in terms of $S$.
3. While partitioning states into two sets according to their importance is a known approach in RL, but its replicable variant is non-trivial, which is the main technique of this paper.

**Weakness**
1. The paper is unusual in structure and is not well written. The main text only includes Introduction section, and there is no method or algorithm section. The paper is hard to read. It would be better to formally define some notions (e.g., correlated sampling, $\mathcal U_h$, $\mathcal B(\mu)$, $\mu_{s,h}$, $R_h^k$, etc.) and introduce the proposed algorithm in the main text. Currently, these are deferred to the appendix.

2. The main algorithm whose sample complexity is $\tilde O(S^2A/\epsilon^2)$ is not computationally efficient. Nevertheless, the paper proposes a computationally efficient algorithm with additional a $S$ multiplicative factor.

3. The paper lacks numerical experiments validating its main contributions.

Refs

[Eaton et al.] Replicable Reinforcement Learning, NeurIPS 2023.

[Karbasi et al.] Replicability in reinforcement learning, NeurIPS 2023.

---

> ### Author Rebuttal · Authors · 2026-03-31
>
> We thank the reviewer for their careful reading and their positive feedback regarding the optimality of our results, and novel insights into replicable exploration and RL. We appreciate their thoughtful comments and address specific weaknesses and questions below.
>
> **Presentation**
> Regarding the structure of the paper, the presentation format we chose—focusing the main text on the conceptual framework, problem formulation, and the statement of main theorems while deferring dense technical proofs to the appendix—is the standard practice for theory-heavy submissions at ICML, NeurIPS, and COLT. Given the 9-page limit and the mathematical depth required to prove theorems regarding replicability, moving the fairly involved implementation details of the algorithm to the appendix allows for a more rigorous discussion of the proof techniques and the underlying theory in the main body. Nevertheless, we appreciate the feedback and will take the reviewer's advice to move some of the high-level formal definitions, such as correlated sampling, into the main text to improve immediate readability.
>
> **Computational Efficiency**
> While our main sample-optimal algorithm is not efficient, even for the significantly simpler task of mean estimation (e.g. estimating the mean of an S-dimensional distribution), there are no known efficient replicable algorithms with sample complexity $o(S^3)$. Indeed, there are concrete barriers to obtaining sub-cubic sample complexity with efficient algorithms (e.g. [HIK+24] demonstrates any such algorithm would lead to a breakthrough in computational geometry). Furthermore, we emphasize that regardless of computational efficiency, no previous algorithm achieves sample complexity $o(S^7)$.
>
> **Numerical Experiments**
> We emphasize that our work is primarily a theoretical contribution, presenting near-optimal sample bounds for a core problem in the study of replicable learning that significantly improve both computationally and statistically over prior works (e.g. Eaton et al.), and require substantial new ideas.
>
> Many influential theory works in NeurIPS/ICML/etc, including sample-bounds in RL theory (e.g., [JKMY20]), do not have corresponding numerical experiments, and while implementing and practically optimizing these algorithms is certainly an important and worthwhile direction for future work, the theory alone already provides substantial value.
>
> **Comparison with Eaton et al. and Karbasi et al.**
> In the generative setting, Karbasi et al. replicably estimate the value of every state-action pair, while our algorithm instead replicably finds the optimal action for each state. We give a more efficient algorithm for this task, thus saving a factor of A in sample complexity.
>
> In the episodic setting, (which Karbasi et al. do not address), Eaton et al. similarly attempts to replicably learn all transition probabilities. This turns out to be extremely sample and computationally intensive, in large part contributing to their $S^7 A^7$ sample complexity. Our algorithm, again using the observation that we only want the output policy to be replicable, is significantly more efficient. In particular, our novel exploration procedure for replicably determining reachability and collecting samples from reachable states shows that a policy can be learned replicably in $S^2 A$ samples.
>
> **Reachability Function**
> The reachability function $R_h^k(s)$ intuitively represents the maximum probability, across all policies, of an agent reaching an "under-explored" state from a given state $s$ at time step $h$ during episode $k$. Conceptually, it is mathematically equivalent to a value function in a modified MDP where the agent receives a unit reward of 1 only for visiting any under-explored state and 0 otherwise. During our exploration phase, we run a $Q$-learning agent on a simulated environment where all actual rewards are explicitly set to 0. Since $R_h^k(s)$ behaves exactly like a value function, we can formally prove that our optimistic $Q$-value bounds from above this reachability function. Since the true rewards of this simulated MDP are 0, the algorithm's optimistic estimates naturally decay to 0 as exploration progresses. This cleanly guarantees that the probability of reaching any remaining under-explored state under any policy also shrinks to 0, proving these remaining states are strictly "ignorable".
>
> References
>
> [HIK+24] Replicability in High Dimensional Statistics, FOCS 2024
> [JKMY20] Reward-Free Exploration for Reinforcement Learning, ICML ‘20

---

> > ### Author Rebuttal · Reviewer_5Day · 2026-04-02
> >
> > Thanks for the author's response.
> >
> > To clarify for the presentation, I am fully aware that theory-heavy papers commonly defer the details to the appendix. However, it is still awkward that the main text ends with Section 1.6. In the main text, it would be great to provide proper sectioning and definitions to help readers' understanding. As this is expected to be addressed in the camera-ready version, I will increase my score.

---

### Official Review · Reviewer_Ew9n · 2026-03-13

**Soundness:** 3
**Presentation:** 3
**Significance:** 4
**Originality:** 3
**Overall Recommendation:** 5
**Confidence:** 4

**Summary:**

This paper nearly resolves the sample complexity of replicable RL in tabular MDPs, proving that exploration is not a significant challenge to replicability by achieving an $O(S^2A)$ sample algorithm that bridges the generative and episodic settings.

**Compliance With Llm Reviewing Policy:**

Affirmed.

**Key Questions For Authors:**

1. It seems like the dependency on $H$ is merely an artifact of stacking algorithms together. Is there insights on if this dependency can be avoidable or unavoidable?

2. Lemma E.10 shows that sampling from $B(\hat{\mu})$ preserves the expected reachability, and then applies Markov's inequality to get a high-probability bound. This Markov step introduces a constant-factor slack. In the final algorithm this slack propagates through the tier structure. Is there a sharper concentration argument (e.g., using the specific product structure of  $B(\hat{\mu})$ that could tighten this, and would it improve the $H$ dependence?

3. The ignorable state combination guarantees that $\sum_h \mathbb{P}[x_h \in I_h \mid \pi^\ast] \ll \varepsilon/H$ for some optimal policy $\pi^\ast$ . In MDPs with multiple optimal policies, different optimal policies may have very different reachability structures. Does the algorithm's guarantee degrade if the MDP has many near-optimal policies with very different reachability patterns, and is there a version of the guarantee that holds simultaneously for all near-optimal policies?

4. The paper's replicability definition requires the same policy to be output across two runs. This is a very strong requirement that might not be needed for behavioral reproducibility. Is there a separation result showing that replicability of the policy requires strictly more samples than replicability of the value or the induced trajectory distribution?

**Limitations:**

Yes

**Strengths And Weaknesses:**

Strength:
I think the paper gave a nice insight about the relationship between replicability and exploration, i.e. you do not need the exploration trajectories to be replicable, you only need the output of exploration to be replicable. This might have further impact beyond tabular RL, in settings where direct replicability of a stochastic process is expensive or impossible.

Weakness:
1. The order on $H$ is very high, unlike the typical $H^3$ dependency on non-replicable minimax bound.
2. Requiring identical policy outputs across two runs is a very strong stability notion. For RL specifically, one might care about replicability of value or behavior rather than exact policy identity, especially in MDPs where multiple optimal policies exist.

---

> ### Author Rebuttal · Authors · 2026-03-31
>
> We thank the reviewer for their careful reading and their positive feedback regarding our results, and novel insights into replicable exploration and RL. We appreciate their thoughtful comments and address specific weaknesses and questions below.
>
> **Horizon Dependence Barriers**
> Improving the sample dependency on $H$ is an important and challenging direction, and one that took many years to resolve even in the non-replicable setting. Previous work in replicable RL [EHKS23] incurs similarly poor dependence on $H$ to our methods ($H^{10} \varepsilon^{-10}$) sample complexity in addition to the $S^{7} A^{7}$ dependence which we improve significantly upon. We focus in this work on improving the latter dependence on the state space $S$, the action space $A$, and the approximation factor $\varepsilon$ to near-optimal. Given that our submission already presents significantly improved results that require substantially new ideas and technical effort, we believe it is reasonable to defer improvements in the horizon, which also likely require substantially new ideas, to future work. We will modify the text to better emphasize the importance of this direction.
>
> **Improvement on Lemma E.10**
> We thank the reviewer for the insightful question. While employing a sharper concentration bound in Lemma E.10 would tighten the analysis, it would only bring polylogarithmic improvements (as the number of tiers) but would not improve the polynomial dependence on $H$. The heavy dependency on $H$ fundamentally stems from the structural requirements of the exploration and backward induction phases. Specifically, the exploration subroutine requires $\tilde{O}(S^2 A H^7 \zeta^{-2})$ samples, where $S$ is the number of states, $A$ is the number of actions, and $\zeta$ is the niceness parameter dictating dataset sizes. To guarantee success in the backward induction step, $\zeta$ must scale as $O(H^{-2})$. Squaring this inverse introduces an $H^4$ factor that multiplies with the baseline $H^7$ to yield the $H^{11}$ dependency. The slack from Markov's inequality is isolated entirely within the failure probability parameter $\kappa$, which is defined as $\Omega(\log^{-1}(1/\zeta))$. Because $\zeta$ scales polynomially with $H^{-1}$, the parameter $\kappa$ depends only logarithmically on $H$. Consequently, upgrading to a tighter Hoeffding's or Bernstein’s inequality would only shave off polylogarithmic factors, leaving the core polynomial dependence on $H$ largely unchanged.
>
> **MDPs with Multiple Optimal Policies**
> Thank you for this insightful question. The performance of our algorithm does not degrade when there are many near-optimal policies with very different reachability patterns. As established in our analysis, the distribution $\mathcal{B}(\hat{\mu})$ preserves the expected reachability for any arbitrary policy. However, after rounding the fractional solution, the resulting discrete state combination may preserve the reachability structure for some policies but fail to do so for others, an artifact of applying Markov's inequality. Fortunately, universal preservation is not required for performance. Our downstream arguments work perfectly as long as the reachability is preserved with respect to any single fixed near-optimal policy. While it is theoretically possible to enhance the reachability guarantee so that the rounded partition preserves reachability simultaneously with respect to all policies, achieving this universal guarantee blows-up the sample complexity's dependency on the state space size, $S$. Since the primary objective of our work is to avoid this and achieve near-optimal sample efficiency—specifically $\tilde{O}(S^2A)$ samples—we deliberately avoided this approach.
>
> **Behavioral Replicability**
> While replicability of the output policy/trajectory distribution is a stringent one, it is a well-motivated definition with many desirable properties and important connections to other stability notions in the literature such as differential privacy (see response to Reviewer y52N for further motivation). For replicable value estimation, there is indeed a separation. Value estimation has a simple replicable algorithm: replicably estimate the mean of a (non-replicable) policy estimator with $SA/\rho^2$ samples. Our lower bound of $S^2$ samples establishes a strong separation between value replicability and policy replicability.
>
> Regarding trajectory replicability, we suspect that replicating the exact trajectory distribution is likely as difficult as policy replicability, since different policies likely produce distinct trajectory distributions. If you meant only to enforce some sort of closeness in trajectory distribution, this is indeed an interesting question: such approximate notions are often statistically separated from exact replicability, though it may depend on the notion of "closeness" and understanding how such notions diverge from exact replicability is often a non-trivial challenge.

---

> > ### Author Rebuttal · Reviewer_Ew9n · 2026-04-03
> >
> > Thank you for the detailed response, all of my questions were resolved and I maintain my recommendation for acceptance.

---

### Official Review · Reviewer_y52N · 2026-03-19

**Soundness:** 3
**Presentation:** 3
**Significance:** 3
**Originality:** 3
**Overall Recommendation:** 4
**Confidence:** 3

**Summary:**

This paper studies the sample complexity of replicable reinforcement learning (RL) in tabular MDPs. The central question is whether exploration imposes an inherent additional cost for replicability beyond what is needed in the batch/generative model setting. It shows that a replicable RL algorithm in the episodic setting can achieve $\tilde{O}(S^2A)$ sample complexity in terms of $S$ and $A$.

**Compliance With Llm Reviewing Policy:**

Affirmed.

**Final Justification:**

I maintain my positive score as this work has obtained non-trivial improvements on the order of $S$ and $A$ for replicable RL.

**Key Questions For Authors:**

Please refer to the weaknesses above.

**Limitations:**

The paper briefly discusses some open problems.

**Strengths And Weaknesses:**

Strengths:
+ The upper and lower bounds almost match on $S$ and $A$, which are non-trivial to obtain.

+ The notion of "ignorable state combinations" and the fractional rounding strategy for making exploration outputs replicable without requiring a fully replicable exploration process is interesting.

Weaknesses:
- The sample complexity has a poly(H) factor ($H^{11}$ according to Theorem F.1), which is quite large and limits applicability. Simply ignoring such dependency and claiming that “exploration is not a significant barrier to replicable RL” is misleading. Did the paper achieve $S^2A$ dependence precisely because it was willing to tolerate $H^{11}$? Is there a concrete barrier to improving the H dependence? Is this an artifact of the backward induction approach or something more fundamental about replicability?

- The motivation for replicable RL is not very convincing. In practice, would it suffice to output policies that are close in some behavioral sense? Why does the algorithm must output almost identical policy?

- The distinction between this work and reward-free RL is not very clear. In some sense, they both try to learn an accurate enough transition model, based on which planning can be performed. What makes the replicable RL fundamentally more challenging than reward-free RL?

- The main result requires correlated sampling over a space of size $A^S$, which is exponential, and the efficient variant still requires multiple layers of subroutines, each with their own overhead. This hinders the applicability of the proposed algorithms. The paper would be more convincing with some simple experimental validation.

---

> ### Author Rebuttal · Authors · 2026-03-31
>
> We thank the reviewer for their careful reading and their positive feedback regarding our results, and novel techniques. We appreciate their thoughtful comments and address specific weaknesses and questions below.
>
> **Horizon Dependence**
> Improving the sample dependency on H is an important and challenging direction, and one that took many years to resolve even in the non-replicable setting. Previous work in replicable RL [EHKS23] incurs similarly poor dependence on H to our methods (in fact $H^{10} \varepsilon^{-10}$) sample complexity in addition to the $S^{7} A^{7}$ dependence which we improve significantly upon. We focus in this work on improving the latter dependence on the state space $S$, the action space $A$, and the approximation factor $\varepsilon$ to near-optimal. Improving the H dependency to near-optimal likely requires substantially new ideas. For example, the underlying non-replicable Q-learning algorithm for identifying ignorable states we used from [JAZBJ18] already possessed a non-optimal dependency on $H$. It is not clear at all whether the sample-optimal tabular RL algorithm from [ZCLD24] will suffice for the purposes of ignorable state identification. Given that our submission already presents significantly improved results that require substantially new ideas and technical effort, we believe it is reasonable to defer improvements in the horizon, which also likely require substantially new ideas, to future work. We will modify the text to better emphasize the importance of this direction.
>
> **Motivation for Replicable RL and Relaxation of Replicability**
> We agree that relaxations of replicability are interesting objects of study, and determining the right relaxation, e.g., what notion of closeness between policies should be required and its resulting sample cost is an interesting avenue for future work. That said, we emphasize the strong notion of replicability studied in this work is well-studied in the literature, and comes with several important advantages not shared by relaxed “close” notions:
>
> 1. Replicability is preserved under arbitrary post-processing: given that RL is used widely in applications, replicability guarantees that outputs are stable in any potential applications.
>
> 2. Replicability is easy to amplify: there are efficient ways to amplify any replicable algorithm to arbitrarily high replicability parameters.
>
> 4. Replicability has well-established connections with many notions of stability (see e.g. [BGH+23]) and in fact open problems in differential privacy (DP) have been resolved via studying replicability.
>
> **Comparison with Reward-Free RL**
> Although our techniques bear some high-level conceptual overlap with the ideas of reward-free RL in terms of “reachable states”, we would like to point out that the actual definition of reachability, the technical algorithm for finding "reachable" states, and most importantly the goals of reward-free RL and replicable RL are all different. Indeed it was not clear a priori reward-free RL has anything to do with replicability, since the former does not guarantee any kind of strong stability to the best of our knowledge.
>
> **Computational Efficiency**
> Our most sample-efficient algorithm relies on the correlated sampling procedure, which is not known to have a polynomial time algorithm. In turn, we do provide a computationally efficient replicable RL algorithm with an additional $S$ factor, which is still a substantial improvement from the sample complexity of previous algorithms, which scales as $S^7$ with the size of the state space. See Remark 1.6 and Theorem H.5.
>
> We would like to emphasize that such computational-statistical gaps are common for a wide range of replicable algorithmic tasks (e.g., mean estimation and sign-one-way marginals problems). Indeed, there are concrete barriers to closing such gaps — [HIK+24] demonstrates that such an algorithm would lead to a breakthrough in computational geometry. Since our lower bound Theorem 1.4 is accomplished via a reduction from the sign-one-way marginals problem, computationally efficient replicable RL inherits this same barrier.
>
> Regarding experimental validation, we emphasize that our paper is primarily a theoretical contribution, and that our algorithms are already significantly more efficient than previous work [EHKS23] and introduce substantially new ideas and algorithmic methods for stability. Many influential theory works in NeurIPS/ICML/etc, including many sample-bounds in RL theory (e.g., [JKMY20]), do not have corresponding numerical experiments, and while implementing and practically optimizing these algorithms is certainly an important and worthwhile direction for future work, the theory alone already provides substantial value.
>
> References
>
> [HIK+24] Replicability in High Dimensional Statistics, FOCS 2024
> [EHKS23] Replicable Reinforcement Learning, NeurIPS 2023
> [JKMY20] Reward-Free Exploration for Reinforcement Learning, ICML ‘20

---

> > ### Author Rebuttal · Reviewer_y52N · 2026-04-01
> >
> > I understand the difficulty of reducing the dependency on $H$. It remains unclear to me whether the reduced order on $S$ and $ A$ comes at the cost of increased order of $H$. I will keep my rating.

---

> > > ### Author Response · Authors · 2026-04-05
> > >
> > > We thank the reviewer for their acknowledgement and agree that significantly reducing the horizon dependence is a challenging and interesting avenue for future work beyond the scope of this rebuttal. That being said, we emphasize that previous work incurred similarly high dependence on $H$ (namely $H^{10}/\varepsilon^{10}$): our improvement to near-optimal sample complexity in S, A, and $\varepsilon$ does not come at the cost of significantly higher dependence on horizon compared to prior work ($H^{11}$ vs $H^{10}$). It’s also plausible the $H$-dependence of our algorithm can be shaved to $H^9$ using a more optimized non-replicable learner, though this would require a great deal of technical effort. In the authors’ opinion the more important question is whether $H^3$ can be achieved (and if not, what is the optimal overhead), which is likely to require substantially new ideas.

---

### Official Review · Reviewer_U7Ln · 2026-03-24

**Soundness:** 3
**Presentation:** 3
**Significance:** 3
**Originality:** 3
**Overall Recommendation:** 4
**Confidence:** 2

**Summary:**

This paper studies Replicable RL for low-horizon, tabular MDPs and shows the existence of an algorithm with improved sample complexity, specifically in terms of state and action space sizes (S and A), of order O(S^2A). Moreover, they show that this dependence is tight in S.

**Compliance With Llm Reviewing Policy:**

Affirmed.

**Final Justification:**

I am leaning towards accept given the interesting setting and the challenges in the analysis. My main concerns regarding the algorithm complexity and possible extensions to linear MDPs were reasonably answered and hence I maintain my positive score.

**Key Questions For Authors:**

Please see weaknesses listed above.

**Limitations:**

Yes.

**Strengths And Weaknesses:**

Strengths:

* Problem setting (Replicable RL) seems important yet less studied.
* The problem (tightening existing guarantees) appears genuinely challenging, and the analysis used in the paper contains novel components and is not just a combination of existing techniques.
* The paper was written clearly enough for a non-expert in replicable RL to understand key parts (though understanding the proof details remained challenging, which may be due to the topic). The exposition states which parts of the analysis are inspired by existing work and which are newly introduced here (e.g., the comparison with Jin et al. 2018 and Zhang et al. 2020).

Weaknesses:

* Although this is a theoretical paper, the gap between the theoretical results and practical applicability remains large. It would really help if any insights could be given that transfer to real-world settings. The main bottlenecks I see are the following:
  * This work considers tabular MDPs (finite states and actions), with no function approximation. How likely are the results here to at least transfer to the linear MDP case (which is considered a natural extension of the tabular case and also commonly considered in many theoretical works)?
  * Furthermore, the algorithm for the “main result” (Theorem 1.3) is extremely computationally intensive (runtime is exponential in S and horizon H). As a result, the authors provide an alternative approach with slightly worse sample-complexity dependence on S (S^3 from S^2) and whose runtime is polynomial in the sample complexity, poly(1/\epsilon). What is the degree of poly(1/\epsilon) approximately? If it is linear or quadratic, that is not too bad. But if it is a really high-degree polynomial, then it seems too prohibitive for applications (especially since the paper mentions game environments a few times as motivation for low-horizon MDPs, which, anyway, have access to cheap samples from simulators, negating the sample-complexity result in view of the very high runtime).
  * Also, the low-horizon condition seems a little restrictive. It would be interesting to see an extension to large horizons.
* Minor points: Unify to 'runtime' (or ‘run-time’) instead of 'runtime' and 'run-time.' Also, clarity would be improved if the algorithms were mentioned alongside or near the theorem statements, rather than just stating that they exist in the theorem statements and providing them in the appendix.

---

> ### Author Rebuttal · Authors · 2026-03-31
>
> We thank the reviewer for their careful reading and their positive feedback regarding our results, novel techniques, and paper presentation. We appreciate their thoughtful comments and will incorporate their recommendations into our revision.
>
> **Extension to Linear MDPs**
> We thank the reviewer for highlighting the natural extension to linear Markov Decision Processes (MDPs). This is a very interesting problem, and in fact it is only very recent (appearing online several months after our work) that any replicable algorithm for linear MDPs was known at all (see [EH+25]).
>
> The core algorithmic framework we develop offers a promising pathway to significantly improve upon [EH+25]’s independent replicable RL algorithm with linear function approximation. Specifically, our tabular concepts map to the linear setting in two key ways:
>
> One-Shot Rounding via Decoupled Exploration: Current linear approaches adopted by [EH+25] tightly intertwine decision-making and exploration by replicably updating the empirical uncentered covariance matrix $\Sigma \in \mathbb{R}^{d \times d}$ at every round to maintain Upper Confidence Bound (UCB) bonuses. By porting our decoupling strategy, future work could instead deploy a non-replicable reward-free agent to collect a dataset spanning the reachable feature space. Rounding the final $\Sigma$ just once and projecting it back to the positive semidefinite cone would replicably identify the dominant subspace of reachable features, effectively reducing the episodic linear RL problem to a highly efficient generative model setting.
>
> Tiered Precision for Linear Function Approximation: Our concept of tiered backward induction naturally extends to the continuous analogue of discrete state reachability: the variance captured along the eigenvectors of $\Sigma \in \mathbb{R}^{d \times d}$. By enforcing looser rounding grids or higher regularization parameters for the subspace spanned by minor eigenvectors (directions rarely encountered by any policy), the sample complexity budget can be reduced to ensure sufficient accuracy only along the principal components of the feature space.
>
> [EH+25] Eaton, Eric, et al. "Replicable reinforcement learning with linear function approximation." arXiv preprint arXiv:2509.08660 (2025).
>
> **Complexity of computationally efficient algorithm**
> Our computationally efficient algorithm has sample dependence $\varepsilon^{-2}$ (see Theorem H.5), which is the tight dependence.
> Regarding the runtime, our subroutines all run in either linear in the total number of time-dependent state-action pairs ($O(SAH)$) or linear in the total number of trajectories collected times the size of the action space and the horizon factor ($O(n A H)$). Thus, the total runtime is indeed still quadratic in $\varepsilon^{-1}$. We will emphasize this in the final text.
>
> **Horizon Dependence**
> Improving the sample dependency on H is an important and challenging direction, and one that took many years to resolve even in the non-replicable setting. Previous work in replicable RL [EHKS23] incurs similarly poor dependence on H to our methods (in fact $H^{10} \varepsilon^{-10}$) sample complexity in addition to the $S^{7} A^{7}$ dependence which we improve significantly upon. We focus in this work on improving the latter dependence on the state space $S$, the action space $A$, and the approximation factor $\varepsilon$ to near-optimal. Given that our submission already presents significantly improved results that require substantially new ideas and technical effort, we believe it is reasonable to defer improvements in the horizon, which also likely require substantially new ideas, to future work. We will modify the text to better emphasize the importance of this direction.

---

> > ### Author Rebuttal · Reviewer_U7Ln · 2026-04-03
> >
> > Thank you, all my questions have been addressed satisfactorily and I maintain my positive score for this paper;

---

### Decision · Program_Chairs · 2026-04-30

**Decision:**

Accept (regular)

**Comment:**

The reviewers agree that this paper presents a technically sound and mathematically rigorous contribution to the theory of replicable reinforcement learning. The primary strengths of the work lie in its novel decoupling of exploration from decision-making, yielding near-optimal sample complexity bounds that significantly bridge the theoretical gap between generative and episodic settings. During the discussion phase, reviewers raised valid concerns regarding the substantial polynomial dependence on the horizon, the computational inefficiency of the sample-optimal variant, and the absence of numerical experiments. I have carefully read the authors' rebuttals and confidential comments addressing these points, particularly regarding the evaluation criteria; I concur that the establishment of these theoretical bounds and the proposed algorithmic framework hold substantial standalone value for the learning theory community, rendering empirical validation unnecessary for acceptance in this context. While the high horizon dependence remains a practical limitation, the authors adequately defended this as a well-understood theoretical barrier that does not detract from the clear improvements made with respect to the state and action spaces. Contingent upon the authors fulfilling their commitment to reorganize the main text to properly introduce algorithmic definitions in the camera-ready version, this paper provides a useful and original theoretical advancement and is recommended for acceptance.